# RL4CO: an Extensive Reinforcement Learning for Combinatorial Optimization Benchmark

**Federico Berto**[*1] , **Chuanbo Hua**[*1], **Junyoung Park**[*1,2], **Minsu Kim**[1],
**Hyeonah Kim**[1], **Jiwoo Son**[1], **Haeyeon Kim**[1], **Joungho Kim**[1], **Jinkyoo Park**[1,2]
[1] Korea Advanced Institute of Science and Technology (KAIST)
[2] OMELET

## Abstract

We introduce RL4CO, an extensive reinforcement learning (RL) for combinatorial optimization (CO) benchmark. RL4CO employs state-of-the-art software libraries as well as best practices in implementation, such as modularity and configuration management, to be efficient and easily modifiable by researchers for adaptations of neural network architecture, environments, and RL algorithms. Contrary to the existing focus on specific tasks like the traveling salesman problem (TSP) for performance assessment, we underline the importance of scalability and generalization capabilities for diverse CO tasks. We also systematically benchmark zero-shot generalization, sample efficiency, and adaptability to changes in data distributions of various models. Our experiments show that some recent SOTA methods fall behind their predecessors when evaluated using these metrics, suggesting the necessity for a more balanced view of the performance of neural CO (NCO) solvers. We hope RL4CO will encourage the exploration of novel solutions to complex real-world tasks, allowing the NCO community to compare with existing methods through a standardized interface that decouples the science from software engineering. We make our library publicly available at https://github.com/kaist-silab/rl4co.

## 1 Introduction

Combinatorial optimization (CO) is a mathematical optimization area that encompasses a wide variety of important practical problems, such as routing problems and hardware design, whose solution space typically grows exponentially to the size of the problem (also often referred to as NP-hardness). As a result, CO problems can take considerable expertise to craft solvers and raw computational power to solve. Neural Combinatorial Optimization (NCO) [7; 44; 56] provides breakthroughs in CO by leveraging recent advances in deep learning, especially by automating the design of solvers and considerably improving the efficiency in providing solutions. While conventional operations research (OR) approaches [17; 23; 69] have achieved significant progress in CO, they encounter limitations when addressing new CO tasks, as they necessitate extensive expertise. In contrast, NCO trained with reinforcement learning (RL) overcomes the limitations of OR-based approaches (i.e., manual designs) by harnessing RL's ability to learn in the absence of optimal solutions.[2] NCO presents possibilities as a general problem-solving approach in CO, handling chal-

---

[*]Equal contribution authors

[2]Supervised learning approaches also offer notable improvements; However, their use is restricted due to the requirements of (near) optimal solutions during training.

Submitted to the 37th Conference on Neural Information Processing Systems (NeurIPS 2023) Track on Datasets and Benchmarks. Do not distribute.

lenging problems with minimal dependent (or even independent) of problem-specific knowledge [6; 38; 40; 36; 24; 5; 4; 2].

Among CO tasks, the routing problems, such as Traveling Salesman Problem (TSP) and Capacitated Vehicle Routing Problem (CVRP), serve as one of the central test suites for the capabilities of NCO due to the extensive NCO research on that types of problems [49; 38; 40; 36] and also, the applicability of at-hand comparison of highly dedicated heuristic solvers investigated over several decades of study by the OR community [17; 23]. Recent advances [20; 42; 30] of NCO achieve comparable or superior performance to state-of-the-art solvers on these benchmarks, implying the potential of NCO to revolutionize the laborious manual design of CO solvers [69; 63].

However, despite the successes and popularity of RL for CO, the NCO community still lacks unified implementations of NCO solvers for easily benchmarking different NCO solvers. Similar to the other ML research, in NCO research, a unified open-source software would serve as a cornerstone for progress, bolstering reproducibility, and ensuring findings can be reliably validated by peers. This would provide a flexible and extensive RL for CO foundation and a unified library can thus bridge the gap between innovative ideas and practical applications, enabling convenient training and testing of different solvers under new settings, and decoupling science from engineering. In practice, this would also serve to expand the NCO area and make it accessible to researchers and practitioners.

Another problem that NCO research faces is the absence of standardized evaluation metrics that, especially account for the practical usage of CO solvers. Although most NCO solvers are customarily assessed based on their performance within training distributions [38; 40; 36], ideally, they should solve CO problems from out-of-training-distribution well. However, such out-of-distribution evaluation is overlooked in the literature. Furthermore, unlike the other ML research that already has shown the importance of the volume of training data, in NCO, the evaluation of the methods with the controls on the number of training samples is not usually discussed (e.g., state-of-the-art methods can underperform than the other methods). This also hinders the use of NCO in the real world, where the evaluation of solutions becomes expensive (e.g., evaluation of solutions involves the physical dispatching of goods in logistic systems or physical design problems) [14; 35; 2].

**Contributions.** In this work, we introduce RL4CO, a new reinforcement learning (RL) for combinatorial optimization (CO) benchmark. RL4CO is first and foremost a library of several environments, baselines and boilerplate from the literature implemented in a *modular*, *flexible*, and *unified* way with what we found are the best software practices and libraries, including TorchRL [47], PyTorch Lightning [18], TensorDict [46] and Hydra [74]. Through thoroughly tested unified implementations, we conduct several experiments to explore best practices in RL for CO and benchmark our baselines. We demonstrate that existing state-of-the-art methods may perform poorly on different evaluation metrics and sometimes even underperform their predecessors. We also introduce a new Pareto-optimal, simple-yet-effective sampling scheme based on greedy rollouts from random symmetric augmentations. Additionally, we incorporate real-world tasks, specifically hardware design, to highlight the importance of sample efficiency in scenarios where objective evaluation is black-box and expensive, further validating that the functionally decoupled implementation of RL4CO enhances accessibility for achieving better performance in a variety of tasks.

## 2 Preliminaries

The solution space of CO problems generally grows exponentially to their size. Such solution space of CO hinders the learning of NCO solvers that generate the solution in a single shot[3]. As a way to mitigate such difficulties, the *constructive* (e.g., [49; 70; 38; 40; 36]) methods generate solutions one step at a time in an autoregressive fashion, akin to language models [13; 68; 50]. In RL4CO we focus primarily on benchmarking autoregressive approaches for the above reasons.

---

[3]Also known as non-autoregressive approaches (NAR) [21; 31; 39; 66]. Imposing the feasibility of NAR-generated solutions is also not straightforward, especially for CO problems with complicated constraints.

**Solving Combinatorial Optimization with Autoregressive Sequence Generation**   Autoregressive (or *constructive*) methods assume the autoregressive solution construction schemes, which decide the next "action" based on the current (partial) solution, and repeat this until the solver generates the complete solution (e.g., in TSP, the next action is deciding on a city to visit). Formally speaking,

$$\pi(\boldsymbol{a}|\boldsymbol{x}) \triangleq \prod_{t=1}^{T-1} \pi(a_t|a_{t-1},...a_1,\boldsymbol{x}), \tag{1}$$

where $\boldsymbol{a} = (a_1, ..., a_T)$, $T$ is the solution construction steps, is a feasible (and potentially optimal) solution to CO problems, $\boldsymbol{x}$ is the problem description of CO, $\pi$ is a (stochastic) solver that maps $\boldsymbol{x}$ to a solution $\boldsymbol{a}$. For example, for a 2D TSP with $N$ cities, $\boldsymbol{x} = \{(x_i, y_i)\}_{i=1}^N$, where $(x_i, y_i)$ is the coordinates of $i$th city $v_i$, a solution $a = (v_1, v_2, ...v_N)$.

**Training NCO Solvers via Reinforcement Learning**   The solver $\pi_\theta$ parameterized with the parameters $\theta$ can be trained with supervised learning (SL) or RL schemes. In this work, we focus on RL-based solvers as they can be trained without relying on the optimal (or high-quality) solutions Under the RL formalism, the training problem of NCOs becomes as follows:

$$\theta^* = \underset{\theta}{\operatorname{argmax}} \Big[ \mathbb{E}_{\boldsymbol{x} \sim P(\boldsymbol{x})} \big[ \mathbb{E}_{a \sim \pi_\theta(\boldsymbol{a}|\boldsymbol{x})} R(\boldsymbol{a}, \boldsymbol{x}) \big] \Big], \tag{2}$$

where $P(\boldsymbol{x})$ is problem distribution, $R(\boldsymbol{a}, \boldsymbol{x})$ is reward (i.e., the negative cost) of $\boldsymbol{a}$ given $\boldsymbol{x}$.

To solve Eq. (2) via gradient-based optimization method, calculating the gradient of the objective function w.r.t. $\theta$ is required. However, due to the discrete nature of the CO, the computation of the gradient is not straightforward and often requires certain levels of approximation. Even though few researchers show breakthroughs for solving Eq. (2) with gradient-based optimization, they are restricted to some relatively simpler cases of CO problems [58; 60; 72]. Instead, it is common to rely on RL-formalism to solve Eq. (2). In theory, value-based methods [33] and policy gradient methods [38; 40; 36; 53], and also actor-critic methods [52; 75] are applicable to solve Eq. (2). However, in practice, it is shown that the policy gradient methods (e.g., REINFORCE [73] with proper baselines), generally outperform the value-based methods [38] in NCO.

**General Structure of Autoregressive Policies**   The autoregressive NCO solver (i.e., policy) *encodes* the given problem $\boldsymbol{x}$ and auto-regressively *decodes* the solution. This can be seen as a processing input problem with the encoder and planning (i.e., computing a complete solution) with the decoder. To maximize the solution-finding speed, a common design of the decoder is to fuse the RL environment (e.g., TSP solution construction schemes that update the partial solutions and constraints of CO as well) into the decoder. This aspect of NCO policy is distinctive from the other RL tasks, which maintains the environment separately from the policy. As a result, most competitive autoregressive NCO solver implementations show significant coupling with network architecture and targeting CO problems. This can hinder the reusability of NCO solver implementation to the new types of CO problems. Furthermore, this design choice introduces difficulties for the fairer comparison among the trained solvers, especially related to the effect of encoder/decoder architectures and training/evaluation data usage on the solver's solution qualities.

# 3   RL4CO

In this paper, we present RL4CO, an extensive reinforcement learning (RL) for Combinatorial Optimization (CO) benchmark. RL4CO aims to provide a *modular*, *flexible*, and *unified* code base that addresses the challenges of autoregressive policy training/evaluation for NCO (discussed in Section 2) and performs extensive benchmarking capabilities on various settings.

## 3.1   Unified and Modular Implementation

As shown in Fig. 3.1, RL4CO decouples the major components of the autoregressive NCO solvers and its training routine while prioritizing reusability. We consider the five major components, which are explained in the following paragraphs.

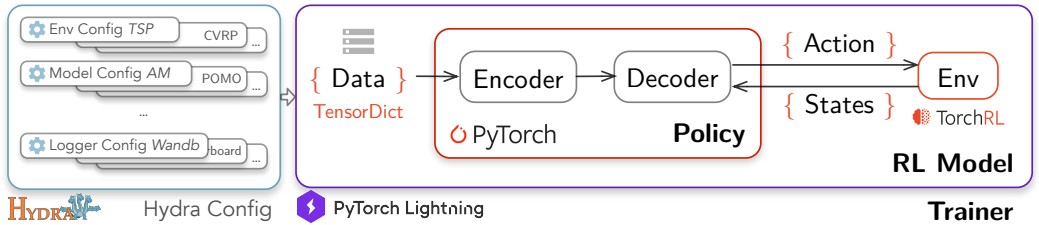

Figure 3.1: An overview of RL4CO. Our goal is to provide a unified framework for RL-based CO algorithms, and to facilitate reproducible research in this field, decoupling the science from the engineering.

**Policy** This module is responsible for constructing solutions for CO problems autoregressively. Our initial investigation into various autoregressive NCO solvers, such as AM, POMO, Sym-NCO, across CO problems like Traveling TSP, Capacitated Vehicle Routing Problem (CVRP), Orienteering Problem (OP), Prize-collecting TSP (PCTSP), among others, has revealed a common structural pattern. The policy network $\pi_\theta$ follows an architecture that combines an encoder $f_\theta$ and a decoder $g_\theta$ as follows:

$$\pi_\theta(\boldsymbol{a}|\boldsymbol{x}) \triangleq g_\theta(f_\theta(\boldsymbol{x})) \tag{3}$$

Upon analyzing encoder-decoder architectures, we have identified components that hinder the encapsulation of the policy from the environment. To achieve greater modularity, RL4CO modularizes such components in the form of *embeddings*: `InitEmbedding`, `ContextEmbedding` and `DynamicEmbedding` [4].

The encoder's primary task is to encode input $\boldsymbol{x}$ into a hidden embedding $\boldsymbol{h}$. The structure of $f_\theta$ comprises two trainable modules: the `InitEmbedding` and encoder blocks. The `InitEmbedding` module typically transforms problem features into the latent space and problem-specific compared to the encoder blocks, which often involve plain multi-head attention (MHA):

$$\boldsymbol{h} = f_\theta(\boldsymbol{x}) \triangleq \text{EncoderBlocks}(\texttt{InitEmbedding}(\boldsymbol{x})) \tag{4}$$

The decoder autoregressively constructs the solution based on the encoder output $\boldsymbol{h}$. Solution decoding involves iterative steps until a complete solution is constructed:

$$q_t = \texttt{ContextEmbedding}(\boldsymbol{h}, a_{t-1:0}), \tag{5}$$

$$\bar{q}_t = \text{MHA}(q_t, W_k^g \boldsymbol{h}, W_v^g \boldsymbol{h}), \tag{6}$$

$$\pi(a_t) = \text{MaskedSoftmax}(\bar{q}_t \cdot W_v \boldsymbol{h}, M_t), \tag{7}$$

where the `ContextEmbedding` is tailored to the specific problem environment, $q_t$ and $\bar{q}_t$ represent the query and attended query (also referred to as glimpse in Mnih et al. [45]) at the $t$-th decoding step, $W_k^g$, $W_v^g$ and $W_v$ are trainable linear projections computing keys and values from $\boldsymbol{h}$, and $M_t$ denotes the action mask, which is provided by the environment to ensure solution feasibility. It is noteworthy that we also modularize the `DynamicEmbedding`, which dynamically updates the keys and values of MHA and Softmax during decoding. This approach is often used in dynamic routing settings, such as split delivery VRP. For the details, please refer to Appendix A.4.

From Eqs. (4) and (5), it is evident that creating embeddings demands problem-specific handling, often trigger coherence between the policy and CO problems. In RL4CO, we offer pre-coded environment embeddings investigated from NCO literature [35; 38; 41] and, more importantly, allow a drop-in replacement of pre-coded embedding modules to user-defined embedding modules to attain higher modularity. Furthermore, we accommodate various decoding schemes (which will be further discussed in § 4) proposed from milestone papers [38; 40; 36] into a unified decoder implementation so that those schemes can be applied to the different model, such as applying greedy multi-starts to the Attention Model from Kool et al. [38].

**Environment** This module fully specifies the problem, updates the problem construction steps based on the input action and provides the result of updates (e.g., action masks) to the policy

---

[4]Also available at: https://rl4co.readthedocs.io/en/latest/_content/api/models/env_embeddings.html

module. When implementing the `environment`, we focus on parallel execution of rollouts (i.e., problem-solving) while maintaining *statelessness* in updating every step of solution decoding. These features are essential for ensuring the reproducibility of NCO and supporting "look-back" decoding schemes such as Monte-Carlo Tree Search. Our environment designs and implementations are flexible enough to accommodate various types of NCO solvers that generate a single action $a_t$ at each decision-making step [3; 33; 52; 53; 75]. Additionally, our framework is extensible beyond routing problems. We investigate the use of RL4CO for electrical design automation in Appendix B.

Our environment implementation is based on `TorchRL` [10], an open-source RL library for `PyTorch` [54], which aims at high modularity and good runtime performance, especially on GPUs. This design choice makes the `Environment` implementation standalone, even outside of RL4CO, and consistently empowered by a community-supporting library – `TorchRL`. Moreover, we employ `TensorDicts` [46] to move around data which allows for further flexibility.

**RL Algorithm**   This module defines the routine that takes the `Policy`, `Environment`, and problem instances and computes the gradients of the policy (and possibly the critic for actor-critic methods). We intentionally decouple the routines for gradient computations and parameter updates to support modern training practices, which will be explained in the next paragraph.

**Trainer**   Training a single NCO model is typically computationally demanding, especially since most CO problems are NP-hard. Therefore, implementing a modernized training routine becomes crucial. To this end, we implement the `Trainer` using `Lightning` [18], which seamlessly supports features of modern training pipelines, including logging, checkpoint management, automatic mixed-precision training, various hardware acceleration supports (e.g., CPU, GPU, TPU, and Apple Silicon), multi-GPU support, and even multi-machine expansion. We have found that using mixed-precision training significantly decreases training time without sacrificing NCO solver quality and enables us to leverage recent routines such as FlashAttention [16; 15].

**Configuration Management**   Optionally, but usefully, we adopt `Hydra` [74], an open-source Python framework that enables hierarchical config management. It promotes modularity, scalability, and reproducibility, making it easier to manage complex configurations and experiments with different settings and maintain consistency across different environments.

## 3.2   Availability and Future Support

RL4CO can be installed through PyPI [1][5] and we adhere to continuous integration, deployment, and testing to ensure reproducibility and accessibility.[6]

```
1    $ pip install rl4co
```

Listing 1: Installation of RL4CO with PyPI

Our goal is to provide long-term support for RL4CO. It is actively maintained and will continue to update to accommodate new features and contributions from the community. Ultimately, our aim is to make RL4CO the to-go library in the RL for CO research area that provides encompassing, accessible, and extensive boilerplate code.

## 4   Benchmark Experiments

Our focus is to benchmark the NCO solvers under controlled settings, aiming to compare all benchmarked methods as closely as possible in terms of network architectures and the number of training samples consumed.

---

[5]Listed at https://pypi.org/project/rl4co/

[6]Documentation is also available on ReadTheDocs: https://rl4co.readthedocs.io/en/latest/

Table 4.1: In-domain benchmark results. Gurobi † [22] results are reproduced from [38]. As the non-learned heuristic baselines, we report the results of LKH3 [23] and algorithm-specific methods. For TSP, we used Concorde [48] as the classical method baseline. For CVRP, we used HGS [69] as the classical method baseline. The gaps are measured w.r.t. the best classical heuristic methods.

| Method | TSP ($N = 20$) | | | TSP ($N = 50$) | | | CVRP ($N = 20$) | | | CVRP ($N = 50$) | | |
|---|---|---|---|---|---|---|---|---|---|---|---|---|
| | Cost ↓ | Gap | Time | Cost ↓ | Gap | Time | Cost ↓ | Gap | Time | Cost ↓ | Gap | Time |
| *Gurobi*† | 3.84 | – | 7s | 5.70 | – | 2m | 6.10 | – | – | – | – | – |
| *Concorde* | 3.84 | 0.00% | 1m | 5.70 | 0.00% | 2m | | | N/A | | | |
| *HGS* | | | N/A | | | | 6.13 | 0.00% | 4h | 10.37 | 0.00% | 10h |
| *LKH3* | 3.84 | 0.00% | 15s | 5.70 | 0.00% | (<5m) | 6.14 | 0.00% | 5h | 10.38 | 0.00% | 12h |
| *Greedy One Shot Evaluation* | | | | | | | | | | | | |
| AM-critic | 3.86 | 0.64% | (<1s) | 5.83 | 2.22% | (<1s) | 6.46 | 5.00% | (<1s) | 11.16 | 7.09% | (<1s) |
| AM | 3.84 | 0.19% | (<1s) | 5.78 | 1.41% | (<1s) | 6.39 | 3.92% | (<1s) | 10.95 | 5.30% | (<1s) |
| POMO | 3.84 | 0.18% | (<1s) | 5.75 | 0.89% | (<1s) | 6.33 | 3.00% | (<1s) | 10.80 | 3.99% | (1s) |
| Sym-NCO | 3.84 | 0.05% | (<1s) | 5.72 | 0.47% | (<1s) | 6.30 | 2.58% | (<1s) | 10.87 | 4.61% | (1s) |
| AM-XL | 3.84 | 0.07% | (<1s) | 5.73 | 0.54% | (<1s) | 6.31 | 2.81% | (<1s) | 10.84 | 4.31% | (1s) |
| *Sampling with width $M = 1280$* | | | | | | | | | | | | |
| AM-critic | 3.84 | 0.15% | 20s | 5.74 | 0.72% | 40s | 6.26 | 2.08% | 24s | 10.70 | 3.07% | 1m24s |
| AM | 3.84 | 0.04% | 20s | 5.72 | 0.40% | 40s | 6.24 | 1.78% | 24s | 10.60 | 2.22% | 1m24s |
| POMO | 3.84 | 0.02% | 36s | 5.71 | 0.18% | 1m | 6.20 | 1.06% | 40s | 10.54 | 1.64% | 2m3s |
| Sym-NCO | 3.84 | 0.01% | 36s | 5.70 | 0.14% | 1m | 6.22 | 1.44% | 40s | 10.58 | 2.03% | 2m3s |
| AM-XL | 3.84 | 0.02% | 36s | 5.71 | 0.17% | 1m | 6.22 | 1.46% | 40s | 10.57 | 1.91% | 2m3s |
| *Greedy Multistart ($N$)* | | | | | | | | | | | | |
| AM-critic | 3.85 | 0.36% | (<1s) | 5.80 | 1.81% | 2s | 6.33 | 3.04% | 3s | 10.90 | 4.86% | 6s |
| AM | 3.84 | 0.12% | (<1s) | 5.77 | 1.21% | 2s | 6.28 | 2.27% | 3s | 10.73 | 3.39% | 6s |
| POMO | 3.84 | 0.05% | (<1s) | 5.71 | 0.29% | 3s | 6.21 | 1.27% | 4s | 10.58 | 2.04% | 8s |
| Sym-NCO | 3.84 | 0.03% | (<1s) | 5.72 | 0.36% | 3s | 6.22 | 1.48% | 4s | 10.71 | 3.17% | 8s |
| AM-XL | 3.84 | 0.05% | (<1s) | 5.72 | 0.42% | 3s | 6.22 | 1.38% | 4s | 10.68 | 2.88% | 8s |
| *Greedy with Augmentation (1280)* | | | | | | | | | | | | |
| AM-critic | 3.84 | 0.01% | 20s | 5.71 | 0.18% | 40s | 6.22 | 1.35% | 24s | 10.63 | 2.49% | 1m24s |
| AM | 3.84 | 0.00% | 20s | 5.70 | 0.07% | 40s | 6.20 | 1.07% | 24s | 10.53 | 1.56% | 1m24s |
| POMO | 3.84 | 0.00% | 36s | 5.70 | 0.06% | 1m | 6.18 | 0.84% | 45s | 10.55 | 1.72% | 2m30s |
| Sym-NCO | 3.84 | 0.00% | 36s | 5.70 | 0.01% | 1m | 6.17 | 0.71% | 45s | 10.53 | 1.54% | 2m30s |
| AM-XL | 3.84 | 0.00% | 36s | 5.70 | 0.01% | 1m | 6.17 | 0.68% | 45s | 10.52 | 1.47% | 2m30s |
| *Greedy Multistart with Augmentation ($N \times 16$)* | | | | | | | | | | | | |
| AM-critic | 3.84 | 0.01% | 9s | 5.72 | 0.41% | 32s | 6.20 | 1.12% | 48s | 10.67 | 2.81% | 1m |
| AM | 3.84 | 0.00% | 9s | 5.71 | 0.21% | 32s | 6.18 | 0.78% | 48s | 10.55 | 1.73% | 1m |
| POMO | 3.84 | 0.00% | 13s | 5.70 | 0.05% | 48s | 6.16 | 0.50% | 1m | 10.48 | 1.11% | 2m |
| Sym-NCO | 3.84 | 0.00% | 13s | 5.70 | 0.03% | 48s | 6.17 | 0.61% | 1m | 10.54 | 1.63% | 2m |
| AM-XL | 3.84 | 0.00% | 13s | 5.70 | 0.04% | 48s | 6.16 | 0.44% | 1m | 10.53 | 1.50% | 2m |

**TL; DR**   Here is a summary of the benchmark results.

- AM [38], with minor encoder modifications and trained with a sufficient number of samples, can at times outperform or closely match state-of-the-art (SOTA) methods such as POMO and Sym-NCO for TSP and CVRP with 20 and 50 nodes. (See § 4.1)

- The choice of decoding schemes has a significant impact on the solution quality of NCO solvers. We introduce a simple-yet-effective decoding scheme based on greedy augmentations that significantly enhances the solution quality of the trained solver. (See § 4.1)

- We find that in-distribution performance trends do not always match with out-of-distribution ones when testing with different problem sizes. (See § 4.2)

- When the number of samples is limited, the ranking of baseline methods can significantly change. Actor-critic methods can be a good choice in data-constrained applications. (See § 4.3)

- We find that in-distribution results may not easily determine the downstream performance of pre-trained models when search methods are used, and models that perform worse in-distribution may perform better during adaptation. (See § 4.4)

**Benchmarked Solvers**   We evaluate the following NCO solvers:

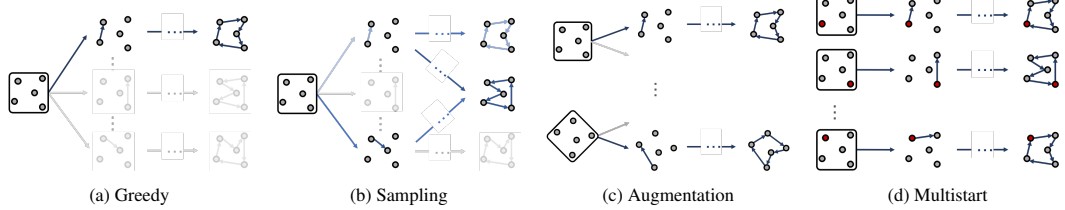

(a) Greedy       (b) Sampling       (c) Augmentation       (d) Multistart

Figure 4.1: Decoding schemes of the autoregressive NCO solvers evaluated in this paper.

- `AM` [38] employs the multi-head attention (MHA) encoder and single-head attention decoder trained using REINFORCE and the rollout baseline.

- `AM-Critic` evaluates the baseline using the learned critic.

- `POMO` [40] is an extension of AM that employs the shared baseline instead of the rollout baseline.

- `Sym-NCO` [36] introduces a symmetric baseline to train the AM instead of the rollout baseline.

- `AM-XL` is AM that adopts `POMO`-style MHA encoder, using six MHA layers and InstanceNorm instead of BatchNorm. We train `AM-XL` on the same number of samples as `POMO`.

For all benchmarked solvers, we schedule the learning rate with `MultiStepLinear`, which seems to have a non-negligible effect on the performances of NCO solvers - for instance, compared to the original AM implementation and with the same hyperparameters, we can consistently improve performance, i.e. greedy one-shot evaluation on TSP50 from 5.80 to 5.78 and on CVRP50 from 10.98 to 10.95. In addition to the NCO solvers, we compare them to SOTA classical solvers that specialize in solving specific types of CO problems.

**Decoding Schemes** The solution quality of NCO solvers often shows large variations in performances to the different decoding schemes, even though using the same NCO solvers. Regarding that, we evaluate the trained solvers using five schemes:

- `Greedy` elects the highest probabilities at each decoding step.

- `Sampling` concurrently samples $N$ solutions using a trained stochastic policy.

- `Multistart Greedy`, inspired by POMO, decodes from the first given nodes and considers the best results from $N$ cases starting at $N$ different cities. For example, in TSP with $N$ nodes, a single problem involves starting from $N$ different cities.

- `Augmentation` selects the best greedy solutions from randomly augmented problems (e.g., random rotation and flipping) during evaluation.

- `Multistart Greedy + Augmentation` combines the Multistart Greedy with Augmentation.

We emphasize that our work introduces the new greedy Symmetric `Augmentation` during evaluation, a simple-yet-effective scheme. POMO utilized the 'x8 augmentation' through the dihedral group of order 8. However, we found that generalized symmetric augmentations - even without multistarts - as in Kim et al. [36] can perform better than other decoding schemes. For a visual explanation of the decoding scheme, please refer to Fig. 4.1.

### 4.1 In-distribution Benchmark

We first measure the performances of NCO solvers on the datasets on which they are trained on. The results are summarized in Table 4.1. We first observe that, counter to the commonly known trends that AM < POMO < Sym-NCO, the trend can change to the selection of decoding schemes. Especially when the solver decodes the solutions with `Augmentation` or `Greedy Multistart + Augmentation`, the performance differences among the benchmarked solvers on TSP20/50, CVRP20/50 become insignificant. That implies we can improve the solution qualities by increasing the computational budget. These observations lead us to the requirements for an in-depth investigation of the sampling methods and their efficiency.

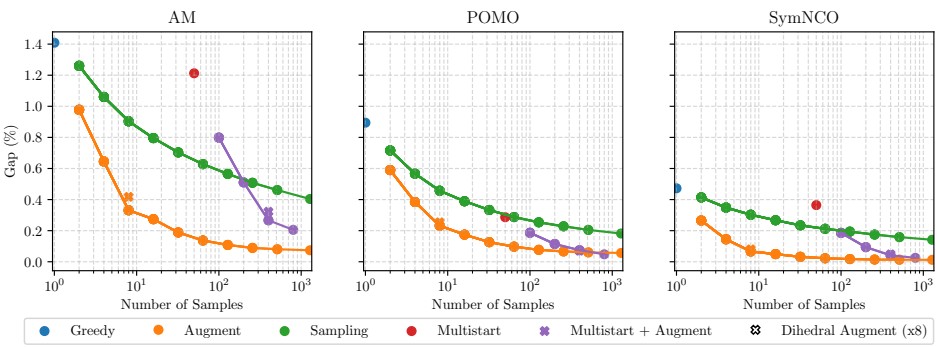

Figure 4.2: Pareto front of decoding schemes vs. number of samples on TSP50

Figure 4.3: Pareto front of decoding schemes vs. number of samples on CVRP50

**More Sampling, which Decoding Scheme?** Based on our previous findings, we anticipate that by investing more computational resources (i.e., increasing the number of samples), the trained NCO solver can discover improved solutions. In this investigation, we examine the performance gains achieved with varying numbers of samples on the TSP50 dataset. As shown in Fig. 4.2, all solvers demonstrate that the Augmentation decoding scheme achieves the Pareto front with limited samples and, notably, generally outperforms other decoding schemes. We observed a similar tendency in CVRP50 (see Fig. 4.3). Additional results on OP and PCTSP are available in Appendix E.

## 4.2 Out-of-distribution Benchmark

In this section, we evaluate the out-of-distribution performance of the NCO solvers by measuring the optimality gap compared to the best-known tractable solver. The evaluation results are visualized in § 4.2. Contrary to the in-distribution results, we find that NCO solvers with sophisticated baselines (i.e., POMO and Sym-NCO) tend to exhibit worse generalization when the problem size changes, either for solving smaller or larger instances. This can be seen as an indication of "overfitting" to the training sizes. On the other hand, the variant of AM shows relatively better generalization results overall. We also evaluate the solvers in two canonical public benchmark instances (TSPLib and CVRPLib) in Appendix F, which exhibit both variations in the number of nodes as well as their distributions and find a similar trend.

## 4.3 Sample Efficiency Benchamrk

We evaluate the NCO solvers based on the number of training samples (i.e., the number of reward evaluations). As shown in Fig. 4.5, we found that actor-critic methods (e.g., AM trained with PPO detailed in Appendix D.7 or AM Critic) can exhibit efficacy in scenarios with limited training samples, as demonstrated by the TSP50/100 results in Fig. 4.5. This observation suggests that NCO solvers with control over the number of samples may exhibit a different trend from the commonly recognized trends. In the extension of this viewpoint, we conducted additional benchmarks in a different problem domain: electrical design automation (EDA) where reward evaluation is resource-

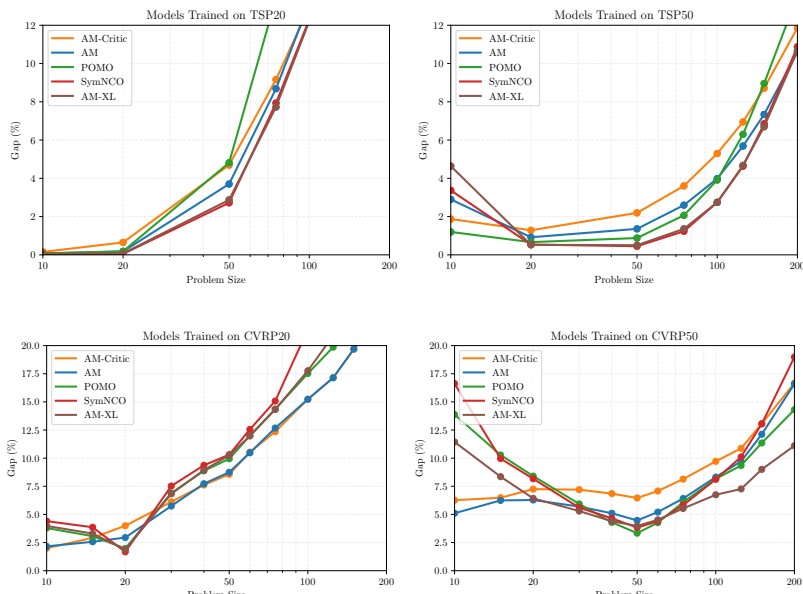

Figure 4.4: Out-of-distribution generalization results.

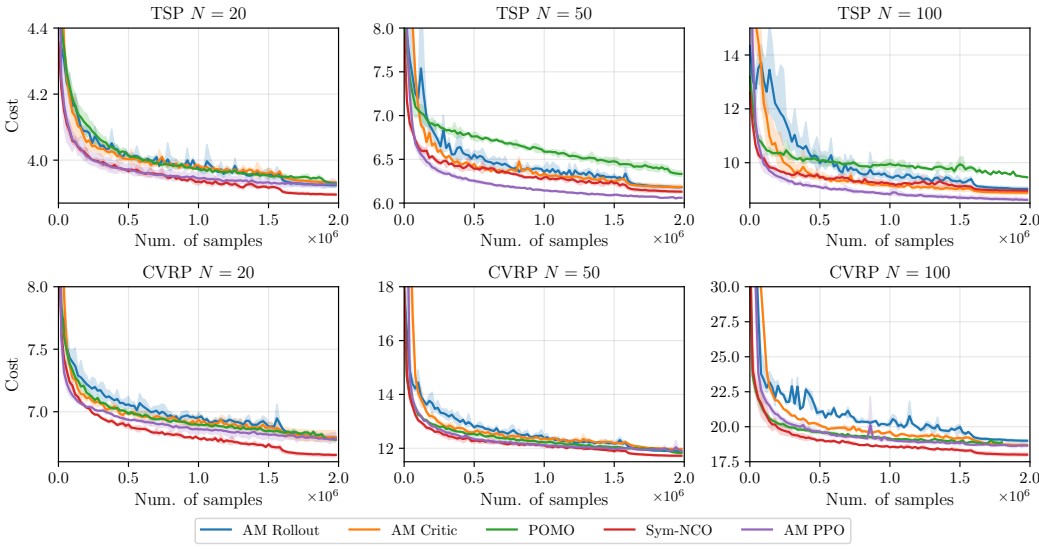

Figure 4.5: Validation cost over the number of training samples (i.e., number of reward evaluations).

intensive, due to the necessity of electrical simulations. Therefore, sample efficiency becomes even more critical. For more details, please refer to Appendix B.

## 4.4 Search Methods Benchmark

One viable and prominent approach of NCO that mitigates distributional shift (e.g., the size of problems) is the (post) search methods which involve training (a part of) a pre-trained NCO solver to adapt to CO instances of interest.

**Benchmarked Search Methods**   We evaluate the following search methods:

- Active Search (AS) from Bello et al. [6] finetunes a pre-trained model on the searched instances by adapting all the policy parameters.

- Efficient Active Search (EAS) from Hottung et al. [25] finetunes a subset of parameters (i.e., embeddings or new layers) and adds an imitation learning loss to improve convergence.

Table 4.2: Search Methods Benchmark results of models pre-trained on 50 nodes. We apply the search methods with default parameters from the literature. *Classic* refers to Concorde [17] for TSP and LKH3 [23] for CVRP. OOM denotes "Out of Memory", which occurred with AS on large-scale instances.

| Type | Metric | TSP | | | | | | CVRP | | | | | |
|---|---|---|---|---|---|---|---|---|---|---|---|---|---|
| | | POMO | | | Sym-NCO | | | POMO | | | Sym-NCO | | |
| | | 200 | 500 | 1000 | 200 | 500 | 1000 | 200 | 500 | 1000 | 200 | 500 | 1000 |
| *Classic* | Cost | 10.17 | 16.54 | 23.13 | 10.72 | 16.54 | 23.13 | 27.95 | 63.45 | 120.47 | 27.95 | 63.45 | 120.47 |
| *Zero-shot* | Cost | 13.15 | 29.96 | 58.01 | 13.30 | 29.42 | 56.47 | 29.16 | 92.30 | 141.76 | 32.75 | 86.82 | 190.69 |
| | Gap[%] | 29.30 | 81.14 | 150.80 | 24.07 | 77.87 | 144.14 | 4.33 | 45.47 | 17.67 | 17.17 | 36.83 | 58.29 |
| | Time[s] | 2.52 | 11.87 | 96.30 | 2.70 | 13.19 | 104.91 | 1.94 | 15.03 | 250.71 | 2.93 | 15.86 | 150.69 |
| *AS* | Cost | 11.16 | 20.03 | OOM | 11.92 | 22.41 | OOM | 28.12 | 63.98 | OOM | 28.51 | 66.49 | OOM |
| | Gap[%] | 4.13 | 21.12 | OOM | 11.21 | 35.48 | OOM | 0.60 | 0.83 | OOM | 2.00 | 4.79 | OOM |
| | Time[s] | 7504 | 10070 | OOM | 7917 | 10020 | OOM | 8860 | 21305 | OOM | 9679 | 24087 | OOM |
| *EAS* | Cost | 11.10 | 20.94 | 35.36 | 11.65 | 22.80 | 38.77 | 28.10 | 64.74 | 125.54 | 29.25 | 70.15 | 140.97 |
| | Gap[%] | 3.55 | 26.64 | 52.89 | 8.68 | 37.86 | 67.63 | 0.52 | 2.04 | 4.21 | 4.66 | 10.57 | 17.02 |
| | Time[s] | 348 | 1562 | 13661 | 376 | 1589 | 14532 | 432 | 1972 | 20650 | 460 | 2051 | 17640 |

**Results**  We extend RL4CO and apply AS and EAS to POMO and Sym-NCO pre-trained on TSP and CVRP with 50 nodes from § 4.1 to solve larger instances having $N \in [200, 500, 1000]$ nodes. As shown in Table 4.2, solvers with search methods improve the solution quality. However, POMO generally shows better improvements over Sym-NCO. This may again imply the "overfitting" of sophisticated baselines that can perform better in-training but eventually worse in downstream tasks.

# 5   Discussion

## 5.1   Future Directions in RL4CO

The utilization of symmetries in learning, such as by POMO and Sym-NCO, has its limitations in sample efficiency and generalizability, but recent studies like Kim et al. [34] offer promising results by exploring symmetries without reward simulation. There is also a trend toward few-shot learning, where models adapt rapidly to tasks and scales; yet, the transition from tasks like TSP to CVRP still requires investigation [43; 65]. Meanwhile, as AM's neural architecture poses scalability issues, leveraging architectures such as Hyena [59] that scale sub-quadratically might be key. Furthermore, the emergence of foundation models akin to LLMs, with a focus on encoding continuous features and applying environment-specific constraints, can reshape the landscape of NCO [68; 50]. Efficient finetuning methods could also be pivotal for optimizing performance under constraints [26; 67].

## 5.2   Limitations

We identify some limitations with our current benchmark. In terms of benchmarking, we majorly focus on training the solvers on relatively smaller sizes, due to our limited computational budgets. Another limitation is the main focus on routing problems, even if RL4CO can be easily extended for handling different classes of CO problems, such as scheduling problems. Moreover, we did not benchmark shifts in data distributions for the time being (except for the real-world instances of TSPLib and CVRPLib), which could lead to new insights. In future works, we plan to implement new CO problems that stretch beyond the routing and tackle even larger instances, also owing to the capability of RL4CO library.

## 5.3   Conclusion

This paper introduces RL4CO, a *modular*, *flexible*, and *unified* software library for Reinforcement Learning (RL) for Combinatorial Optimization (CO). Our benchmark library aims at filling the gap in a unified implementation for the NCO area by utilizing several best practices with the goal provide researchers and practitioners with a flexible starting point for NCO research. With RL4CO, we rigorously benchmarked various NCO solvers in the measures of in-distribution, out-of-distribution, sample-efficiency, and search methods performances. Our findings show that a comparison of NCO solvers across different metrics and tasks is fundamental, as state-of-the-art approaches may in fact perform worse than predecessors under these metrics. We hope that our benchmark library will inspire NCO researchers to explore new avenues and drive advancements in this field.

# Acknowledgements

We want to express our gratitude towards all reviewers and people in the community who have contributed - and those who will - to RL4CO. A special thanks goes to the TorchRL team for helping us in solving issues and improving the library. This work was supported by a grant of the KAIST-KT joint research project through AI2XL Laboratory, Institute of Convergence Technology, funded by KT [Project No. G01210696, Development of Multi-Agent Reinforcement Learning Algorithm for Efficient Operation of Complex Distributed Systems].

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
