# RL4CO: an Extensive Reinforcement Learning for Combinatorial Optimization Benchmark
## *Supplementary Material*

## Table of Contents

## A  Additional Discussion about the RL4CO Library

### A.1  Why Choosing RL4CO?

In this paper, we introduce RL4CO, a *modular*, *flexible*, and *unified* implementation of NCO. In designing the library, we intended RL4CO to be used for various purposes ranging from research to production. RL4CO enables the users to have the following benefits.

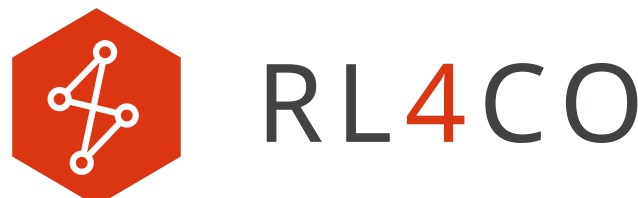

Figure A.1: RL4CO logo.

**Minimal Implementation of Boilerplate Codes for NCO**    As with the other RL projects, implementation of NCO with RL involves designing and coding the systems composed mainly of agents (i.e., policy) and environment (i.e., CO problem). However, this often involves a serious amount of engineering, especially to attain higher executions in training routines. Moreover, we found that a significant chunk of NCO solvers are based on AM or POMO implementation and the subroutines that have been done in the implementation. Regarding those practical aspects, RL4CO provides a modularized code base for each routine of NCO, including environment, policy network architecture, RL algorithm, and training. So that the users can easily mix and match SoTA NCO practices and user-defined modules while having full control over the entire RL pipeline.

**Easier Comparison Among NCO Algorithms**    Current NCO research shows a tendency to rely on two cornerstone implementations: AM and POMO. However, due to differences in implementation (e.g., network architecture, training scheme), direct head-to-head comparisons among the algorithms might not be straightforward. For example, applying POMO's state augmentation to AM's policy to reveal the effect of augmentation from the baseline selections can be challenging. RL4CO provides a unified implementation of NCO models (and their subroutines) to offer higher adaptability of routines from one algorithm to another. We believe this will promote easier comparisons among the models while developing novel NCO solvers to address various CO problems.

**Leveraging Standardized Open-Source Libraries**    During the development of RL4CO, we have decided to utilize standardized and reputable open-source libraries based on extensive research and expertise at the edge of software engineering and research, such as the recent TorchRL [10] and the `TensorDict` data structure from Moens [46]. We believe these design choices will yield various practical benefits both in research and production. For instance, by disentangling the RL algorithm from the training subroutines of RL4CO, NCO solvers can undergo training using an array of state-of-the-art training methods supported by PyTorch Lightning [18]. Additionally, deploying the trained NCO solvers to production becomes seamless through the utilization of tools such as TorchServe, to name just a couple of examples.

### A.2  On the Choice of the Framework

During the development of RL4CO, we wanted to make it as simple as possible to integrate reproducible and standardized code adhering to the latest guidelines. As a main template for our codebase, we use Lightning-Hydra-Template [7] which we acknowledge being a solid starting point for reproducible deep learning. We further discuss framework choices below.

---

[7]https://github.com/ashleve/lightning-hydra-template

**TorchRL and TensorDict**   One of the software hindrances in RL is the bottleneck between CPU and GPU communication majorly due to CPU-based operating environments. For this reason, we did not opt for OpenAI Gym [12] since, although it includes some level of parallelization, this does not happen on GPU and would thus greatly hinder performance. Kool et al. [38] creates *ad-hoc* environments in PyTorch to handle batched data efficiently. However, it could be cumbersome to integrate into standardized routines that include `step` and `reset` functions. As we searched for a better alternative, we found that TorchRL library [47], an official PyTorch project that allows for efficient batched implementations on (multiple) GPUs as well as functions akin to OpenAI Gym. We also employ the TensorDict [47] to handle tensors efficiently on multiple keys (i.e. in CVRP, we can directly operate transforms on multiple keys as locations, capacities, and more). This makes our environments compatible with the models in TorchRL, which we believe could further the interest in the CO area.

**PyTorch Lightning**   PyTorch Lightning [18] is a useful tool for abstracting away the boilerplate code allowing researchers and practitioners to focus more on the core ideas and innovations. With its standardized training loop and extensive set of pre-built components, including automated check-pointing, distributed training, and logging, PyTorch Lightning accelerates development time and facilitates scalability. We employ PyTorch Lightning in RL4CO to integrate with the PyTorch ecosystem - which includes TorchRL- enabling us to leverage the rich set of tools and libraries available.

**Hydra**   Hydra [74] is a powerful open-source framework for managing complex configurations in machine learning models and other software in the form of `yaml` files. Hydra facilitates creating hierarchical configurations, making it easy to manage even very large and intricate configurations. Moreover, it integrates with command-line interfaces, allowing the execution of different configurations directly from the command line, thereby enhancing reproducibility. We found Hydra to be effective when dealing with multiple experiments, since configurations are saved both locally as `yaml` files as well as uploaded on Wandb [8].

### A.3   Open Source Licenses

In this paragraph, we summarize the license of the software that we've employed in developing RL4CO. The license lists are as follows:

- PyTorch, Matplotlib: BSD license
- TorchRL, Einops, Hydra: MIT license
- Lightning: Apache-2.0 license
- Numpy, Scipy: BSD-3-Clause license

For reproducing the previous NCO research, we refer to the original implementation of AM [9], the original implementation of POMO [10], and the original implementation of SymNCO [11]. However, while faithful, our implementation considerably modifies the structure of the original implementation introducing several optimizations to single lines of codes to the overall structure. RL4CO is published under the liberal Apache-2.0 license.

### A.4   Decoder with `DynamicEmbedding`

As discussed in § 3.1, RL4CO provides a modular, flexible, and unified policy network design that is used for various RL algorithms to solve multiple CO problems. In this section, we provide additional discussion about the decoder structure with the dynamic embedding module that

---

[8]https://wandb.ai/
[9]https://github.com/wouterkool/attention-learn-to-route
[10]https://github.com/yd-kwon/POMO
[11]https://github.com/alstn12088/Sym-NCO

is used to solve complex routing problems involving information updates during decoding. The `DynamicEmbedding` module dynamically updates the keys and values of multi-head-attention (MHA) and softmax to reflect the 'dynamic' change of information-related unselected nodes (e.g., unvisited cities in VRPs) during decoding. To be specific, the decoder with `DynamicEmbedding` is defined as follows:

$$q_t = \texttt{ContextEmbedding}(\boldsymbol{h}, a_{t-1:0}), \tag{8}$$

$$\boldsymbol{K}_t^g, \boldsymbol{V}_t^g, \boldsymbol{V}_t = \texttt{DynamicEmbedding}(W_k^g \boldsymbol{h}, W_v^g \boldsymbol{h}, W_v \boldsymbol{h}, a_{t-1:0}, \boldsymbol{h}, \boldsymbol{x}), \tag{9}$$

$$\bar{q}_t = \mathrm{MHA}(q_t, \boldsymbol{K}_t^g, \boldsymbol{V}_t^g), \tag{10}$$

$$\pi(a_t) = \mathrm{MaskedSoftmax}(\bar{q}_t \cdot \boldsymbol{V}_t, M_t), \tag{11}$$

where the dynamic embedding dynamically modulates keys and values of MHA and softmax based on the decoded results (i.e., $a_{t-1:0}$) and inputs (i.e., $\boldsymbol{x}$ and $\boldsymbol{h}$).

## A.5 Related Work

RL4CO aims to provide a unified and modular implementation of NCO solvers along with benchmark results derived from these implementations. In line with RL4CO's goals, two other recent works in the public domain share similar yet different objectives.

In recent work, Wan et al. [71] propose RLOR, which tries to enhance the implementation of AM using a standardized RL framework, specifically CleanRL [27][12]. It's important to note that while RLOR partially succeeds in augmenting the capabilities of AM [38], it is currently limited to solving TSP and CVRP, with only the original AM available. As a result, RLOR's scope deviates slightly from our primary objective of creating a comprehensive and modular suite of NCO solvers, adaptable to a broad range of CO problems. Moreover, by not leveraging parallelizable GPU environments, we tested that the performance is, in fact, worse compared to our implementation. One training run for the AM takes around half the time in RL4CO compared to RLOR with the same hyperparameters.

A concurrent work, Jumanji [9], offers a diverse set of NP-hard environments using Jax [11]. However, its implementation of the policy network and RL algorithms is limited due to the library's scope. While Jumanji is compatible with a wrapper that allows integration with well-known RL frameworks such as Stable Baselines3 [61], adapting it to implement mainstream NCO solvers would require a significant commitment of effort. This arises due to the substantial differences between conventional RL, as discussed in § 2. Its scope also differs from ours in terms of it being primarily a collection of environments rather than being a full-fledged benchmark in itself. Finally, while it could be a matter of personal choice, we believe that PyTorch [55] and libraries based on it are more popular than Jax, including for the NCO area to our knowledge, thus making RL4CO more broadly applicable and easier to approach for most.

## A.6 Quickstart Notebook

We showcase a quickstart Jupyter Notebook for training the Attention Model on TSP with 20 nodes that can be run in less than 2 minutes on Google Colab [8] on a free-tier GPU runtime[13].

```python
import torch
from rl4co.envs import TSPEnv
from rl4co.models.zoo.am import AttentionModel
from rl4co.utils.trainer import RL4COTrainer

# RL4CO env based on TorchRL
```

---

[12]CleanRL also deviates considerably from us in principle. We adopt a *divide-and-conquer* strategy with modularity, while CleanRL tries to include everything in a single file.

[13]Code based on version 0.2.0 of RL4CO, that can be installed in Colab with `!pip install rl4co==0.2.0`. Please refer to the library on GitHub for up-to-date versions.

```
env = TSPEnv(num_loc=20)

# Model: default is AM with REINFORCE and greedy rollout baseline
model = AttentionModel(env,
                       baseline='rollout',
                       train_data_size=100_000,
                       val_data_size=10_000)
trainer = RL4COTrainer(max_epochs=3)

# Greedy rollouts over untrained model
device = torch.device("cuda" if torch.cuda.is_available() else "cpu")
td_init = env.reset(batch_size=[3]).to(device)
model = model.to(device)
out = model(td_init, phase="test", decode_type="greedy", return_actions=True)

# Plotting
print(f"Tour lengths: {[f'{-r.item():.2f}' for r in out['reward']]}")
for td, actions in zip(td_init, out['actions'].cpu()):
    env.render(td, actions)
```

Tour lengths: ['10.38', '7.69', '8.32']

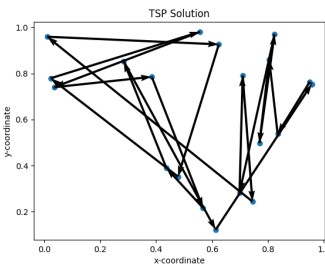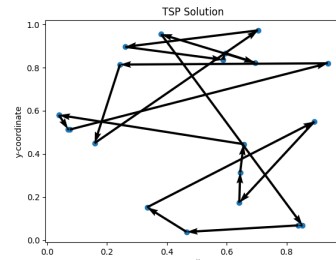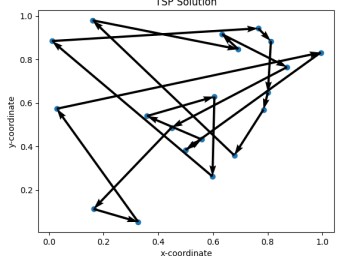

```
# RL4COTrainer with few epochs (wrapper around Lightning Trainer)
trainer = RL4COTrainer(
    max_epochs=3,
    accelerator="gpu",
    logger=None,
)

# Fit the model
trainer.fit(model)
```

```
INFO:
  | Name  | Type           | Params
----------------------------------------
0 | env   | TSPEnv         | 0
1 | model | AttentionModel | 1.4 M
----------------------------------------
1.4 M     Trainable params
```

```
0           Non-trainable params
1.4 M       Total params
5.669       Total estimated model params size (MB)
```

```
Epoch 2: [|||||||||||||||||||||||||||||||||||||||||||||||||||||||] 100% 196/196
[00:44<00:00, 4.36it/s, train/reward=-4.09, train/loss=-.104, val/reward=-4.05]
```

```python
# Greedy rollouts over trained model (same states as previous plot)
model = model.to(device)
out = model(td_init, phase="test", decode_type="greedy", return_actions=True)

# Plotting
print(f"Tour lengths: {[f'{-r.item():.2f}' for r in out['reward']]}")
for td, actions in zip(td_init, out['actions'].cpu()):
    env.render(td, actions)
```

```
Tour lengths: ['3.35', '3.51', '4.22']
```

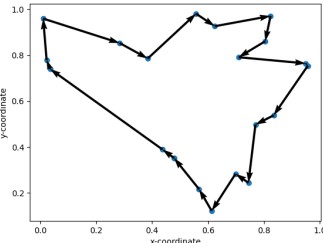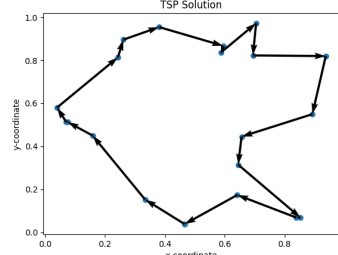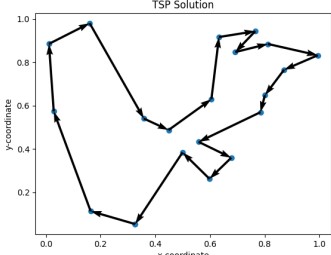

## B Elecronic Design Automation: the Decap Placement Problem

### B.1 Introduction

The optimal placement of a given number of decoupling capacitors (decaps) can significantly impact electrical performance, specifically in terms of power integrity (PI) optimization. PI optimization is crucial in modern chip design, especially with the preference for 3D stacking memory systems like high bandwidth memory (HBM), shown in Fig. B.1. One of the challenges in power supplementation is the vertical transmission of power to 3D memory, which is located at the bottom of memory chips. Consequently, the optimal placement of decaps becomes increasingly important to support the current progress in 3D chip design and high bandwidth, which is critical for meeting the high memory requirements of deep learning technology [28].

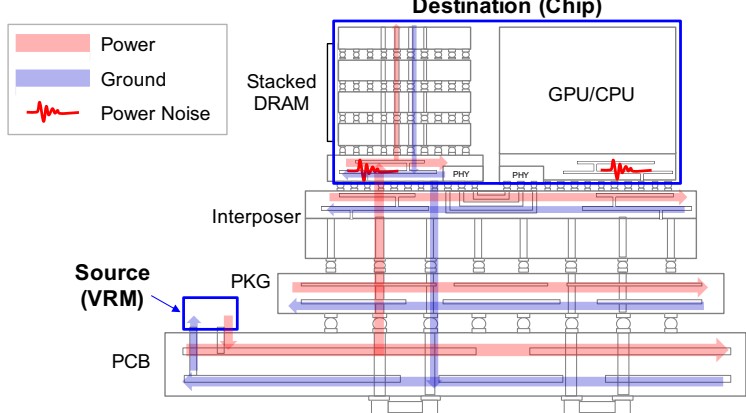

Figure B.1: An example of the power distribution network (PDN) of a high bandwidth memory (HBM) module for a high-performance AI computing system.

The optimal placement of decaps, also known as the decap placement problem (DPP), involves combinatorial optimization with a black box electrical simulator as the scoring function. Typically, this simulator is an expensive electromagnetic (EM) simulator that requires significant computational resources. To address this, we propose a fast approximated simulator that can be executed within a Python environment. While sacrificing some accuracy, our circuit-modeled simulation serves as a proxy score function. Additionally, we limit the number of simulation calls due to the high cost associated with running the real-world EM simulator.

The decap placement problem (DPP) is a highly complex task in hardware design due to two main reasons. Firstly, it involves exploring an extensive range of possibilities in a combinatorial space. Secondly, evaluating the objectives of DPP requires significant computational resources and time, making it necessary to achieve high sample efficiency. Genetic algorithm (GA) based methods have shown promise in addressing DPP because they can reduce the combinatorial space compared to exhaustive search methods [32]. However, GAs still require a large number of iterations (M) for each problem as they are memoryless optimization methods. Reinforcement learning approaches have been attempted to solve DPP, and the most recently proposed DevFormer achieved the state of the art performance while achieving sample efficient in training and zero-shot inference [35].

### B.2 Problem Setting

The decap placement problem (DPP) represents a sample-efficient setting of NCO, in which calculating the reward is expensive due to complex simulation, and hence the number of samples should be limited (i.e. $10,000$ samples for pretraining each model in our experiments). The DPP aims to determine the optimal locations for decaps within a given instance. This instance consists of three key components: (1) the locations of probing ports, (2) the locations of keep-out regions (where

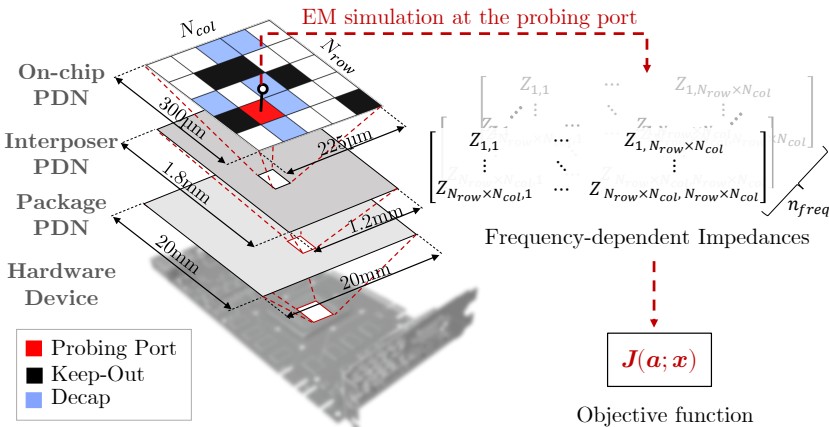

Figure B.2: Grid representation of the target on-chip PDN for the DPP problem with a single probing port. Image from https://github.com/kaist-silab/devformer.

decaps cannot be placed), and (3) the available regions where decaps can be positioned as shown in Fig. B.2. For comprehensive details, we follow the configuration guidelines provided in [35].

**Environments: DPP and mDPP**   In order to compare our approach with the somewhat simplistic solution to DPP presented in [35], which emphasizes placing decaps near probing ports as a good solution, we introduce multi-port DPP (mDPP) and adopt different objectives:

- *Maxsum*: the objective is to maximize the average PI among multiple probing ports

- *Maxmin*: the objective is to maximize the minimum PI among multiple probing ports

These more intricate and practical approaches add complexity to the problem, enabling us to assess the effectiveness of the proposed method in a real-world scenario. Additional details, including data generation, can be found in Appendix C.5 and Appendix C.6.

**Baselines**   We employ two meta-heuristic baselines as outlined in [35]: random search (RS) and genetic algorithm (GA). GA has shown promise as a method for addressing the decap placement problem (DPP). In addition, we introduce DevFormer [35], an AM-variant model specifically designed for DPP. It is important to note that DevFormer is initially designed for offline training; however, in this study, we benchmark DevFormer as a sample-efficient online reinforcement learning approach.

We benchmark the DevFormer version for RL with the same embedding structure as the original in [35]. We benchmark 3 variants, namely DF(PG,Critic): REINFORCE with Critic baseline, DF(PG,Rollout): REINFORCE with Rollout baseline as well as PPO. All experiments are run with the same hyperparameters as the other experiments except for the batch size set to 64, maximum number of samples set to 10,000, and a total of only 10 epochs due to the nature of the benchmark sample efficiency.

## B.3   Benchmark Results

**Main benchmark**   Table B.1 shows the main numerical results for the task when RS, GA and DF models are trained for placing 20 decaps. While RS and GA need to take online shots to solve the problems (we restricted the number to 100), DF models can successfully predict in a zero-shot manner and outperform the classical approaches. Interestingly, the vanilla critic-based method performed the worst, while our implementation of PPO almost matched the rollout policy gradients (PG) baseline; since extensive hyperparameter tuning was not performed, we expect PPO could outperform the rollout baseline given it requires fewer samples. Fig. B.3 shows example renderings of the solved environment.

Table B.1: Performance of different methods on the mDPP benchmark

| Method | # Shots | Score ↑ | |
| --- | --- | --- | --- |
| | | maxsum | maxmin |
| *Online Test Time Search* | | | |
| Random Search | 100 | 11.55 | 10.63 |
| Genetic Algorithm | 100 | 11.93 | 11.07 |
| *RL Pretraining & Zero Shot Inference* | | | |
| DF-(PG,Critic) | 0 | $10.89 \pm 0.63$ | $9.51 \pm 0.68$ |
| DF-(PPO) | 0 | $12.16 \pm 0.03$ | $11.17 \pm 0.11$ |
| DF-(PG,Rollout) | 0 | $12.21 \pm 0.01$ | $11.26 \pm 0.03$ |

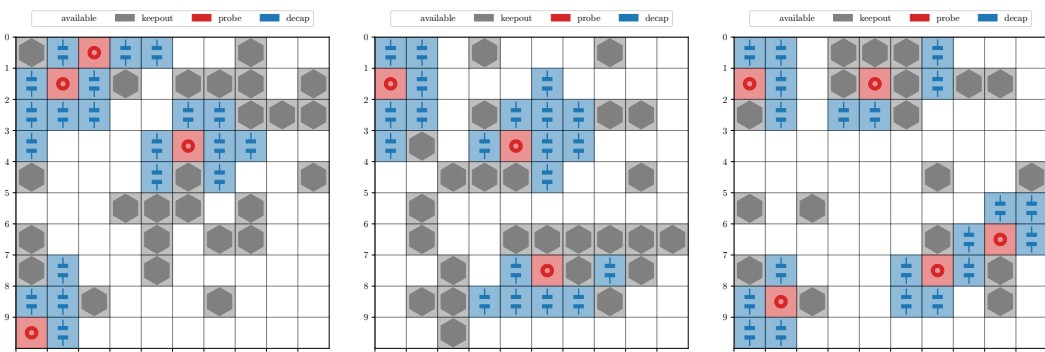

Figure B.3: Renders of the environment with *maxmin* objective solved by DF-(PG,Rollout). The model successfully learned one main heuristic for DPP problems, which is that optimal placement is generally close to probing ports.

**Generalization to different number of decaps**  In hardware design, the number of components is one major contribution to cost; ideally, one would want to use the least number of components possible with the best performance. In the DPP, increasing the number of decaps *generally* improves the performance at a greater cost, hence Pareto-efficient models are essential to identify. Fig. B.4 shows the performance of DF models trained on 20 decaps against the baselines. DF models PPO and PG-rollout can successfully generalize and are also Pareto-efficient with fewer decaps, important in practice for cost and material saving.

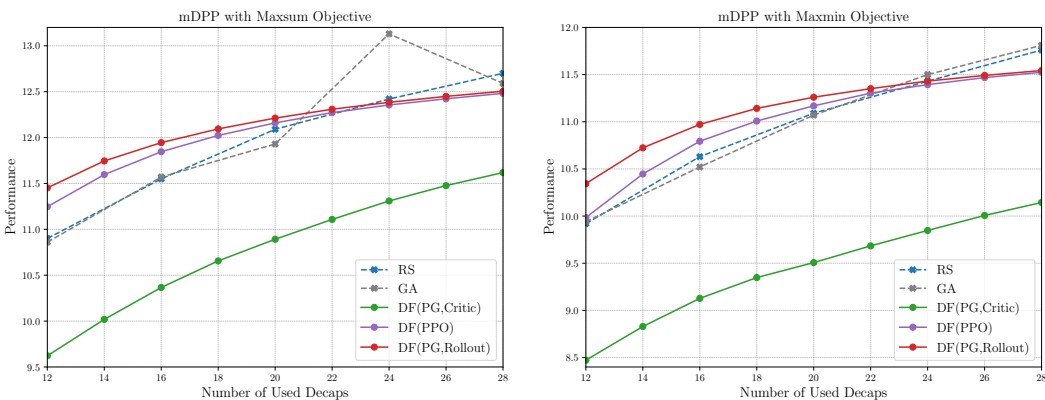

Figure B.4: Performance vs number of used decaps for mDPP with *maxsum* objective [Left] and *maxmin* objective [Right].

## C Environments

In this section, we discuss further details of environment implementation. As Kool et al. [38] environment implementation de-facto standard of implementation, RL4CO also aims to reproduce the same environment settings along with generating the same data.

**Common instance generation details** Following the standard protocol of NCO, we randomly sample node coordinates from the 2D unit square (i.e., $[0, 1]^2$). When generating the training data, we regulate the randomness by setting the random seed to 1234. Conversely, when generating 10,000 validation instances, we use random seed 4321. For the creation of the testing 10,000 instances, we use the random seed 1234. All protocols, including seed selection, align with the practices outlined by Kool et al. [38].

### C.1 Traveling Salesman Problem (TSP)

The Traveling Salesman Problem (TSP) is a fundamental routing problem that aims to find the Hamiltonian cycle of minimum length. While the original TSP formulation employs mixed-integer linear programming (MILP), to integrate TSP into autoregressive solution decoding (i.e., the construction process), we reinterpret the solution-finding process as sequential node selection in line with Kool et al. [38]. In each step of node selection, we preclude the selection of nodes already picked in previous rounds. This procedure ensures the feasibility of constructed solutions and also allows for the potential construction of an optimal solution for any TSP instance.

### C.2 Capacitated Vehicle Routing Problem (CVRP)

The Capacitated Vehicle Routing Problem (CVRP) is a popular extension of TSP, applicable to a variety of real-world logistics/routing problems (e.g., delivery services). In CVRP, each node has its own demand, and each vehicle has a specific capacity. A vehicle can "tour" until the total demand does not exceed its capacity. The vehicle returns to the depot - a unique type of node - and releases all demand, then embarks on another tour from the depot until all nodes have been visited. Similar to TSP, the original CVRP formulation utilizes MILP. However, by applying a similar logic to that of the TSP environment, we can reformulate CVRP as a sequential node selection problem, taking into account demands and capacity.

**Additional generation details** To generate demand, we randomly sample integers between 1 and 10. Without loss of generality, we fix the capacity of the vehicle at 1.0. Instead, we normalize demand by multiplying it by a constant that varies according to the size of the CVRP. The specific constant can be found in our implementation.

### C.3 Orienteering Problem (OP)

The Orienteering Problem (OP) is a variant of TSP. In OP, each node (i.e., city) is assigned a prize. The objective of OP is to find a tour, starting and ending at the depot, that maximizes the total prize collected from visited cities, while abiding by a maximum tour length constraint. Like the previous problems, the original formulation utilizes MILP. However, OP can also be framed as a sequential decision-making problem by enforcing the "return to depot" action when no cities are visitable due to the maximal tour length constraint.

**Additional generation details** To generate the prize, we use the prize distribution proposed in Fischetti et al. [19], particularly the distribution that allocates larger prizes to nodes further from the depot.

### C.4 Prize Collecting TSP (PCTSP)

In the Prize Collecting TSP (PCTSP), each node is assigned both a prize and a penalty. The objective is to accumulate a minimum total prize while minimizing the combined length of the tour and the penalties for unvisited nodes. By making a minor adjustment to PCTSP, it becomes applicable for solving different subproblems that arise in routing problems when using Branch-Price-and-Cut algorithms.

### C.5 Decap Placement Problem (DPP)

The decap placement problem (DPP) is an electronic design automation problem (EDA) in which the goal is to maximize the performance with a limited number of the decoupling capacitor (decap) placements on a hardware board characterized by asymmetric properties, measured via a probing port. The decaps cannot be placed on the location of the probing port or in keep-out regions (which represent other hardware components). The full problem description is provided in Appendix B.

**Instance generation details** We use the same data for simulating the hardware board as Kim et al. [35]. We randomly select one probing port and a number between 1 and 50 keep-out regions sampled from a uniform distribution for generating instances. As in the routing benchmarks, we select seed 1234 for testing the 100 instances.

### C.6 Multi-Port Decap Placement Problem (mDPP)

The multi-port decap placement problem (mDPP) is a generalization of DPP from Appendix C.5 in which measurements from multiple probing ports are performed. The objective function can be either the mean of the reward from the probing ports (*maxsum*) or the minimum between them (*maxmin*).The full problem description is provided in Appendix B.

**Instance generation details** The generation details are the same as DPP, except for the probing port. A number between 2 and 5 probing ports is sampled from a uniform distribution and probing ports are randomly placed on the board as the other components.

### C.7 Additional Environments

We also include in the RL4CO library additional environments on which we did not benchmark models for the time being due to time and resource constraints: the Pickup and Delivery Problem and its multi-agent version (PDP and mPDP), the multiple Traveling Salesman Problem (mTSP), as well as asymmetric CO environments Asymmetric Traveling Salesman Problem (ATSP) and Flexible Flow Shop Problem (FFSP). We include also the Stochastic variant of PCTSP (SPCTSP) and a variation of the CVRP that allows for split deliveries to be considered, namely the Split Delivery Vehicle Routing Problem (SDVRP) - we show an example notebook on the latter under the `notebook/` folder of the library. Given both near and longer-term plans for the library and the RL4CO community, we expect to add several more variations as well as new environments in the future.

## D  Experimental Details

### D.1  Hardware

Experiments were carried out on a machine equipped with two AMD EPYC 7542 32-CORE PROCESSOR CPU with $64$ threads and four NVIDIA RTX A6000 graphic cards with $48$ GB of VRAM.

### D.2  Software

Software-wise, we used `Python 3.10` and the latest PyTorch 2.0 [55] (during development, we used beta wheels as well as manually installed version of FlashAttention [16]), most notably due to the native implementation of `scaled_dot_product_attention`. Given that most models in RL constructive methods for CO generally use attention for encoding states, FlashAttention has some boost on the performance (between $5\%$ and $20\%$ saved time depending on the problem size) when training is subject to mixed-precision training, which we do for all experiments. During decoding, the FlashAttention routine is not called since, at the time of writing, it does not support maskings other than causal; this could further boost performance compared to older implementations. Refer to Appendix A.2 for additional details regarding notable software choices of our library, namely TorchRL, PyTorch Lightning and Hydra.

### D.3  Common Hyperparameters

Common hyperparameters can be found in the `config/` folder from the RL4CO library [14], which can be conveniently loaded by Hydra. We provide yaml-like configuration files below, divided by experiments in Listing 1.

### D.4  Main Tables

We run all models to try and match as much as possible the original implementation details. In particular, we run all models for $250,000$ gradient steps with the same Adam [37] optimizer with a learning rate of $10^{-4}$ and $0$ weight decay. For POMO, we match the original implementation details of weight decay as $10^{-6}$. For POMO, the number of multistarts is the same as the number of possible initial locations in the environment (for instance, for TSP50, $50$ starts are considered). In the case of Sym-NCO, we use $10$ as augmentation for the shared baseline; we match the number of effective samples of AM-XL to the ones of Sym-NCO to demonstrate the differences between models.

---

[14]https://github.com/kaist-silab/rl4co

```
     Common configuration
1    # RL Model configuration (policy+baseline)
2    model:
3        policy:
4            encoder:
5                type: GraphAttentionEncoder
6                num_heads: 8
7                num_layers: 3 # POMO uses 6
8                normalization: "batch" # POMO uses "instance"
9                hidden_dim: 512
10               embedding_dim: 128
11           decoder:
12               num_heads: 8
13               embedding_dim: 128
14               use_graph_context: True # POMO does not use it
15               tanh_clipping: 10.0
16               mask_inner: True
17               mask_logits: True
18               normalize: True
19               softmax_temp: 1.0

20       baseline:
21           "rollout" # default baseline
22
23   # Training configuration
24   train:
25       optimizer:
26           type: Adam
27           learning_rate: 1e-4
28           weight_decay: 0 # POMO uses 1e-6
29       scheduler:
30           type: MultiStepLR # original AM implementation does not use this
31           step_size: [80, 95]
32           gamma: 0.1
33           scheduler_interval: epoch
34       gradient_clip_val: 1.0
35       max_epochs: 100 # we set AM-XL to 500
36       precision: "16-mixed" # allows for FlashAttention
37       strategy: DDPStrategy # efficient for multiple GPUs
38
```

Listing 1: Common configuration for RL model (policy + baseline) and training. Some differences are high-lighted, such as the POMO implementation-level ones, and are provided as comments.

The number of epochs for all models is 100, except for AM-XL (500). We also employ learning rate scheduling, in particular, `MultiStepLR` [15] with $\gamma = 0.1$ on epoch 80 and 95; for AM-XL, this applies on epoch 480 and 495.

## D.5  Sample Efficiency Experiments

We keep the same hyperparameters as Appendix D.4, except for the number of epochs and scheduling. We consider 5 independent runs that match the number of samples *per step* (i.e., the batch size is exactly the same for all models after considering techniques such as the multistart and symmetric baselines). For AM Rollout, we employ half the batch size of other models since it requires double the number of evaluations due to its baseline.

---

[15] https://pytorch.org/docs/stable/generated/torch.optim.lr_scheduler.MultiStepLR

## D.6 Search Methods Experiments

For these experiments, we employ the same models trained in the in-distribution benchmark on 50 nodes. For Active Search (AS), we run 200 iterations for each instance and an augmentation size of 8. The Adam optimizer is used with learning rate of $2.6 \times 10^{-4}$ and weight decay of $10^{-6}$. For Efficient Active Search, we benchmark EAS-Lay (with an added layer during the single-head computation, `LogitAttention` in our code) with the original hyperparameters proposed by Hottung et al. [25]. The learning rate is set to $0.0041$ and weight decay to $10^{-6}$. The search is restricted to 200 iterations with dihedral augmentation of 8 as well as imitation learning weight $\lambda = 0.013$.

Testing is performed on 100 instances on both TSP and CVRP for $N \in [200, 500, 1000]$, generated with the usual random seed for testing 1234.

## D.7 PPO Implementation Details

We also implemented AM-PPO, the Attention Model with Proximal Policy Optimization [64], which was not considered in previous works. We use the same critic network as the critic network for the critic REINFORCE baseline and perform training with the same settings as the AM-Critic based on REINFORCE.

Contrary to the common interpretation that views the solution generation process of CO problems (i.e., Eq. (1)) as a Markov Decision Process (MDP), where each decoding step corresponds to a state transition, we view it as a single-stage problem with an auto-regressive policy construction structure. Following this interpretation, the iterations in the decoding steps do not coincide with MDP's time step updates.

This interpretation requires some modifications to PPO, especially in computing the training labels of the value network (i.e., the value of the value function) to train NCO solvers. Since we have only one stage, our value function predicts the expected cost given the policy from the given problem instance. It is noteworthy that the problem is single-staged; hence, the Generalized Advantage Estimator (GAE) is not applicable, as GAE computes the values in the multi-stage setting. We found that recent fine-tuning of a large language model [51] carried out by PPO also interprets the decoding scheme as a single-state problem and hence does not apply GAE.

As for other hyperparameters, we set the number of epochs to 2, mini-batch size to 512, clip range to 0.2, and entropy coefficient $c_2 = 0.01$. Interestingly, we found that normalizing the advantage as done in the Stable Baselines PPO2 implementation[16] slightly hurt performance, so we set the normalize advantage parameter to `False`. We suspect this is because the NCO solvers are trained on *multiple* problem instances, unlike the other RL applications that aim to learn a policy for a single MDP.

---

[16]https://stable-baselines.readthedocs.io/en/master/modules/ppo2.html

# E  Orienteering Problem and Prize Collecting TSP Experiments

## E.1  Problem Setup

We employ the same setup as the experiments for TSP and CVRP, which can be found at Appendix D. We do note that multi-starts as in POMO had not been implemented before. Thanks to our modular implementation, we allow the first actions to be any node except the depot and can successfully train and test models on these environments.

## E.2  Results

Results can be found in Table E.1. We can see at a glance that the shared POMO baseline struggles to achieve good results; with even the somewhat weak AM-critic outperforming it by far. Moreover, we can see that multi-starts are much less effective compared to the TSP and CVRP problems. This can be attribute to the fact that selecting *any* starting point in POMO for PCTSP and OP is in fact a *suboptimal* strategy since not all nodes may need to be visited. In such case, symmetric baselines can be more effective - unlike in the TSP and CVRP results, in which AM-XL performed similarly to Sym-NCO, in this case Sym-NCO effectively outperforms all neural baselines without even when accounting for larger training sample sizes.

Table E.1: In-domain benchmark results.  † results are reproduced from [38].  For OP, we used Compass as the classical method baseline.  For PCTSP, we used ORTools[57] as the classical method baseline.  The rewards/gaps are measured w.r.t. the best classical heuristic methods.

| Method | OP ($N = 20$) | | | OP ($N = 50$) | | | PCTSP ($N = 20$) | | | PCTSP ($N = 50$) | | |
|---|---|---|---|---|---|---|---|---|---|---|---|---|
| | Reward ↑ | Gap | Time | Reward ↑ | Gap | Time | Cost ↓ | Gap | Time | Cost ↓ | Gap | Time |
| *Gurobi*† | 5.39 | – | 16m | – | – | – | 3.13 | – | 2m | – | – | – |
| *Compass* | 5.37 | 0.00% | 2m | 16.17 | 0.00% | 5m | – | – | – | – | – | – |
| *ORTools* | – | – | – | – | – | – | 3.13 | 0.00% | 5h | 4.48 | 0.00% | 5h |
| *Greedy One Shot Evaluation* | | | | | | | | | | | | |
| AM-critic | 5.01 | 6.70% | (<1s) | 14.77 | 8.64% | (<1s) | 3.36 | 7.35% | (<1s) | 5.15 | 14.96% | (<1s) |
| AM | 5.20 | 3.17% | (<1s) | 15.46 | 4.40% | (<1s) | 3.17 | 1.28% | (<1s) | 4.59 | 2.46% | (<1s) |
| POMO | 4.69 | 12.69% | (<1s) | 13.86 | 14.26% | (<1s) | 3.41 | 8.95% | (<1s) | 5.00 | 11.61% | (<1s) |
| Sym-NCO | 5.30 | 1.37% | (<1s) | 15.67 | 3.09% | (<1s) | 3.15 | 0.64% | (<1s) | 4.52 | 2.12% | (<1s) |
| AM-XL | 5.25 | 2.23% | (<1s) | 15.69 | 2.98% | (<1s) | 3.17 | 1.26% | (<1s) | 4.53 | 2.44% | (<1s) |
| *Sampling with width $M = 1280$* | | | | | | | | | | | | |
| AM-critic | 5.12 | 4.66% | 2m30s | 15.14 | 6.37% | 5m10s | 3.28 | 4.79% | 2m10s | 4.96 | 10.71% | 4m40s |
| AM | 5.30 | 1.30% | 2m30s | 15.90 | 1.68% | 5m10s | 3.15 | 0.78% | 2m10s | 4.52 | 0.99% | 4m40s |
| POMO | 4.90 | 8.83% | 4m50s | 14.62 | 9.56% | 8m30s | 3.33 | 6.39% | 4m20s | 4.82 | 7.59% | 7m50s |
| Sym-NCO | 5.34 | 0.59% | 4m50s | 16.02 | 0.93% | 8m30s | 3.14 | 0.35% | 4m20s | 4.52 | 0.82% | 7m50s |
| AM-XL | 5.32 | 0.93% | 4m50s | 15.97 | 1.25% | 8m30s | 3.15 | 0.56% | 4m20s | 4.52 | 0.88% | 7m50s |
| *Greedy Multistart ($N$)* | | | | | | | | | | | | |
| AM-critic | 5.06 | 5.77% | 2s | 14.61 | 9.65% | 4s | 3.30 | 5.43% | 1s | 5.12 | 14.29% | 2s |
| AM | 5.24 | 2.42% | 2s | 15.71 | 2.84% | 4s | 3.16 | 0.82% | 1s | 4.56 | 1.89% | 2s |
| POMO | 4.76 | 11.32% | 4s | 13.95 | 13.71% | 7s | 3.35 | 7.03% | 3s | 4.98 | 11.16% | 5s |
| Sym-NCO | 5.32 | 0.87% | 4s | 15.88 | 1.79% | 7s | 3.15 | 0.54% | 3s | 4.55 | 1.59% | 5s |
| AM-XL | 5.29 | 1.49% | 4s | 15.85 | 1.95% | 7s | 3.15 | 0.64% | 3s | 4.56 | 1.79% | 5s |
| *Greedy with Augmentation (1280)* | | | | | | | | | | | | |
| AM-critic | 5.04 | 6.10% | 2m35s | 14.89 | 7.91% | 5m20s | 3.33 | 6.39% | 2m10s | 5.15 | 14.96% | 4m40s |
| AM | 5.25 | 2.25% | 2m35s | 15.88 | 1.79% | 5m20s | 3.16 | 0.96% | 2m10s | 4.59 | 2.46% | 4m40s |
| POMO | 4.85 | 9.76% | 5m | 14.23 | 11.97% | 8m45m | 3.37 | 7.55% | 4m20s | 5.09 | 13.61% | 7m50s |
| Sym-NCO | 5.33 | 0.77% | 5m | 15.94 | 1.41% | 8m45m | 3.15 | 0.63% | 4m20s | 4.58 | 2.17% | 7m50s |
| AM-XL | 5.30 | 1.30% | 5m | 15.90 | 1.66% | 8m45m | 3.15 | 0.68% | 4m20s | 4.59 | 2.54% | 7m50s |
| *Greedy Multistart with Augmentation ($N \times 16$)* | | | | | | | | | | | | |
| AM-critic | 5.20 | 3.17% | 32s | 15.22 | 5.88% | 1m30s | 3.28 | 4.95% | 25s | 5.06 | 12.94% | 1m |
| AM | 5.34 | 0.56% | 32s | 16.05 | 0.76% | 1m30s | 3.14 | 0.32% | 25s | 4.54 | 1.28% | 1m |
| POMO | 5.09 | 5.29% | 45s | 15.05 | 6.94% | 2m | 3.35 | 6.95% | 38s | 4.92 | 9.81% | 1m20s |
| Sym-NCO | 5.35 | 0.39% | 45s | 16.09 | 0.51% | 2m | 3.14 | 0.24% | 38s | 4.53 | 1.17% | 1m20s |
| AM-XL | 5.35 | 0.46% | 45s | 16.08 | 0.57% | 2m | 3.14 | 0.28% | 38s | 4.54 | 1.25% | 1m20s |

## F  TSP and CVRP Public Benchmark

In this section, we evaluate the NCO models trained on randomly generated uniform datasets with 20 and 50 nodes against public benchmark datasets. For the Traveling Salesman Problem (TSP), we evaluate the models using instances from TSPLib [62] with fewer than 250 nodes. For the Capacitated Vehicle Routing Problem (CVRP), we evaluate the models using instances from Set A, B, E, F, and M from CVRPLib [29]. The optimal or best-known solutions (BKS) of evaluated TSP and CVRP instances are taken from [62] and [29]. Note that we observed that NCO models with `Augmentation` discovered the solution with a lower cost than the reported BKS for B-n51-k7 of CVRPLib.

**Evaluation results**   Similar to the random instance evaluations, we tested the models with different decoding schemes, including `Greedy`, `Sampling`, `Multistart`, and `Augmentation`. We provide a shortcut link to each combination of results in Table F.1. From the results, we once again confirmed that `Augmentation` generally outperforms the other sampling techniques, with a similar (or smaller) number of sample evaluations and the network forward, similar to the random instance benchmarks.

Table F.1: Table of TSPLib and CVRPLib results

| Dataset | Trained on | Decoding scheme | | | |
|---------|-----------|---------|----------|-----------|--------------|
|         |           | Greedy | Sampling | Multistart | Augmentation |
| TSPLib  | 20        | Table F.2 | Table F.3 | Table F.4 | Table F.5 |
|         | 50        | Table F.6 | Table F.7 | Table F.8 | Table F.9 |
| CVRPLib | 20        | Table F.10 | Table F.11 | Table F.12 | Table F.13 |
|         | 50        | Table F.14 | Table F.15 | Table F.16 | Table F.17 |

**Visualized results**   Here we share the tours (i.e., solutions) of the CVRP instances where NCO models tend to have worse and better performances compared to the optimal (or BKS) solutions. We found that NCO solvers have a tendency that NCO solvers generalize quite well to the variation of size with proper scaling on the coordinates of city positions by inspecting the A sets in CVRP. (See Fig. F.3.) However, they often show drastic performance degradation in the instances that are not generated from uniform (or close to) distribution by inspecting the non-A datasets (e.g., the sets B and F) as shown in Figs. F.4 and F.5

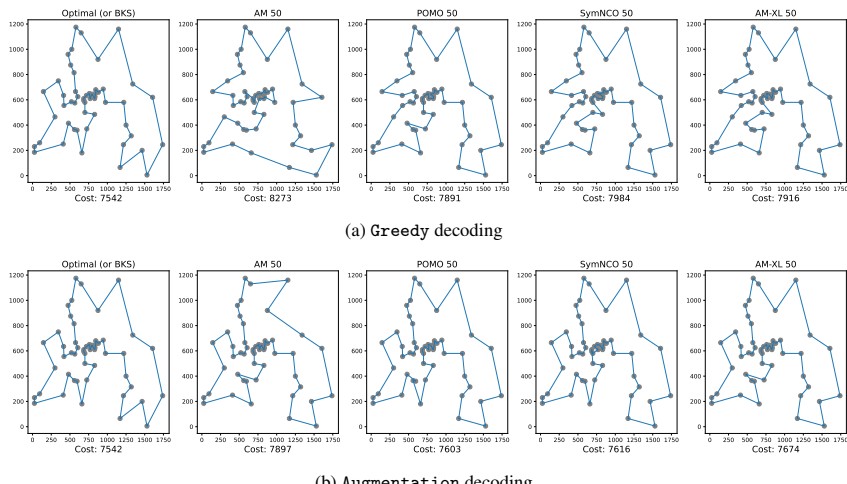

(a) `Greedy` decoding

(b) `Augmentation` decoding

Figure F.1: (TSPLib) Solutions of Berlin52 instance

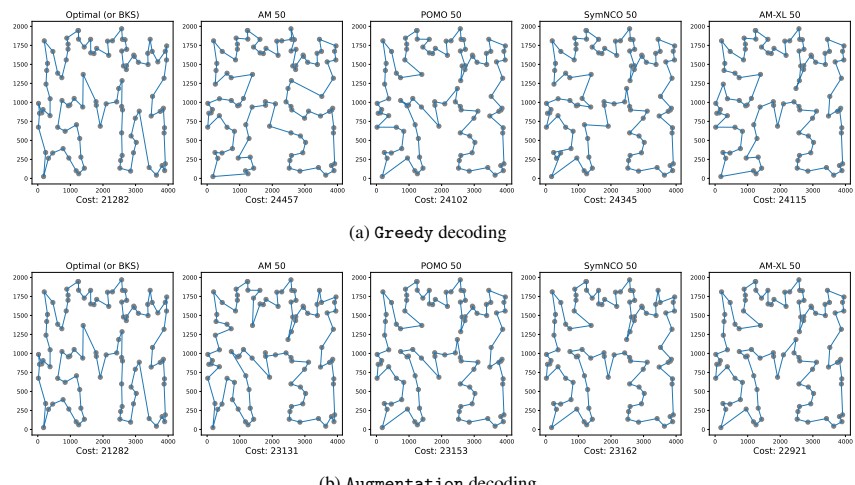

(a) `Greedy` decoding

(b) `Augmentation` decoding

Figure F.2: (TSPLib) Solutions of KroA100 instance

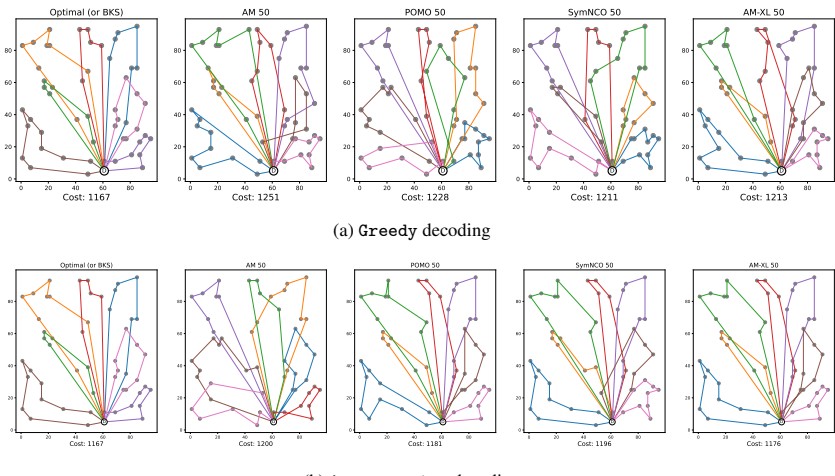

(a) `Greedy` decoding

(b) `Augmentation` decoding

Figure F.3: (CVRPLib) Solutions of A-n54-k7 instance

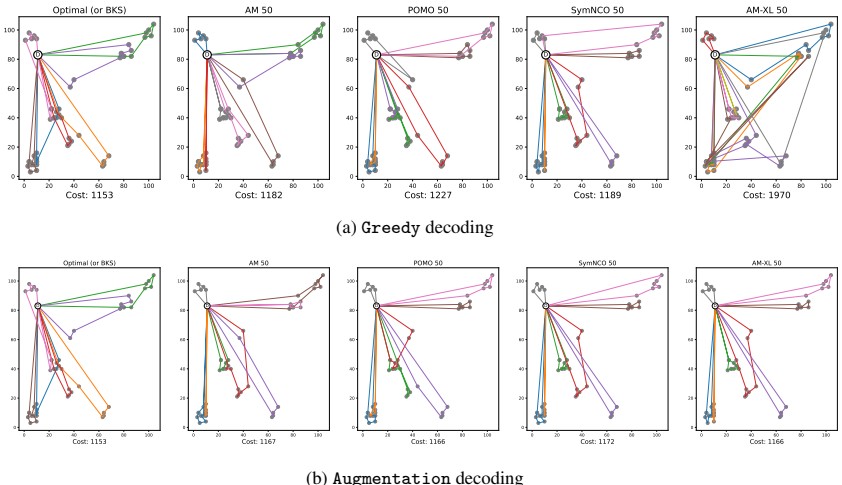

(a) `Greedy` decoding

(b) `Augmentation` decoding

Figure F.4: (CVRPLib) Solutions of B-n57-k7 instance

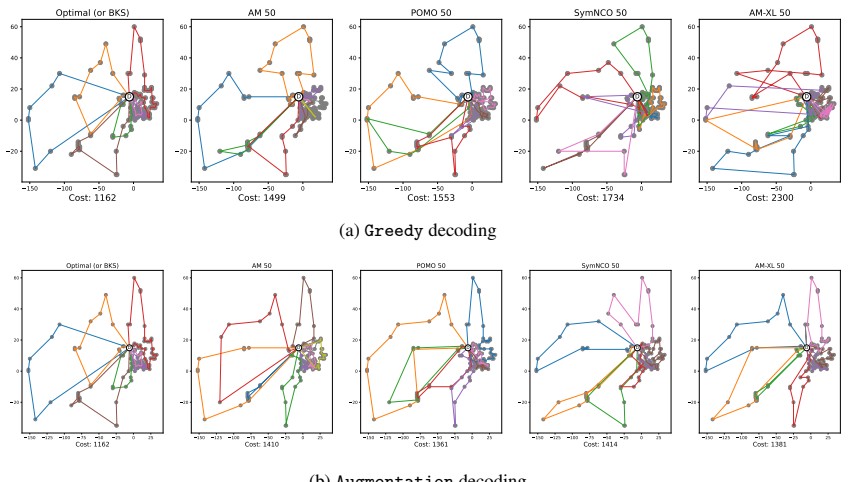

(a) `Greedy` decoding

(b) `Augmentation` decoding

Figure F.5: (CVRPLib) Solutions of F-n135-k7 instance

Table F.2: TSPLib results. The models are trained on TSP20. Greedy decoding is used.

| Instance | Opt. (BKS) | AM | | POMO | | SymNCO | | AM-XL | |
|---|---|---|---|---|---|---|---|---|---|
| | | Cost | Gap ↓ | Cost | Gap ↓ | Cost | Gap ↓ | Cost | Gap ↓ |
| eil51 | 426 | 458 | 6.99 % | 488 | 12.70 % | 462 | 7.79 % | 455 | 6.37 % |
| berlin52 | 7542 | 8623 | 12.54 % | 8757 | 13.87 % | 8518 | 11.46 % | 8621 | 12.52 % |
| st70 | 675 | 718 | 5.99 % | 736 | 8.29 % | 744 | 9.27 % | 733 | 7.91 % |
| eil76 | 538 | 602 | 10.63 % | 606 | 11.22 % | 623 | 13.64 % | 618 | 12.94 % |
| pr76 | 108159 | 116536 | 7.19 % | 123427 | 12.37 % | 119133 | 9.21 % | 115082 | 6.02 % |
| rat99 | 1211 | 1764 | 31.35 % | 1648 | 26.52 % | 1593 | 23.98 % | 1728 | 29.92 % |
| kroA100 | 21282 | 24999 | 14.87 % | 28827 | 26.17 % | 26330 | 19.17 % | 26826 | 20.67 % |
| kroB100 | 22141 | 27325 | 18.97 % | 32980 | 32.87 % | 28256 | 21.64 % | 26209 | 15.52 % |
| kroC100 | 20749 | 24908 | 16.70 % | 25392 | 18.29 % | 26762 | 22.47 % | 26980 | 23.09 % |
| kroD100 | 21294 | 25742 | 17.28 % | 27993 | 23.93 % | 25152 | 15.34 % | 25180 | 15.43 % |
| kroE100 | 22068 | 25985 | 15.07 % | 26754 | 17.52 % | 24751 | 10.84 % | 25494 | 13.44 % |
| rd100 | 7910 | 9324 | 15.17 % | 9011 | 12.22 % | 9121 | 13.28 % | 9096 | 13.04 % |
| eil101 | 629 | 752 | 16.36 % | 769 | 18.21 % | 733 | 14.19 % | 732 | 14.07 % |
| lin105 | 14379 | 17540 | 18.02 % | 17681 | 18.68 % | 17448 | 17.59 % | 17209 | 16.44 % |
| pr124 | 59030 | 63859 | 7.56 % | 68615 | 13.97 % | 65788 | 10.27 % | 68607 | 13.96 % |
| bier127 | 118282 | 141651 | 16.50 % | 167812 | 29.52 % | 145686 | 18.81 % | 139922 | 15.47 % |
| ch130 | 6110 | 6993 | 12.63 % | 8114 | 24.70 % | 7122 | 14.21 % | 7367 | 17.06 % |
| pr136 | 96772 | 116722 | 17.09 % | 117786 | 17.84 % | 114860 | 15.75 % | 112008 | 13.60 % |
| pr144 | 58537 | 64888 | 9.79 % | 65724 | 10.94 % | 66244 | 11.63 % | 66012 | 11.32 % |
| kroA150 | 26524 | 34041 | 22.08 % | 39355 | 32.60 % | 34499 | 23.12 % | 35034 | 24.29 % |
| kroB150 | 26130 | 34394 | 24.03 % | 40379 | 35.29 % | 35975 | 27.37 % | 34700 | 24.70 % |
| pr152 | 73682 | 85034 | 13.35 % | 103086 | 28.52 % | 85237 | 13.56 % | 87594 | 15.88 % |
| u159 | 42080 | 52067 | 19.18 % | 67675 | 37.82 % | 53603 | 21.50 % | 52161 | 19.33 % |
| rat195 | 2323 | 3761 | 38.23 % | 3492 | 33.48 % | 3431 | 32.29 % | 3533 | 34.25 % |
| kroA200 | 29368 | 38338 | 23.40 % | 45189 | 35.01 % | 41007 | 28.38 % | 38696 | 24.11 % |
| ts225 | 126643 | 170904 | 25.90 % | 175644 | 27.90 % | 165864 | 23.65 % | 165968 | 23.69 % |
| tsp225 | 3919 | 5514 | 28.93 % | 5490 | 28.62 % | 5251 | 25.37 % | 5279 | 25.76 % |
| pr226 | 80369 | 96413 | 16.64 % | 117181 | 31.41 % | 94562 | 15.01 % | 88617 | 9.31 % |
| Avg. Gap | 0.00 % | 17.23% | | 22.87% | | 17.53% | | 17.15% | |

Table F.3: TSPLib results. The models are trained on TSP20. Sampling decoding (100 with temperature $\tau = 0.05$) is used.

| Instance | Opt. (BKS) | AM | | POMO | | SymNCO | | AM-XL | |
|---|---|---|---|---|---|---|---|---|---|
| | | Cost | Gap ↓ | Cost | Gap ↓ | Cost | Gap ↓ | Cost | Gap ↓ |
| eil51 | 426 | 452 | 5.75 % | 455 | 6.37 % | 453 | 5.96 % | 452 | 5.75 % |
| berlin52 | 7542 | 8623 | 12.54 % | 8659 | 12.90 % | 8503 | 11.30 % | 8326 | 9.42 % |
| st70 | 675 | 714 | 5.46 % | 715 | 5.59 % | 722 | 6.51 % | 733 | 7.91 % |
| eil76 | 538 | 589 | 8.66 % | 593 | 9.27 % | 594 | 9.43 % | 580 | 7.24 % |
| pr76 | 108159 | 113356 | 4.58 % | 121733 | 11.15 % | 113680 | 4.86 % | 114846 | 5.82 % |
| rat99 | 1211 | 1683 | 28.05 % | 1556 | 22.17 % | 1545 | 21.62 % | 1592 | 23.93 % |
| kroA100 | 21282 | 24614 | 13.54 % | 27074 | 21.39 % | 26549 | 19.84 % | 25947 | 17.98 % |
| kroB100 | 22141 | 25083 | 11.73 % | 28736 | 22.95 % | 26982 | 17.94 % | 24498 | 9.62 % |
| kroC100 | 20749 | 24330 | 14.72 % | 24666 | 15.88 % | 25006 | 17.02 % | 24432 | 15.07 % |
| kroD100 | 21294 | 24307 | 12.40 % | 25464 | 16.38 % | 24590 | 13.40 % | 24602 | 13.45 % |
| kroE100 | 22068 | 25292 | 12.75 % | 26232 | 15.87 % | 24879 | 11.30 % | 24610 | 10.33 % |
| rd100 | 7910 | 9028 | 12.38 % | 8604 | 8.07 % | 8990 | 12.01 % | 8754 | 9.64 % |
| eil101 | 629 | 706 | 10.91 % | 732 | 14.07 % | 715 | 12.03 % | 710 | 11.41 % |
| lin105 | 14379 | 17194 | 16.37 % | 16953 | 15.18 % | 16629 | 13.53 % | 17081 | 15.82 % |
| pr124 | 59030 | 66264 | 10.92 % | 71454 | 17.39 % | 68208 | 13.46 % | 69766 | 15.39 % |
| bier127 | 118282 | 135364 | 12.62 % | 146716 | 19.38 % | 144157 | 17.95 % | 140514 | 15.82 % |
| ch130 | 6110 | 6897 | 11.41 % | 7225 | 15.43 % | 6940 | 11.96 % | 6945 | 12.02 % |
| pr136 | 96772 | 111847 | 13.48 % | 114434 | 15.43 % | 112828 | 14.23 % | 111362 | 13.10 % |
| pr144 | 58537 | 67643 | 13.46 % | 68830 | 14.95 % | 72190 | 18.91 % | 70068 | 16.46 % |
| kroA150 | 26524 | 33358 | 20.49 % | 34358 | 22.80 % | 34281 | 22.63 % | 33881 | 21.71 % |
| kroB150 | 26130 | 31668 | 17.49 % | 35325 | 26.03 % | 34494 | 24.25 % | 33365 | 21.68 % |
| pr152 | 73682 | 86191 | 14.51 % | 89573 | 17.74 % | 89469 | 17.65 % | 88763 | 16.99 % |
| u159 | 42080 | 52987 | 20.58 % | 58999 | 28.68 % | 54263 | 22.45 % | 52393 | 19.68 % |
| rat195 | 2323 | 3558 | 34.71 % | 3472 | 33.09 % | 3329 | 30.22 % | 3557 | 34.69 % |
| kroA200 | 29368 | 37013 | 20.65 % | 43916 | 33.13 % | 41634 | 29.46 % | 39579 | 25.80 % |
| ts225 | 126643 | 175030 | 27.64 % | 179713 | 29.53 % | 182349 | 30.55 % | 176018 | 28.05 % |
| tsp225 | 3919 | 5390 | 27.29 % | 5686 | 31.08 % | 5510 | 28.87 % | 5389 | 27.28 % |
| pr226 | 80369 | 100274 | 19.85 % | 108547 | 25.96 % | 106235 | 24.35 % | 105694 | 23.96 % |
| Avg. Gap | 0.00 % | 15.53% | | 18.85% | | 17.27% | | 16.29% | |

Table F.4: TSPLib results. The models are trained on TSP20. Greedy multi-start decoding is used.

| Instance | Opt. (BKS) | AM | | POMO | | SymNCO | | AM-XL | |
|---|---|---|---|---|---|---|---|---|---|
| | | Cost | Gap ↓ | Cost | Gap ↓ | Cost | Gap ↓ | Cost | Gap ↓ |
| eil51 | 426 | 458 | 6.99 % | 449 | 5.12 % | 453 | 5.96 % | 454 | 6.17 % |
| berlin52 | 7542 | 8623 | 12.54 % | 8075 | 6.60 % | 8251 | 8.59 % | 8419 | 10.42 % |
| st70 | 675 | 713 | 5.33 % | 711 | 5.06 % | 735 | 8.16 % | 716 | 5.73 % |
| eil76 | 538 | 602 | 10.63 % | 606 | 11.22 % | 601 | 10.48 % | 600 | 10.33 % |
| pr76 | 108159 | 113356 | 4.58 % | 113227 | 4.48 % | 118432 | 8.67 % | 113647 | 4.83 % |
| rat99 | 1211 | 1680 | 27.92 % | 1556 | 22.17 % | 1559 | 22.32 % | 1647 | 26.47 % |
| kroA100 | 21282 | 23771 | 10.47 % | 26118 | 18.52 % | 25840 | 17.64 % | 26826 | 20.67 % |
| kroB100 | 22141 | 25721 | 13.92 % | 27440 | 19.31 % | 26081 | 15.11 % | 25275 | 12.40 % |
| kroC100 | 20749 | 24377 | 14.88 % | 23883 | 13.12 % | 25633 | 19.05 % | 24224 | 14.35 % |
| kroD100 | 21294 | 25429 | 16.26 % | 24914 | 14.53 % | 23878 | 10.82 % | 25047 | 14.98 % |
| kroE100 | 22068 | 25354 | 12.96 % | 25843 | 14.61 % | 24751 | 10.84 % | 24857 | 11.22 % |
| rd100 | 7910 | 9035 | 12.45 % | 8708 | 9.16 % | 9088 | 12.96 % | 9062 | 12.71 % |
| eil101 | 629 | 740 | 15.00 % | 731 | 13.95 % | 720 | 12.64 % | 726 | 13.36 % |
| lin105 | 14379 | 17466 | 17.67 % | 17149 | 16.15 % | 16968 | 15.26 % | 16616 | 13.46 % |
| pr124 | 59030 | 63859 | 7.56 % | 64806 | 8.91 % | 64998 | 9.18 % | 67139 | 12.08 % |
| bier127 | 118282 | 140462 | 15.79 % | 141116 | 16.18 % | 142547 | 17.02 % | 139922 | 15.47 % |
| ch130 | 6110 | 6993 | 12.63 % | 6936 | 11.91 % | 6960 | 12.21 % | 6857 | 10.89 % |
| pr136 | 96772 | 115122 | 15.94 % | 112504 | 13.98 % | 113532 | 14.76 % | 110596 | 12.50 % |
| pr144 | 58537 | 63789 | 8.23 % | 65724 | 10.94 % | 66029 | 11.35 % | 66012 | 11.32 % |
| kroA150 | 26524 | 33285 | 20.31 % | 33897 | 21.75 % | 33870 | 21.69 % | 34631 | 23.41 % |
| kroB150 | 26130 | 32483 | 19.56 % | 33308 | 21.55 % | 32578 | 19.79 % | 33184 | 21.26 % |
| pr152 | 73682 | 84881 | 13.19 % | 84819 | 13.13 % | 85237 | 13.56 % | 86964 | 15.27 % |
| u159 | 42080 | 50776 | 17.13 % | 54142 | 22.28 % | 52220 | 19.42 % | 51686 | 18.59 % |
| rat195 | 2323 | 3568 | 34.89 % | 3258 | 28.70 % | 3341 | 30.47 % | 3533 | 34.25 % |
| kroA200 | 29368 | 37345 | 21.36 % | 38861 | 24.43 % | 38629 | 23.97 % | 37628 | 21.95 % |
| ts225 | 126643 | 167811 | 24.53 % | 159519 | 20.61 % | 155767 | 18.70 % | 156938 | 19.30 % |
| tsp225 | 3919 | 5389 | 27.28 % | 5392 | 27.32 % | 5134 | 23.67 % | 5184 | 24.40 % |
| pr226 | 80369 | 91949 | 12.59 % | 92146 | 12.78 % | 92011 | 12.65 % | 88476 | 9.16 % |
| Avg. Gap | 0.00 % | | 15.45% | | 15.30% | | 15.25% | | 15.25% |

Table F.5: TSPLib results. The models are trained on TSP20. Augmentation decoding (100) is used.

| Instance | Opt. (BKS) | AM | | POMO | | SymNCO | | AM-XL | |
|---|---|---|---|---|---|---|---|---|---|
| | | Cost | Gap ↓ | Cost | Gap ↓ | Cost | Gap ↓ | Cost | Gap ↓ |
| eil51 | 426 | 445 | 4.27 % | 443 | 3.84 % | 438 | 2.74 % | 441 | 3.40 % |
| berlin52 | 7542 | 7874 | 4.22 % | 8253 | 8.62 % | 7805 | 3.37 % | 7914 | 4.70 % |
| st70 | 675 | 696 | 3.02 % | 701 | 3.71 % | 704 | 4.12 % | 696 | 3.02 % |
| eil76 | 538 | 582 | 7.56 % | 598 | 10.03 % | 574 | 6.27 % | 571 | 5.78 % |
| pr76 | 108159 | 110726 | 2.32 % | 111534 | 3.03 % | 111035 | 2.59 % | 111416 | 2.92 % |
| rat99 | 1211 | 1426 | 15.08 % | 1526 | 20.64 % | 1533 | 21.00 % | 1493 | 18.89 % |
| kroA100 | 21282 | 24327 | 12.52 % | 26154 | 18.63 % | 23784 | 10.52 % | 24798 | 14.18 % |
| kroB100 | 22141 | 25582 | 13.45 % | 26913 | 17.73 % | 25068 | 11.68 % | 24378 | 9.18 % |
| kroC100 | 20749 | 23991 | 13.51 % | 22781 | 8.92 % | 23780 | 12.75 % | 23712 | 12.50 % |
| kroD100 | 21294 | 24289 | 12.33 % | 25099 | 15.16 % | 23815 | 10.59 % | 24052 | 11.47 % |
| kroE100 | 22068 | 24889 | 11.33 % | 25946 | 14.95 % | 24373 | 9.46 % | 24728 | 10.76 % |
| rd100 | 7910 | 8738 | 9.48 % | 8840 | 10.52 % | 8674 | 8.81 % | 8419 | 6.05 % |
| eil101 | 629 | 701 | 10.27 % | 711 | 11.53 % | 706 | 10.91 % | 696 | 9.63 % |
| lin105 | 14379 | 16670 | 13.74 % | 16956 | 15.20 % | 16297 | 11.77 % | 16656 | 13.67 % |
| pr124 | 59030 | 63859 | 7.56 % | 66313 | 10.98 % | 62469 | 5.51 % | 62260 | 5.19 % |
| bier127 | 118282 | 134016 | 11.74 % | 149337 | 20.80 % | 135108 | 12.45 % | 136150 | 13.12 % |
| ch130 | 6110 | 6816 | 10.36 % | 6995 | 12.65 % | 6732 | 9.24 % | 6791 | 10.03 % |
| pr136 | 96772 | 113441 | 14.69 % | 112384 | 13.89 % | 110893 | 12.73 % | 109425 | 11.56 % |
| pr144 | 58537 | 63032 | 7.13 % | 63877 | 8.36 % | 62506 | 6.35 % | 63297 | 7.52 % |
| kroA150 | 26524 | 32533 | 18.47 % | 33432 | 20.66 % | 32734 | 18.97 % | 31569 | 15.98 % |
| kroB150 | 26130 | 31116 | 16.02 % | 32583 | 19.80 % | 31105 | 15.99 % | 31246 | 16.37 % |
| pr152 | 73682 | 81811 | 9.94 % | 81998 | 10.14 % | 81797 | 9.92 % | 81166 | 9.22 % |
| u159 | 42080 | 51050 | 17.57 % | 52328 | 19.58 % | 50831 | 17.22 % | 50282 | 16.31 % |
| rat195 | 2323 | 3231 | 28.10 % | 3273 | 29.03 % | 3238 | 28.26 % | 3250 | 28.52 % |
| kroA200 | 29368 | 36065 | 18.57 % | 38996 | 24.69 % | 36201 | 18.88 % | 37249 | 21.16 % |
| ts225 | 126643 | 160088 | 20.89 % | 161870 | 21.76 % | 152170 | 16.78 % | 153857 | 17.69 % |
| tsp225 | 3919 | 5273 | 25.68 % | 5247 | 25.31 % | 5169 | 24.18 % | 5150 | 23.90 % |
| pr226 | 80369 | 87684 | 8.34 % | 90981 | 11.66 % | 87330 | 7.97 % | 85462 | 5.96 % |
| Avg. Gap | 0.00 % | | 12.43% | | 14.71% | | 11.82% | | 11.74% |

Table F.6: TSPLib results. The models are trained on TSP50. Greedy decoding is used.

| Instance | Opt. (BKS) | AM | | POMO | | SymNCO | | AM-XL | |
|---|---|---|---|---|---|---|---|---|---|
| | | Cost | Gap ↓ | Cost | Gap ↓ | Cost | Gap ↓ | Cost | Gap ↓ |
| eil51 | 426 | 440 | 3.18 % | 436 | 2.29 % | 434 | 1.84 % | 436 | 2.29 % |
| berlin52 | 7542 | 8273 | 8.84 % | 7891 | 4.42 % | 7984 | 5.54 % | 7916 | 4.72 % |
| st70 | 675 | 694 | 2.74 % | 700 | 3.57 % | 696 | 3.02 % | 686 | 1.60 % |
| eil76 | 538 | 580 | 7.24 % | 562 | 4.27 % | 572 | 5.94 % | 553 | 2.71 % |
| pr76 | 108159 | 110798 | 2.38 % | 113731 | 4.90 % | 111953 | 3.39 % | 111360 | 2.87 % |
| rat99 | 1211 | 1482 | 18.29 % | 1503 | 19.43 % | 1442 | 16.02 % | 1456 | 16.83 % |
| kroA100 | 21282 | 24457 | 12.98 % | 24102 | 11.70 % | 24345 | 12.58 % | 24115 | 11.75 % |
| kroB100 | 22141 | 26447 | 16.28 % | 24086 | 8.08 % | 25146 | 11.95 % | 24607 | 10.02 % |
| kroC100 | 20749 | 24211 | 14.30 % | 23334 | 11.08 % | 22725 | 8.70 % | 23362 | 11.18 % |
| kroD100 | 21294 | 23117 | 7.89 % | 24180 | 11.94 % | 23326 | 8.71 % | 23751 | 10.34 % |
| kroE100 | 22068 | 24476 | 9.84 % | 28393 | 22.28 % | 23933 | 7.79 % | 24865 | 11.25 % |
| rd100 | 7910 | 8163 | 3.10 % | 8139 | 2.81 % | 8072 | 2.01 % | 8082 | 2.13 % |
| eil101 | 629 | 678 | 7.23 % | 681 | 7.64 % | 691 | 8.97 % | 679 | 7.36 % |
| lin105 | 14379 | 15590 | 7.77 % | 16418 | 12.42 % | 16545 | 13.09 % | 16029 | 10.29 % |
| pr124 | 59030 | 61155 | 3.47 % | 61062 | 3.33 % | 59809 | 1.30 % | 59332 | 0.51 % |
| bier127 | 118282 | 130236 | 9.18 % | 154102 | 23.24 % | 137533 | 14.00 % | 133332 | 11.29 % |
| ch130 | 6110 | 6440 | 5.12 % | 6441 | 5.14 % | 6308 | 3.14 % | 6320 | 3.32 % |
| pr136 | 96772 | 104110 | 7.05 % | 104135 | 7.07 % | 103969 | 6.92 % | 102428 | 5.52 % |
| pr144 | 58537 | 63959 | 8.48 % | 63372 | 7.63 % | 60421 | 3.12 % | 61613 | 4.99 % |
| kroA150 | 26524 | 30287 | 12.42 % | 30933 | 14.25 % | 30614 | 13.36 % | 30685 | 13.56 % |
| kroB150 | 26130 | 30565 | 14.51 % | 30997 | 15.70 % | 29083 | 10.15 % | 29528 | 11.51 % |
| pr152 | 73682 | 84632 | 12.94 % | 79409 | 7.21 % | 78854 | 6.56 % | 78362 | 5.97 % |
| u159 | 42080 | 46792 | 10.07 % | 46249 | 9.01 % | 46818 | 10.12 % | 44975 | 6.44 % |
| rat195 | 2323 | 3182 | 27.00 % | 3357 | 30.80 % | 3149 | 26.23 % | 3403 | 31.74 % |
| kroA200 | 29368 | 35068 | 16.25 % | 37667 | 22.03 % | 35349 | 16.92 % | 34728 | 15.43 % |
| ts225 | 126643 | 147273 | 14.01 % | 145364 | 12.88 % | 139645 | 9.31 % | 138527 | 8.58 % |
| tsp225 | 3919 | 4836 | 18.96 % | 5196 | 24.58 % | 4813 | 18.57 % | 5103 | 23.20 % |
| pr226 | 80369 | 86665 | 7.26 % | 87200 | 7.83 % | 85375 | 5.86 % | 86397 | 6.98 % |
| Avg. Gap | 0.00 % | | 10.31% | | 11.34% | | 9.11% | | 9.09% |

Table F.7: TSPLib results. The models are trained on TSP50. Sampling decoding (100 with temperature $\tau = 0.05$) is used.

| Instance | Opt. (BKS) | AM | | POMO | | SymNCO | | AM-XL | |
|---|---|---|---|---|---|---|---|---|---|
| | | Cost | Gap ↓ | Cost | Gap ↓ | Cost | Gap ↓ | Cost | Gap ↓ |
| eil51 | 426 | 440 | 3.18 % | 436 | 2.29 % | 433 | 1.62 % | 436 | 2.29 % |
| berlin52 | 7542 | 8258 | 8.67 % | 7891 | 4.42 % | 7984 | 5.54 % | 7916 | 4.72 % |
| st70 | 675 | 694 | 2.74 % | 699 | 3.43 % | 695 | 2.88 % | 686 | 1.60 % |
| eil76 | 538 | 572 | 5.94 % | 562 | 4.27 % | 570 | 5.61 % | 553 | 2.71 % |
| pr76 | 108159 | 110798 | 2.38 % | 113034 | 4.31 % | 111953 | 3.39 % | 111360 | 2.87 % |
| rat99 | 1211 | 1476 | 17.95 % | 1461 | 17.11 % | 1442 | 16.02 % | 1423 | 14.90 % |
| kroA100 | 21282 | 23863 | 10.82 % | 23674 | 10.10 % | 24230 | 12.17 % | 24115 | 11.75 % |
| kroB100 | 22141 | 24852 | 10.91 % | 24086 | 8.08 % | 24816 | 10.78 % | 24472 | 9.53 % |
| kroC100 | 20749 | 23350 | 11.14 % | 23250 | 10.76 % | 22725 | 8.70 % | 23358 | 11.17 % |
| kroD100 | 21294 | 23117 | 7.89 % | 24103 | 11.65 % | 23326 | 8.71 % | 23720 | 10.23 % |
| kroE100 | 22068 | 24464 | 9.79 % | 24321 | 9.26 % | 23718 | 6.96 % | 24561 | 10.15 % |
| rd100 | 7910 | 8092 | 2.25 % | 8070 | 1.98 % | 8068 | 1.96 % | 8014 | 1.30 % |
| eil101 | 629 | 677 | 7.09 % | 677 | 7.09 % | 684 | 8.04 % | 679 | 7.36 % |
| lin105 | 14379 | 15559 | 7.58 % | 16369 | 12.16 % | 16073 | 10.54 % | 15520 | 7.35 % |
| pr124 | 59030 | 61017 | 3.26 % | 60711 | 2.77 % | 59809 | 1.30 % | 59332 | 0.51 % |
| bier127 | 118282 | 127519 | 7.24 % | 145576 | 18.75 % | 136891 | 13.59 % | 127554 | 7.27 % |
| ch130 | 6110 | 6354 | 3.84 % | 6399 | 4.52 % | 6291 | 2.88 % | 6308 | 3.14 % |
| pr136 | 96772 | 103066 | 6.11 % | 103024 | 6.07 % | 103054 | 6.10 % | 101760 | 4.90 % |
| pr144 | 58537 | 60827 | 3.76 % | 61126 | 4.24 % | 60040 | 2.50 % | 60694 | 3.55 % |
| kroA150 | 26524 | 30015 | 11.63 % | 30664 | 13.50 % | 30510 | 13.06 % | 30355 | 12.62 % |
| kroB150 | 26130 | 29521 | 11.49 % | 29313 | 10.86 % | 28883 | 9.53 % | 29029 | 9.99 % |
| pr152 | 73682 | 80769 | 8.77 % | 78405 | 6.02 % | 77298 | 4.68 % | 77052 | 4.37 % |
| u159 | 42080 | 45508 | 7.53 % | 45598 | 7.72 % | 46084 | 8.69 % | 44547 | 5.54 % |
| rat195 | 2323 | 3051 | 23.86 % | 3120 | 25.54 % | 3060 | 24.08 % | 3098 | 25.02 % |
| kroA200 | 29368 | 34515 | 14.91 % | 35815 | 18.00 % | 33866 | 13.28 % | 34432 | 14.71 % |
| ts225 | 126643 | 141706 | 10.63 % | 142392 | 11.06 % | 139255 | 9.06 % | 138130 | 8.32 % |
| tsp225 | 3919 | 4726 | 17.08 % | 4935 | 20.59 % | 4560 | 14.06 % | 4644 | 15.61 % |
| pr226 | 80369 | 85410 | 5.90 % | 85033 | 5.48 % | 85232 | 5.71 % | 85741 | 6.27 % |
| Avg. Gap | 0.00 % | | 8.73% | | 9.36% | | 8.27% | | 7.85% |

Table F.8: TSPLib results. The models are trained on TSP50. Greedy multi-start decoding is used.

| Instance | Opt. (BKS) | AM | | POMO | | SymNCO | | AM-XL | |
|---|---|---|---|---|---|---|---|---|---|
| | | Cost | Gap ↓ | Cost | Gap ↓ | Cost | Gap ↓ | Cost | Gap ↓ |
| eil51 | 426 | 440 | 3.18 % | 431 | 1.16 % | 430 | 0.93 % | 436 | 2.29 % |
| berlin52 | 7542 | 8270 | 8.80 % | 7679 | 1.78 % | 7984 | 5.54 % | 7731 | 2.44 % |
| st70 | 675 | 693 | 2.60 % | 693 | 2.60 % | 696 | 3.02 % | 686 | 1.60 % |
| eil76 | 538 | 580 | 7.24 % | 559 | 3.76 % | 563 | 4.44 % | 553 | 2.71 % |
| pr76 | 108159 | 110601 | 2.21 % | 111732 | 3.20 % | 111953 | 3.39 % | 111360 | 2.87 % |
| rat99 | 1211 | 1475 | 17.90 % | 1422 | 14.84 % | 1442 | 16.02 % | 1417 | 14.54 % |
| kroA100 | 21282 | 23993 | 11.30 % | 23241 | 8.43 % | 23817 | 10.64 % | 24102 | 11.70 % |
| kroB100 | 22141 | 25097 | 11.78 % | 24071 | 8.02 % | 24026 | 7.85 % | 24607 | 10.02 % |
| kroC100 | 20749 | 23354 | 11.15 % | 22539 | 7.94 % | 22725 | 8.70 % | 22943 | 9.56 % |
| kroD100 | 21294 | 23117 | 7.89 % | 23025 | 7.52 % | 22731 | 6.32 % | 23136 | 7.96 % |
| kroE100 | 22068 | 24476 | 9.84 % | 23746 | 7.07 % | 23908 | 7.70 % | 23834 | 7.41 % |
| rd100 | 7910 | 8159 | 3.05 % | 8047 | 1.70 % | 8072 | 2.01 % | 8021 | 1.38 % |
| eil101 | 629 | 666 | 5.56 % | 661 | 4.84 % | 671 | 6.26 % | 661 | 4.84 % |
| lin105 | 14379 | 15509 | 7.29 % | 15619 | 7.94 % | 16404 | 12.34 % | 15570 | 7.65 % |
| pr124 | 59030 | 61134 | 3.44 % | 59365 | 0.56 % | 59809 | 1.30 % | 59332 | 0.51 % |
| bier127 | 118282 | 129821 | 8.89 % | 129934 | 8.97 % | 137100 | 13.73 % | 128116 | 7.68 % |
| ch130 | 6110 | 6368 | 4.05 % | 6315 | 3.25 % | 6308 | 3.14 % | 6314 | 3.23 % |
| pr136 | 96772 | 102727 | 5.80 % | 100055 | 3.28 % | 102949 | 6.00 % | 100513 | 3.72 % |
| pr144 | 58537 | 61943 | 5.50 % | 60386 | 3.06 % | 60421 | 3.12 % | 61131 | 4.24 % |
| kroA150 | 26524 | 30112 | 11.92 % | 29083 | 8.80 % | 30478 | 12.97 % | 30438 | 12.86 % |
| kroB150 | 26130 | 29673 | 11.94 % | 29123 | 10.28 % | 28947 | 9.73 % | 28912 | 9.62 % |
| pr152 | 73682 | 81953 | 10.09 % | 76996 | 4.30 % | 78300 | 5.90 % | 78214 | 5.79 % |
| u159 | 42080 | 46594 | 9.69 % | 44452 | 5.34 % | 46503 | 9.51 % | 44917 | 6.32 % |
| rat195 | 2323 | 3095 | 24.94 % | 3075 | 24.46 % | 3088 | 24.77 % | 3100 | 25.06 % |
| kroA200 | 29368 | 34825 | 15.67 % | 34971 | 16.02 % | 34050 | 13.75 % | 34422 | 14.68 % |
| ts225 | 126643 | 144315 | 12.25 % | 137942 | 8.19 % | 139548 | 9.25 % | 138438 | 8.52 % |
| tsp225 | 3919 | 4749 | 17.48 % | 4580 | 14.43 % | 4709 | 16.78 % | 4908 | 20.15 % |
| pr226 | 80369 | 86582 | 7.18 % | 83980 | 4.30 % | 85375 | 5.86 % | 86237 | 6.80 % |
| Avg. Gap | 0.00 % | | 9.24% | | 7.00% | | 8.25% | | 7.72% |

Table F.9: TSPLib results. The models are trained on TSP50. Augmentation decoding (100) is used.

| Instance | Opt. (BKS) | AM | | POMO | | SymNCO | | AM-XL | |
|---|---|---|---|---|---|---|---|---|---|
| | | Cost | Gap ↓ | Cost | Gap ↓ | Cost | Gap ↓ | Cost | Gap ↓ |
| eil51 | 426 | 431 | 1.16 % | 432 | 1.39 % | 429 | 0.70 % | 429 | 0.70 % |
| berlin52 | 7542 | 7897 | 4.50 % | 7722 | 2.33 % | 7576 | 0.45 % | 7674 | 1.72 % |
| st70 | 675 | 678 | 0.44 % | 680 | 0.74 % | 678 | 0.44 % | 678 | 0.44 % |
| eil76 | 538 | 557 | 3.41 % | 559 | 3.76 % | 552 | 2.54 % | 551 | 2.36 % |
| pr76 | 108159 | 110215 | 1.87 % | 109523 | 1.25 % | 109920 | 1.60 % | 109775 | 1.47 % |
| rat99 | 1211 | 1435 | 15.61 % | 1436 | 15.67 % | 1412 | 14.24 % | 1424 | 14.96 % |
| kroA100 | 21282 | 23253 | 8.48 % | 23125 | 7.97 % | 23056 | 7.69 % | 22915 | 7.13 % |
| kroB100 | 22141 | 23987 | 7.70 % | 23840 | 7.13 % | 23583 | 6.11 % | 23381 | 5.30 % |
| kroC100 | 20749 | 22041 | 5.86 % | 22458 | 7.61 % | 22504 | 7.80 % | 21706 | 4.41 % |
| kroD100 | 21294 | 22826 | 6.71 % | 22540 | 5.53 % | 22865 | 6.87 % | 22880 | 6.93 % |
| kroE100 | 22068 | 23436 | 5.84 % | 23623 | 6.58 % | 23396 | 5.68 % | 23257 | 5.11 % |
| rd100 | 7910 | 7935 | 0.32 % | 8006 | 1.20 % | 7945 | 0.44 % | 7972 | 0.78 % |
| eil101 | 629 | 662 | 4.98 % | 665 | 5.41 % | 655 | 3.97 % | 661 | 4.84 % |
| lin105 | 14379 | 15386 | 6.54 % | 15638 | 8.05 % | 15270 | 5.83 % | 15228 | 5.58 % |
| pr124 | 59030 | 60586 | 2.57 % | 60150 | 1.86 % | 59565 | 0.90 % | 59332 | 0.51 % |
| bier127 | 118282 | 124017 | 4.62 % | 129382 | 8.58 % | 127516 | 7.24 % | 125509 | 5.76 % |
| ch130 | 6110 | 6248 | 2.21 % | 6351 | 3.79 % | 6217 | 1.72 % | 6239 | 2.07 % |
| pr136 | 96772 | 100325 | 3.54 % | 99948 | 3.18 % | 99458 | 2.70 % | 98595 | 1.85 % |
| pr144 | 58537 | 60478 | 3.21 % | 59754 | 2.04 % | 59202 | 1.12 % | 59255 | 1.21 % |
| kroA150 | 26524 | 29298 | 9.47 % | 29127 | 8.94 % | 29587 | 10.35 % | 29278 | 9.41 % |
| kroB150 | 26130 | 29116 | 10.26 % | 28840 | 9.40 % | 28469 | 8.22 % | 28498 | 8.31 % |
| pr152 | 73682 | 76378 | 3.53 % | 76074 | 3.14 % | 75825 | 2.83 % | 75495 | 2.40 % |
| u159 | 42080 | 44124 | 4.63 % | 43795 | 3.92 % | 43765 | 3.85 % | 43846 | 4.03 % |
| rat195 | 2323 | 3040 | 23.59 % | 3060 | 24.08 % | 3043 | 23.66 % | 2976 | 21.94 % |
| kroA200 | 29368 | 33751 | 12.99 % | 33743 | 12.97 % | 32690 | 10.16 % | 33660 | 12.75 % |
| ts225 | 126643 | 140185 | 9.66 % | 139927 | 9.49 % | 139024 | 8.91 % | 138401 | 8.50 % |
| tsp225 | 3919 | 4633 | 15.41 % | 4624 | 15.25 % | 4528 | 13.45 % | 4622 | 15.21 % |
| pr226 | 80369 | 83220 | 3.43 % | 83512 | 3.76 % | 83164 | 3.36 % | 83479 | 3.73 % |
| Avg. Gap | 0.00 % | | 6.52% | | 6.61% | | 5.82% | | 5.69% |

Table F.10: CVRPLib results. The models are trained on CVRP20. Greedy decoding is used.

| Instance | Opt. (BKS) | AM | | POMO | | SymNCO | | AM-XL | |
|---|---|---|---|---|---|---|---|---|---|
| | | Cost | Gap ↓ | Cost | Gap ↓ | Cost | Gap ↓ | Cost | Gap ↓ |
| A-n53-k7 | 1010 | 1180 | 14.41 % | 1138 | 11.25 % | 1138 | 11.25 % | 1115 | 9.42 % |
| A-n54-k7 | 1167 | 1272 | 8.25 % | 1351 | 13.62 % | 1300 | 10.23 % | 1303 | 10.44 % |
| A-n55-k9 | 1073 | 1283 | 16.37 % | 1252 | 14.30 % | 1196 | 10.28 % | 1226 | 12.48 % |
| A-n60-k9 | 1354 | 1430 | 5.31 % | 1511 | 10.39 % | 1522 | 11.04 % | 1583 | 14.47 % |
| A-n61-k9 | 1034 | 1186 | 12.82 % | 1222 | 15.38 % | 1201 | 13.91 % | 1244 | 16.88 % |
| A-n62-k8 | 1288 | 1464 | 12.02 % | 1422 | 9.42 % | 1393 | 7.54 % | 1386 | 7.07 % |
| A-n63-k9 | 1616 | 1697 | 4.77 % | 1824 | 11.40 % | 1864 | 13.30 % | 1815 | 10.96 % |
| A-n63-k10 | 1314 | 1371 | 4.16 % | 1577 | 16.68 % | 1536 | 14.45 % | 1549 | 15.17 % |
| A-n64-k9 | 1401 | 1545 | 9.32 % | 1642 | 14.68 % | 1559 | 10.13 % | 1490 | 5.97 % |
| A-n65-k9 | 1174 | 1358 | 13.55 % | 1287 | 8.78 % | 1408 | 16.62 % | 1322 | 11.20 % |
| A-n69-k9 | 1159 | 1345 | 13.83 % | 1360 | 14.78 % | 1319 | 12.13 % | 1334 | 13.12 % |
| A-n80-k10 | 1763 | 2017 | 12.59 % | 2114 | 16.60 % | 2051 | 14.04 % | 2004 | 12.03 % |
| B-n51-k7 | 1032 | 1049 | 1.62 % | 1065 | 3.10 % | 1129 | 8.59 % | 1054 | 2.09 % |
| B-n52-k7 | 747 | 793 | 5.80 % | 893 | 16.35 % | 929 | 19.59 % | 820 | 8.90 % |
| B-n56-k7 | 707 | 789 | 10.39 % | 815 | 13.25 % | 845 | 16.33 % | 905 | 21.88 % |
| B-n57-k7 | 1153 | 1375 | 16.15 % | 1401 | 17.70 % | 1435 | 19.65 % | 1340 | 13.96 % |
| B-n57-k9 | 1598 | 1768 | 9.62 % | 1746 | 8.48 % | 1752 | 8.79 % | 1719 | 7.04 % |
| B-n63-k10 | 1496 | 1629 | 8.16 % | 1627 | 8.05 % | 1712 | 12.62 % | 1696 | 11.79 % |
| B-n64-k9 | 861 | 950 | 9.37 % | 997 | 13.64 % | 1065 | 19.15 % | 1098 | 21.58 % |
| B-n66-k9 | 1316 | 1429 | 7.91 % | 1525 | 13.70 % | 1452 | 9.37 % | 1399 | 5.93 % |
| B-n67-k10 | 1032 | 1163 | 11.26 % | 1151 | 10.34 % | 1237 | 16.57 % | 1183 | 12.76 % |
| B-n68-k9 | 1272 | 1463 | 13.06 % | 1498 | 15.09 % | 1444 | 11.91 % | 1476 | 13.82 % |
| B-n78-k10 | 1221 | 1376 | 11.26 % | 1473 | 17.11 % | 1455 | 16.08 % | 1450 | 15.79 % |
| E-n51-k5 | 521 | 566 | 7.95 % | 629 | 17.17 % | 621 | 16.10 % | 582 | 10.48 % |
| E-n76-k7 | 682 | 840 | 18.81 % | 808 | 15.59 % | 847 | 19.48 % | 860 | 20.70 % |
| E-n76-k8 | 735 | 861 | 14.63 % | 840 | 12.50 % | 884 | 16.86 % | 904 | 18.69 % |
| E-n76-k10 | 830 | 998 | 16.83 % | 957 | 13.27 % | 986 | 15.82 % | 1047 | 20.73 % |
| E-n76-k14 | 1021 | 1151 | 11.29 % | 1232 | 17.13 % | 1216 | 16.04 % | 1184 | 13.77 % |
| E-n101-k8 | 815 | 1113 | 26.77 % | 1051 | 22.45 % | 1183 | 31.11 % | 1101 | 25.98 % |
| E-n101-k14 | 1067 | 1222 | 12.68 % | 1306 | 18.30 % | 1413 | 24.49 % | 1317 | 18.98 % |
| F-n72-k4 | 237 | 290 | 18.28 % | 291 | 18.56 % | 404 | 41.34 % | 344 | 31.10 % |
| F-n135-k7 | 1162 | 1998 | 41.84 % | 2148 | 45.90 % | 2425 | 52.08 % | 2037 | 42.96 % |
| M-n101-k10 | 820 | 1302 | 37.02 % | 1142 | 28.20 % | 1214 | 32.45 % | 1137 | 27.88 % |
| M-n121-k7 | 1034 | 1417 | 27.03 % | 1423 | 27.34 % | 2379 | 56.54 % | 1617 | 36.05 % |
| M-n151-k12 | 1015 | 1284 | 20.95 % | 1374 | 26.13 % | 2013 | 49.58 % | 1582 | 35.84 % |
| M-n200-k16 | 1274 | 1845 | 30.95 % | 1790 | 28.83 % | 2955 | 56.89 % | 1988 | 35.92 % |
| M-n200-k17 | 1275 | 1845 | 30.89 % | 1790 | 28.77 % | 2955 | 56.85 % | 1988 | 35.87 % |
| Avg. Gap | 0.00 % | | 14.81% | | 16.60% | | 21.33% | | 17.56% |

974

Table F.11: CVRPLib results. The models are trained on CVRP20. Sampling decoding (100 with temperature $\tau = 0.05$) is used.

| Instance | Opt. (BKS) | AM | | POMO | | SymNCO | | AM-XL | |
|---|---|---|---|---|---|---|---|---|---|
| | | Cost | Gap ↓ | Cost | Gap ↓ | Cost | Gap ↓ | Cost | Gap ↓ |
| A-n53-k7 | 1010 | 1171 | 13.75 % | 1073 | 5.87 % | 1138 | 11.25 % | 1096 | 7.85 % |
| A-n54-k7 | 1167 | 1257 | 7.16 % | 1257 | 7.16 % | 1299 | 10.16 % | 1271 | 8.18 % |
| A-n55-k9 | 1073 | 1268 | 15.38 % | 1191 | 9.91 % | 1174 | 8.60 % | 1224 | 12.34 % |
| A-n60-k9 | 1354 | 1430 | 5.31 % | 1484 | 8.76 % | 1522 | 11.04 % | 1498 | 9.61 % |
| A-n61-k9 | 1034 | 1112 | 7.01 % | 1183 | 12.60 % | 1136 | 8.98 % | 1202 | 13.98 % |
| A-n62-k8 | 1288 | 1418 | 9.17 % | 1417 | 9.10 % | 1369 | 5.92 % | 1381 | 6.73 % |
| A-n63-k9 | 1616 | 1697 | 4.77 % | 1807 | 10.57 % | 1844 | 12.36 % | 1741 | 7.18 % |
| A-n63-k10 | 1314 | 1366 | 3.81 % | 1443 | 8.94 % | 1441 | 8.81 % | 1488 | 11.69 % |
| A-n64-k9 | 1401 | 1541 | 9.09 % | 1592 | 12.00 % | 1518 | 7.71 % | 1483 | 5.53 % |
| A-n65-k9 | 1174 | 1304 | 9.97 % | 1282 | 8.42 % | 1355 | 13.36 % | 1294 | 9.27 % |
| A-n69-k9 | 1159 | 1303 | 11.05 % | 1288 | 10.02 % | 1272 | 8.88 % | 1317 | 12.00 % |
| A-n80-k10 | 1763 | 2004 | 12.03 % | 2037 | 13.45 % | 2042 | 13.66 % | 1986 | 11.23 % |
| B-n51-k7 | 1032 | 1049 | 1.62 % | 1054 | 2.09 % | 1121 | 7.94 % | 1051 | 1.81 % |
| B-n52-k7 | 747 | 787 | 5.08 % | 805 | 7.20 % | 927 | 19.42 % | 803 | 6.97 % |
| B-n56-k7 | 707 | 779 | 9.24 % | 789 | 10.39 % | 827 | 14.51 % | 786 | 10.05 % |
| B-n57-k7 | 1153 | 1366 | 15.59 % | 1253 | 7.98 % | 1238 | 6.87 % | 1326 | 13.05 % |
| B-n57-k9 | 1598 | 1687 | 5.28 % | 1728 | 7.52 % | 1747 | 8.53 % | 1711 | 6.60 % |
| B-n63-k10 | 1496 | 1628 | 8.11 % | 1576 | 5.08 % | 1654 | 9.55 % | 1673 | 10.58 % |
| B-n64-k9 | 861 | 944 | 8.79 % | 974 | 11.60 % | 996 | 13.55 % | 1011 | 14.84 % |
| B-n66-k9 | 1316 | 1419 | 7.26 % | 1449 | 9.18 % | 1389 | 5.26 % | 1368 | 3.80 % |
| B-n67-k10 | 1032 | 1161 | 11.11 % | 1095 | 5.75 % | 1132 | 8.83 % | 1169 | 11.72 % |
| B-n68-k9 | 1272 | 1444 | 11.91 % | 1454 | 12.52 % | 1415 | 10.11 % | 1371 | 7.22 % |
| B-n78-k10 | 1221 | 1371 | 10.94 % | 1375 | 11.20 % | 1364 | 10.48 % | 1366 | 10.61 % |
| E-n51-k5 | 521 | 563 | 7.46 % | 562 | 7.30 % | 593 | 12.14 % | 580 | 10.17 % |
| E-n76-k7 | 682 | 818 | 16.63 % | 795 | 14.21 % | 806 | 15.38 % | 808 | 15.59 % |
| E-n76-k8 | 735 | 860 | 14.53 % | 824 | 10.80 % | 842 | 12.71 % | 852 | 13.73 % |
| E-n76-k10 | 830 | 959 | 13.45 % | 939 | 11.61 % | 940 | 11.70 % | 919 | 9.68 % |
| E-n76-k14 | 1021 | 1148 | 11.06 % | 1147 | 10.99 % | 1141 | 10.52 % | 1139 | 10.36 % |
| E-n101-k8 | 815 | 1003 | 18.74 % | 1012 | 19.47 % | 1155 | 29.44 % | 1046 | 22.08 % |
| E-n101-k14 | 1067 | 1214 | 12.11 % | 1302 | 18.05 % | 1286 | 17.03 % | 1297 | 17.73 % |
| F-n72-k4 | 237 | 290 | 18.28 % | 288 | 17.71 % | 356 | 33.43 % | 311 | 23.79 % |
| F-n135-k7 | 1162 | 1785 | 34.90 % | 2380 | 51.18 % | 2312 | 49.74 % | 1772 | 34.42 % |
| M-n101-k10 | 820 | 1127 | 27.24 % | 1159 | 29.25 % | 1137 | 27.88 % | 1104 | 25.72 % |
| M-n121-k7 | 1034 | 1386 | 25.40 % | 1575 | 34.35 % | 1972 | 47.57 % | 1407 | 26.51 % |
| M-n151-k12 | 1015 | 1249 | 18.73 % | 1607 | 36.84 % | 2114 | 51.99 % | 1515 | 33.00 % |
| M-n200-k16 | 1274 | 1652 | 22.88 % | 2424 | 47.44 % | 2802 | 54.53 % | 2013 | 36.71 % |
| M-n200-k17 | 1275 | 1640 | 22.26 % | 2414 | 47.18 % | 2807 | 54.58 % | 2003 | 36.35 % |
| Avg. Gap | 0.00 % | | 12.62% | | 15.23% | | 17.96% | | 14.29% |

Table F.12: CVRPLib results. The models are trained on CVRP20. Greedy multi-start decoding is used.

| Instance | Opt. (BKS) | AM | | POMO | | SymNCO | | AM-XL | |
|---|---|---|---|---|---|---|---|---|---|
| | | Cost | Gap ↓ | Cost | Gap ↓ | Cost | Gap ↓ | Cost | Gap ↓ |
| A-n53-k7 | 1010 | 1078 | 6.31 % | 1038 | 2.70 % | 1059 | 4.63 % | 1069 | 5.52 % |
| A-n54-k7 | 1167 | 1262 | 7.53 % | 1273 | 8.33 % | 1258 | 7.23 % | 1250 | 6.64 % |
| A-n55-k9 | 1073 | 1139 | 5.79 % | 1202 | 10.73 % | 1154 | 7.02 % | 1218 | 11.90 % |
| A-n60-k9 | 1354 | 1430 | 5.31 % | 1442 | 6.10 % | 1482 | 8.64 % | 1508 | 10.21 % |
| A-n61-k9 | 1034 | 1125 | 8.09 % | 1170 | 11.62 % | 1129 | 8.41 % | 1169 | 11.55 % |
| A-n62-k8 | 1288 | 1389 | 7.27 % | 1380 | 6.67 % | 1385 | 7.00 % | 1375 | 6.33 % |
| A-n63-k9 | 1616 | 1689 | 4.32 % | 1762 | 8.29 % | 1789 | 9.67 % | 1760 | 8.18 % |
| A-n63-k10 | 1314 | 1371 | 4.16 % | 1480 | 11.22 % | 1396 | 5.87 % | 1422 | 7.59 % |
| A-n64-k9 | 1401 | 1545 | 9.32 % | 1512 | 7.34 % | 1510 | 7.22 % | 1486 | 5.72 % |
| A-n65-k9 | 1174 | 1243 | 5.55 % | 1286 | 8.71 % | 1344 | 12.65 % | 1285 | 8.64 % |
| A-n69-k9 | 1159 | 1256 | 7.72 % | 1279 | 9.38 % | 1248 | 7.13 % | 1299 | 10.78 % |
| A-n80-k10 | 1763 | 1981 | 11.00 % | 2032 | 13.24 % | 1980 | 10.96 % | 1931 | 8.70 % |
| B-n51-k7 | 1032 | 1049 | 1.62 % | 1052 | 1.90 % | 1081 | 4.53 % | 1044 | 1.15 % |
| B-n52-k7 | 747 | 791 | 5.56 % | 822 | 9.12 % | 857 | 12.84 % | 800 | 6.62 % |
| B-n56-k7 | 707 | 757 | 6.61 % | 761 | 7.10 % | 805 | 12.17 % | 774 | 8.66 % |
| B-n57-k7 | 1153 | 1298 | 11.17 % | 1212 | 4.87 % | 1231 | 6.34 % | 1258 | 8.35 % |
| B-n57-k9 | 1598 | 1753 | 8.84 % | 1689 | 5.39 % | 1711 | 6.60 % | 1665 | 4.02 % |
| B-n63-k10 | 1496 | 1629 | 8.16 % | 1560 | 4.10 % | 1624 | 7.88 % | 1635 | 8.50 % |
| B-n64-k9 | 861 | 947 | 9.08 % | 953 | 9.65 % | 993 | 13.29 % | 998 | 13.73 % |
| B-n66-k9 | 1316 | 1423 | 7.52 % | 1423 | 7.52 % | 1388 | 5.19 % | 1396 | 5.73 % |
| B-n67-k10 | 1032 | 1152 | 10.42 % | 1120 | 7.86 % | 1130 | 8.67 % | 1145 | 9.87 % |
| B-n68-k9 | 1272 | 1410 | 9.79 % | 1443 | 11.85 % | 1334 | 4.65 % | 1352 | 5.92 % |
| B-n78-k10 | 1221 | 1368 | 10.75 % | 1367 | 10.68 % | 1340 | 8.88 % | 1357 | 10.02 % |
| E-n51-k5 | 521 | 561 | 7.13 % | 579 | 10.02 % | 590 | 11.69 % | 564 | 7.62 % |
| E-n76-k7 | 682 | 771 | 11.54 % | 777 | 12.23 % | 822 | 17.03 % | 791 | 13.78 % |
| E-n76-k8 | 735 | 825 | 10.91 % | 796 | 7.66 % | 861 | 14.63 % | 827 | 11.12 % |
| E-n76-k10 | 830 | 942 | 11.89 % | 928 | 10.56 % | 950 | 12.63 % | 917 | 9.49 % |
| E-n76-k14 | 1021 | 1099 | 7.10 % | 1150 | 11.22 % | 1135 | 10.04 % | 1134 | 9.96 % |
| E-n101-k8 | 815 | 979 | 16.75 % | 979 | 16.75 % | 1117 | 27.04 % | 1042 | 21.79 % |
| E-n101-k14 | 1067 | 1215 | 12.18 % | 1254 | 14.91 % | 1239 | 13.88 % | 1262 | 15.45 % |
| F-n72-k4 | 237 | 284 | 16.55 % | 291 | 18.56 % | 359 | 33.98 % | 317 | 25.24 % |
| F-n135-k7 | 1162 | 1820 | 36.15 % | 1883 | 38.29 % | 2158 | 46.15 % | 1676 | 30.67 % |
| M-n101-k10 | 820 | 1058 | 22.50 % | 1056 | 22.35 % | 1121 | 26.85 % | 1032 | 20.54 % |
| M-n121-k7 | 1034 | 1355 | 23.69 % | 1302 | 20.58 % | 1936 | 46.59 % | 1415 | 26.93 % |
| M-n151-k12 | 1015 | 1274 | 20.33 % | 1309 | 22.46 % | 1856 | 45.31 % | 1435 | 29.27 % |
| M-n200-k16 | 1274 | 1606 | 20.67 % | 1654 | 22.97 % | 2519 | 49.42 % | 1883 | 32.34 % |
| M-n200-k17 | 1275 | 1606 | 20.61 % | 1654 | 22.91 % | 2519 | 49.38 % | 1883 | 32.29 % |
| Avg. Gap | 0.00 % | | 11.08% | | 11.78% | | 16.00% | | 12.72% |

Table F.13: CVRPLib results. The models are trained on CVRP20. Augmentation decoding (100) is used.

| Instance | Opt. (BKS) | AM | | POMO | | SymNCO | | AM-XL | |
|---|---|---|---|---|---|---|---|---|---|
| | | Cost | Gap ↓ | Cost | Gap ↓ | Cost | Gap ↓ | Cost | Gap ↓ |
| A-n53-k7 | 1010 | 1051 | 3.90 % | 1028 | 1.75 % | 1080 | 6.48 % | 1094 | 7.68 % |
| A-n54-k7 | 1167 | 1204 | 3.07 % | 1255 | 7.01 % | 1224 | 4.66 % | 1256 | 7.09 % |
| A-n55-k9 | 1073 | 1139 | 5.79 % | 1153 | 6.94 % | 1155 | 7.10 % | 1174 | 8.60 % |
| A-n60-k9 | 1354 | 1425 | 4.98 % | 1438 | 5.84 % | 1481 | 8.58 % | 1459 | 7.20 % |
| A-n61-k9 | 1034 | 1119 | 7.60 % | 1144 | 9.62 % | 1127 | 8.25 % | 1149 | 10.01 % |
| A-n62-k8 | 1288 | 1349 | 4.52 % | 1410 | 8.65 % | 1374 | 6.26 % | 1376 | 6.40 % |
| A-n63-k9 | 1616 | 1692 | 4.49 % | 1715 | 5.77 % | 1767 | 8.55 % | 1741 | 7.18 % |
| A-n63-k10 | 1314 | 1364 | 3.67 % | 1380 | 4.78 % | 1368 | 3.95 % | 1395 | 5.81 % |
| A-n64-k9 | 1401 | 1490 | 5.97 % | 1501 | 6.66 % | 1513 | 7.40 % | 1490 | 5.97 % |
| A-n65-k9 | 1174 | 1251 | 6.16 % | 1258 | 6.68 % | 1255 | 6.45 % | 1263 | 7.05 % |
| A-n69-k9 | 1159 | 1250 | 7.28 % | 1263 | 8.23 % | 1238 | 6.38 % | 1254 | 7.58 % |
| A-n80-k10 | 1763 | 1907 | 7.55 % | 1937 | 8.98 % | 1946 | 9.40 % | 1928 | 8.56 % |
| B-n51-k7 | 1032 | 1040 | 0.77 % | 1040 | 0.77 % | 1053 | 1.99 % | 1041 | 0.86 % |
| B-n52-k7 | 747 | 770 | 2.99 % | 775 | 3.61 % | 755 | 1.06 % | 765 | 2.35 % |
| B-n56-k7 | 707 | 765 | 7.58 % | 761 | 7.10 % | 769 | 8.06 % | 766 | 7.70 % |
| B-n57-k7 | 1153 | 1215 | 5.10 % | 1183 | 2.54 % | 1231 | 6.34 % | 1183 | 2.54 % |
| B-n57-k9 | 1598 | 1655 | 3.44 % | 1656 | 3.50 % | 1681 | 4.94 % | 1635 | 2.26 % |
| B-n63-k10 | 1496 | 1603 | 6.67 % | 1553 | 3.67 % | 1604 | 6.73 % | 1603 | 6.67 % |
| B-n64-k9 | 861 | 940 | 8.40 % | 932 | 7.62 % | 922 | 6.62 % | 950 | 9.37 % |
| B-n66-k9 | 1316 | 1407 | 6.47 % | 1406 | 6.40 % | 1375 | 4.29 % | 1357 | 3.02 % |
| B-n67-k10 | 1032 | 1113 | 7.28 % | 1095 | 5.75 % | 1096 | 5.84 % | 1097 | 5.93 % |
| B-n68-k9 | 1272 | 1339 | 5.00 % | 1345 | 5.43 % | 1337 | 4.86 % | 1371 | 7.22 % |
| B-n78-k10 | 1221 | 1349 | 9.49 % | 1348 | 9.42 % | 1315 | 7.15 % | 1335 | 8.54 % |
| E-n51-k5 | 521 | 536 | 2.80 % | 565 | 7.79 % | 573 | 9.08 % | 563 | 7.46 % |
| E-n76-k7 | 682 | 771 | 11.54 % | 747 | 8.70 % | 781 | 12.68 % | 762 | 10.50 % |
| E-n76-k8 | 735 | 819 | 10.26 % | 805 | 8.70 % | 826 | 11.02 % | 815 | 9.82 % |
| E-n76-k10 | 830 | 911 | 8.89 % | 919 | 9.68 % | 915 | 9.29 % | 919 | 9.68 % |
| E-n76-k14 | 1021 | 1097 | 6.93 % | 1109 | 7.94 % | 1131 | 9.73 % | 1114 | 8.35 % |
| E-n101-k8 | 815 | 948 | 14.03 % | 922 | 11.61 % | 1093 | 25.43 % | 946 | 13.85 % |
| E-n101-k14 | 1067 | 1184 | 9.88 % | 1212 | 11.96 % | 1219 | 12.47 % | 1225 | 12.90 % |
| F-n72-k4 | 237 | 274 | 13.50 % | 288 | 17.71 % | 305 | 22.30 % | 291 | 18.56 % |
| F-n135-k7 | 1162 | 1596 | 27.19 % | 1505 | 22.79 % | 1735 | 33.03 % | 1597 | 27.24 % |
| M-n101-k10 | 820 | 1017 | 19.37 % | 1015 | 19.21 % | 1089 | 24.70 % | 1061 | 22.71 % |
| M-n121-k7 | 1034 | 1233 | 16.14 % | 1251 | 17.35 % | 1662 | 37.79 % | 1312 | 21.19 % |
| M-n151-k12 | 1015 | 1208 | 15.98 % | 1236 | 17.88 % | 1655 | 38.67 % | 1334 | 23.91 % |
| M-n200-k16 | 1274 | 1559 | 18.28 % | 1668 | 23.62 % | 2360 | 46.02 % | 1815 | 29.81 % |
| M-n200-k17 | 1275 | 1572 | 18.89 % | 1704 | 25.18 % | 2423 | 47.38 % | 1791 | 28.81 % |
| Avg. Gap | 0.00 % | | 8.70% | | 9.37% | | 13.00% | | 10.55% |

Table F.14: CVRPLib results. The models are trained on CVRP50. Greedy decoding is used.

| Instance | Opt. (BKS) | AM | | POMO | | SymNCO | | AM-XL | |
|---|---|---|---|---|---|---|---|---|---|
| | | Cost | Gap ↓ | Cost | Gap ↓ | Cost | Gap ↓ | Cost | Gap ↓ |
| A-n53-k7 | 1010 | 1077 | 6.22 % | 1071 | 5.70 % | 1086 | 7.00 % | 1079 | 6.39 % |
| A-n54-k7 | 1167 | 1251 | 6.71 % | 1228 | 4.97 % | 1211 | 3.63 % | 1213 | 3.79 % |
| A-n55-k9 | 1073 | 1166 | 7.98 % | 1107 | 3.07 % | 1158 | 7.34 % | 1203 | 10.81 % |
| A-n60-k9 | 1354 | 1460 | 7.26 % | 1419 | 4.58 % | 1452 | 6.75 % | 1417 | 4.45 % |
| A-n61-k9 | 1034 | 1079 | 4.17 % | 1089 | 5.05 % | 1050 | 1.52 % | 1126 | 8.17 % |
| A-n62-k8 | 1288 | 1367 | 5.78 % | 1385 | 7.00 % | 1369 | 5.92 % | 1331 | 3.23 % |
| A-n63-k9 | 1616 | 1682 | 3.92 % | 1679 | 3.75 % | 1671 | 3.29 % | 1659 | 2.59 % |
| A-n63-k10 | 1314 | 1347 | 2.45 % | 1426 | 7.85 % | 1398 | 6.01 % | 1402 | 6.28 % |
| A-n64-k9 | 1401 | 1493 | 6.16 % | 1436 | 2.44 % | 1469 | 4.63 % | 1469 | 4.63 % |
| A-n65-k9 | 1174 | 1247 | 5.85 % | 1247 | 5.85 % | 1216 | 3.45 % | 1255 | 6.45 % |
| A-n69-k9 | 1159 | 1264 | 8.31 % | 1210 | 4.21 % | 1232 | 5.93 % | 1224 | 5.31 % |
| A-n80-k10 | 1763 | 1864 | 5.42 % | 1923 | 8.32 % | 1921 | 8.22 % | 1881 | 6.27 % |
| B-n51-k7 | 1032 | 1134 | 8.99 % | 1153 | 10.49 % | 1166 | 11.49 % | 1116 | 7.53 % |
| B-n52-k7 | 747 | 770 | 2.99 % | 770 | 2.99 % | 809 | 7.66 % | 784 | 4.72 % |
| B-n56-k7 | 707 | 751 | 5.86 % | 748 | 5.48 % | 813 | 13.04 % | 779 | 9.24 % |
| B-n57-k7 | 1153 | 1182 | 2.45 % | 1227 | 6.03 % | 1189 | 3.03 % | 1970 | 41.47 % |
| B-n57-k9 | 1598 | 1670 | 4.31 % | 1686 | 5.22 % | 1660 | 3.73 % | 1660 | 3.73 % |
| B-n63-k10 | 1496 | 1634 | 8.45 % | 1644 | 9.00 % | 1621 | 7.71 % | 1598 | 6.38 % |
| B-n64-k9 | 861 | 964 | 10.68 % | 970 | 11.24 % | 972 | 11.42 % | 922 | 6.62 % |
| B-n66-k9 | 1316 | 1374 | 4.22 % | 1363 | 3.45 % | 1362 | 3.38 % | 1388 | 5.19 % |
| B-n67-k10 | 1032 | 1137 | 9.23 % | 1148 | 10.10 % | 1135 | 9.07 % | 1189 | 13.20 % |
| B-n68-k9 | 1272 | 1397 | 8.95 % | 1329 | 4.29 % | 1317 | 3.42 % | 1374 | 7.42 % |
| B-n78-k10 | 1221 | 1325 | 7.85 % | 1316 | 7.22 % | 1320 | 7.50 % | 1294 | 5.64 % |
| E-n51-k5 | 521 | 554 | 5.96 % | 582 | 10.48 % | 567 | 8.11 % | 569 | 8.44 % |
| E-n76-k7 | 682 | 712 | 4.21 % | 744 | 8.33 % | 750 | 9.07 % | 740 | 7.84 % |
| E-n76-k8 | 735 | 760 | 3.29 % | 816 | 9.93 % | 795 | 7.55 % | 776 | 5.28 % |
| E-n76-k10 | 830 | 867 | 4.27 % | 901 | 7.88 % | 893 | 7.05 % | 880 | 5.68 % |
| E-n76-k14 | 1021 | 1075 | 5.02 % | 1076 | 5.11 % | 1133 | 9.89 % | 1115 | 8.43 % |
| E-n101-k8 | 815 | 920 | 11.41 % | 900 | 9.44 % | 896 | 9.04 % | 892 | 8.63 % |
| E-n101-k14 | 1067 | 1171 | 8.88 % | 1171 | 8.88 % | 1193 | 10.56 % | 1139 | 6.32 % |
| F-n72-k4 | 237 | 305 | 22.30 % | 295 | 19.66 % | 320 | 25.94 % | 293 | 19.11 % |
| F-n135-k7 | 1162 | 1499 | 22.48 % | 1553 | 25.18 % | 1734 | 32.99 % | 2300 | 49.48 % |
| M-n101-k10 | 820 | 929 | 11.73 % | 917 | 10.58 % | 982 | 16.50 % | 888 | 7.66 % |
| M-n121-k7 | 1034 | 1209 | 14.47 % | 1300 | 20.46 % | 1373 | 24.69 % | 1177 | 12.15 % |
| M-n151-k12 | 1015 | 1163 | 12.73 % | 1161 | 12.58 % | 1261 | 19.51 % | 1157 | 12.27 % |
| M-n200-k16 | 1274 | 1597 | 20.23 % | 1505 | 15.35 % | 1742 | 26.87 % | 1479 | 13.86 % |
| M-n200-k17 | 1275 | 1597 | 20.16 % | 1505 | 15.28 % | 1742 | 26.81 % | 1479 | 13.79 % |
| Avg. Gap | 0.00 % | | 8.42% | | 8.58% | | 10.26% | | 9.69% |

Table F.15: CVRPLib results. The models are trained on CVRP50. Sampling decoding (100 with temperature $\tau = 0.05$) is used.

| Instance | Opt. (BKS) | AM | | POMO | | SymNCO | | AM-XL | |
|---|---|---|---|---|---|---|---|---|---|
| | | Cost | Gap ↓ | Cost | Gap ↓ | Cost | Gap ↓ | Cost | Gap ↓ |
| A-n53-k7 | 1010 | 1072 | 5.78 % | 1071 | 5.70 % | 1086 | 7.00 % | 1079 | 6.39 % |
| A-n54-k7 | 1167 | 1251 | 6.71 % | 1204 | 3.07 % | 1211 | 3.63 % | 1213 | 3.79 % |
| A-n55-k9 | 1073 | 1166 | 7.98 % | 1107 | 3.07 % | 1158 | 7.34 % | 1203 | 10.81 % |
| A-n60-k9 | 1354 | 1457 | 7.07 % | 1408 | 3.84 % | 1448 | 6.49 % | 1417 | 4.45 % |
| A-n61-k9 | 1034 | 1076 | 3.90 % | 1089 | 5.05 % | 1050 | 1.52 % | 1123 | 7.93 % |
| A-n62-k8 | 1288 | 1361 | 5.36 % | 1382 | 6.80 % | 1368 | 5.85 % | 1328 | 3.01 % |
| A-n63-k9 | 1616 | 1670 | 3.23 % | 1679 | 3.75 % | 1667 | 3.06 % | 1659 | 2.59 % |
| A-n63-k10 | 1314 | 1346 | 2.38 % | 1366 | 3.81 % | 1398 | 6.01 % | 1402 | 6.28 % |
| A-n64-k9 | 1401 | 1493 | 6.16 % | 1430 | 2.03 % | 1460 | 4.04 % | 1465 | 4.37 % |
| A-n65-k9 | 1174 | 1241 | 5.40 % | 1229 | 4.48 % | 1216 | 3.45 % | 1255 | 6.45 % |
| A-n69-k9 | 1159 | 1234 | 6.08 % | 1199 | 3.34 % | 1230 | 5.77 % | 1219 | 4.92 % |
| A-n80-k10 | 1763 | 1862 | 5.32 % | 1882 | 6.32 % | 1904 | 7.41 % | 1881 | 6.27 % |
| B-n51-k7 | 1032 | 1134 | 8.99 % | 1153 | 10.49 % | 1166 | 11.49 % | 1116 | 7.53 % |
| B-n52-k7 | 747 | 769 | 2.86 % | 768 | 2.73 % | 809 | 7.66 % | 784 | 4.72 % |
| B-n56-k7 | 707 | 751 | 5.86 % | 747 | 5.35 % | 811 | 12.82 % | 762 | 7.22 % |
| B-n57-k7 | 1153 | 1181 | 2.37 % | 1225 | 5.88 % | 1182 | 2.45 % | 1703 | 32.30 % |
| B-n57-k9 | 1598 | 1669 | 4.25 % | 1680 | 4.88 % | 1660 | 3.73 % | 1654 | 3.39 % |
| B-n63-k10 | 1496 | 1634 | 8.45 % | 1592 | 6.03 % | 1620 | 7.65 % | 1598 | 6.38 % |
| B-n64-k9 | 861 | 955 | 9.84 % | 967 | 10.96 % | 972 | 11.42 % | 922 | 6.62 % |
| B-n66-k9 | 1316 | 1371 | 4.01 % | 1356 | 2.95 % | 1362 | 3.38 % | 1383 | 4.84 % |
| B-n67-k10 | 1032 | 1136 | 9.15 % | 1142 | 9.63 % | 1133 | 8.91 % | 1180 | 12.54 % |
| B-n68-k9 | 1272 | 1383 | 8.03 % | 1324 | 3.93 % | 1317 | 3.42 % | 1374 | 7.42 % |
| B-n78-k10 | 1221 | 1325 | 7.85 % | 1285 | 4.98 % | 1316 | 7.22 % | 1294 | 5.64 % |
| E-n51-k5 | 521 | 551 | 5.44 % | 574 | 9.23 % | 565 | 7.79 % | 569 | 8.44 % |
| E-n76-k7 | 682 | 707 | 3.54 % | 738 | 7.59 % | 744 | 8.33 % | 740 | 7.84 % |
| E-n76-k8 | 735 | 760 | 3.29 % | 816 | 9.93 % | 786 | 6.49 % | 774 | 5.04 % |
| E-n76-k10 | 830 | 862 | 3.71 % | 895 | 7.26 % | 893 | 7.05 % | 876 | 5.25 % |
| E-n76-k14 | 1021 | 1075 | 5.02 % | 1076 | 5.11 % | 1128 | 9.49 % | 1105 | 7.60 % |
| E-n101-k8 | 815 | 894 | 8.84 % | 891 | 8.53 % | 890 | 8.43 % | 888 | 8.22 % |
| E-n101-k14 | 1067 | 1171 | 8.88 % | 1161 | 8.10 % | 1166 | 8.49 % | 1138 | 6.24 % |
| F-n72-k4 | 237 | 305 | 22.30 % | 293 | 19.11 % | 307 | 22.80 % | 289 | 17.99 % |
| F-n135-k7 | 1162 | 1453 | 20.03 % | 1521 | 23.60 % | 1557 | 25.37 % | 2141 | 45.73 % |
| M-n101-k10 | 820 | 927 | 11.54 % | 915 | 10.38 % | 969 | 15.38 % | 888 | 7.66 % |
| M-n121-k7 | 1034 | 1145 | 9.69 % | 1276 | 18.97 % | 1347 | 23.24 % | 1121 | 7.76 % |
| M-n151-k12 | 1015 | 1148 | 11.59 % | 1128 | 10.02 % | 1228 | 17.35 % | 1120 | 9.38 % |
| M-n200-k16 | 1274 | 1520 | 16.18 % | 1486 | 14.27 % | 1642 | 22.41 % | 1462 | 12.86 % |
| M-n200-k17 | 1275 | 1558 | 18.16 % | 1475 | 13.56 % | 1664 | 23.38 % | 1460 | 12.67 % |
| Avg. Gap | 0.00 % | | 7.71% | | 7.70% | | 9.40% | | 8.88% |

Table F.16: CVRPLib results. The models are trained on CVRP50. Greedy multi-start decoding is used.

| Instance | Opt. (BKS) | AM | | POMO | | SymNCO | | AM-XL | |
|---|---|---|---|---|---|---|---|---|---|
| | | Cost | Gap ↓ | Cost | Gap ↓ | Cost | Gap ↓ | Cost | Gap ↓ |
| A-n53-k7 | 1010 | 1063 | 4.99 % | 1061 | 4.81 % | 1071 | 5.70 % | 1059 | 4.63 % |
| A-n54-k7 | 1167 | 1217 | 4.11 % | 1188 | 1.77 % | 1201 | 2.83 % | 1211 | 3.63 % |
| A-n55-k9 | 1073 | 1116 | 3.85 % | 1096 | 2.10 % | 1117 | 3.94 % | 1144 | 6.21 % |
| A-n60-k9 | 1354 | 1410 | 3.97 % | 1393 | 2.80 % | 1390 | 2.59 % | 1379 | 1.81 % |
| A-n61-k9 | 1034 | 1067 | 3.09 % | 1077 | 3.99 % | 1050 | 1.52 % | 1082 | 4.44 % |
| A-n62-k8 | 1288 | 1346 | 4.31 % | 1348 | 4.45 % | 1354 | 4.87 % | 1331 | 3.23 % |
| A-n63-k9 | 1616 | 1682 | 3.92 % | 1659 | 2.59 % | 1668 | 3.12 % | 1657 | 2.47 % |
| A-n63-k10 | 1314 | 1347 | 2.45 % | 1336 | 1.65 % | 1362 | 3.52 % | 1348 | 2.52 % |
| A-n64-k9 | 1401 | 1486 | 5.72 % | 1426 | 1.75 % | 1438 | 2.57 % | 1461 | 4.11 % |
| A-n65-k9 | 1174 | 1231 | 4.63 % | 1220 | 3.77 % | 1216 | 3.45 % | 1226 | 4.24 % |
| A-n69-k9 | 1159 | 1189 | 2.52 % | 1189 | 2.52 % | 1200 | 3.42 % | 1209 | 4.14 % |
| A-n80-k10 | 1763 | 1845 | 4.44 % | 1834 | 3.87 % | 1840 | 4.18 % | 1811 | 2.65 % |
| B-n51-k7 | 1032 | 1038 | 0.58 % | 1037 | 0.48 % | 1078 | 4.27 % | 1078 | 4.27 % |
| B-n52-k7 | 747 | 769 | 2.86 % | 764 | 2.23 % | 797 | 6.27 % | 768 | 2.73 % |
| B-n56-k7 | 707 | 748 | 5.48 % | 743 | 4.85 % | 786 | 10.05 % | 774 | 8.66 % |
| B-n57-k7 | 1153 | 1177 | 2.04 % | 1171 | 1.54 % | 1169 | 1.37 % | 1680 | 31.37 % |
| B-n57-k9 | 1598 | 1662 | 3.85 % | 1651 | 3.21 % | 1655 | 3.44 % | 1642 | 2.68 % |
| B-n63-k10 | 1496 | 1612 | 7.20 % | 1590 | 5.91 % | 1606 | 6.85 % | 1598 | 6.38 % |
| B-n64-k9 | 861 | 936 | 8.01 % | 943 | 8.70 % | 939 | 8.31 % | 922 | 6.62 % |
| B-n66-k9 | 1316 | 1365 | 3.59 % | 1348 | 2.37 % | 1351 | 2.59 % | 1376 | 4.36 % |
| B-n67-k10 | 1032 | 1123 | 8.10 % | 1111 | 7.11 % | 1128 | 8.51 % | 1137 | 9.23 % |
| B-n68-k9 | 1272 | 1345 | 5.43 % | 1315 | 3.27 % | 1312 | 3.05 % | 1351 | 5.85 % |
| B-n78-k10 | 1221 | 1289 | 5.28 % | 1278 | 4.46 % | 1298 | 5.93 % | 1288 | 5.20 % |
| E-n51-k5 | 521 | 554 | 5.96 % | 533 | 2.25 % | 553 | 5.79 % | 567 | 8.11 % |
| E-n76-k7 | 682 | 711 | 4.08 % | 719 | 5.15 % | 742 | 8.09 % | 707 | 3.54 % |
| E-n76-k8 | 735 | 760 | 3.29 % | 770 | 4.55 % | 765 | 3.92 % | 776 | 5.28 % |
| E-n76-k10 | 830 | 854 | 2.81 % | 870 | 4.60 % | 890 | 6.74 % | 878 | 5.47 % |
| E-n76-k14 | 1021 | 1062 | 3.86 % | 1059 | 3.59 % | 1078 | 5.29 % | 1073 | 4.85 % |
| E-n101-k8 | 815 | 893 | 8.73 % | 890 | 8.43 % | 885 | 7.91 % | 890 | 8.43 % |
| E-n101-k14 | 1067 | 1154 | 7.54 % | 1155 | 7.62 % | 1182 | 9.73 % | 1139 | 6.32 % |
| F-n72-k4 | 237 | 295 | 19.66 % | 268 | 11.57 % | 294 | 19.39 % | 290 | 18.28 % |
| F-n135-k7 | 1162 | 1414 | 17.82 % | 1409 | 17.53 % | 1554 | 25.23 % | 1984 | 41.43 % |
| M-n101-k10 | 820 | 877 | 6.50 % | 892 | 8.07 % | 922 | 11.06 % | 886 | 7.45 % |
| M-n121-k7 | 1034 | 1151 | 10.17 % | 1234 | 16.21 % | 1352 | 23.52 % | 1113 | 7.10 % |
| M-n151-k12 | 1015 | 1146 | 11.43 % | 1141 | 11.04 % | 1256 | 19.19 % | 1118 | 9.21 % |
| M-n200-k16 | 1274 | 1537 | 17.11 % | 1477 | 13.74 % | 1629 | 21.79 % | 1453 | 12.32 % |
| M-n200-k17 | 1275 | 1537 | 17.05 % | 1477 | 13.68 % | 1629 | 21.73 % | 1453 | 12.25 % |
| Avg. Gap | 0.00 % | | 6.39% | | 5.63% | | 7.88% | | 7.61% |

Table F.17: CVRPLib results. The models are trained on CVRP50. Augmentation decoding (100) is used.

| Instance | Opt. (BKS) | AM | | POMO | | SymNCO | | AM-XL | |
|---|---|---|---|---|---|---|---|---|---|
| | | Cost | Gap ↓ | Cost | Gap ↓ | Cost | Gap ↓ | Cost | Gap ↓ |
| A-n53-k7 | 1010 | 1045 | 3.35 % | 1046 | 3.44 % | 1062 | 4.90 % | 1064 | 5.08 % |
| A-n54-k7 | 1167 | 1200 | 2.75 % | 1181 | 1.19 % | 1197 | 2.51 % | 1176 | 0.77 % |
| A-n55-k9 | 1073 | 1090 | 1.56 % | 1107 | 3.07 % | 1104 | 2.81 % | 1117 | 3.94 % |
| A-n60-k9 | 1354 | 1378 | 1.74 % | 1392 | 2.73 % | 1389 | 2.52 % | 1380 | 1.88 % |
| A-n61-k9 | 1034 | 1058 | 2.27 % | 1050 | 1.52 % | 1050 | 1.52 % | 1065 | 2.91 % |
| A-n62-k8 | 1288 | 1334 | 3.45 % | 1329 | 3.09 % | 1330 | 3.16 % | 1321 | 2.50 % |
| A-n63-k9 | 1616 | 1655 | 2.36 % | 1651 | 2.12 % | 1659 | 2.59 % | 1659 | 2.59 % |
| A-n63-k10 | 1314 | 1339 | 1.87 % | 1358 | 3.24 % | 1344 | 2.23 % | 1351 | 2.74 % |
| A-n64-k9 | 1401 | 1442 | 2.84 % | 1434 | 2.30 % | 1435 | 2.37 % | 1443 | 2.91 % |
| A-n65-k9 | 1174 | 1205 | 2.57 % | 1206 | 2.65 % | 1199 | 2.09 % | 1217 | 3.53 % |
| A-n69-k9 | 1159 | 1201 | 3.50 % | 1199 | 3.34 % | 1186 | 2.28 % | 1196 | 3.09 % |
| A-n80-k10 | 1763 | 1811 | 2.65 % | 1843 | 4.34 % | 1816 | 2.92 % | 1822 | 3.24 % |
| B-n51-k7 | 1032 | 1026 | -0.58 % | 1027 | -0.49 % | 1031 | -0.10 % | 1025 | -0.68 % |
| B-n52-k7 | 747 | 763 | 2.10 % | 759 | 1.58 % | 763 | 2.10 % | 758 | 1.45 % |
| B-n56-k7 | 707 | 736 | 3.94 % | 723 | 2.21 % | 741 | 4.59 % | 732 | 3.42 % |
| B-n57-k7 | 1153 | 1167 | 1.20 % | 1159 | 0.52 % | 1175 | 1.87 % | 1166 | 1.11 % |
| B-n57-k9 | 1598 | 1651 | 3.21 % | 1634 | 2.20 % | 1625 | 1.66 % | 1623 | 1.54 % |
| B-n63-k10 | 1496 | 1581 | 5.38 % | 1580 | 5.32 % | 1576 | 5.08 % | 1587 | 5.73 % |
| B-n64-k9 | 861 | 930 | 7.42 % | 920 | 6.41 % | 920 | 6.41 % | 922 | 6.62 % |
| B-n66-k9 | 1316 | 1349 | 2.45 % | 1338 | 1.64 % | 1345 | 2.16 % | 1359 | 3.16 % |
| B-n67-k10 | 1032 | 1073 | 3.82 % | 1061 | 2.73 % | 1069 | 3.46 % | 1088 | 5.15 % |
| B-n68-k9 | 1272 | 1312 | 3.05 % | 1323 | 3.85 % | 1317 | 3.42 % | 1346 | 5.50 % |
| B-n78-k10 | 1221 | 1265 | 3.48 % | 1286 | 5.05 % | 1277 | 4.39 % | 1269 | 3.78 % |
| E-n51-k5 | 521 | 535 | 2.62 % | 547 | 4.75 % | 545 | 4.40 % | 551 | 5.44 % |
| E-n76-k7 | 682 | 707 | 3.54 % | 718 | 5.01 % | 712 | 4.21 % | 717 | 4.88 % |
| E-n76-k8 | 735 | 753 | 2.39 % | 761 | 3.42 % | 759 | 3.16 % | 763 | 3.67 % |
| E-n76-k10 | 830 | 847 | 2.01 % | 863 | 3.82 % | 861 | 3.60 % | 859 | 3.38 % |
| E-n76-k14 | 1021 | 1052 | 2.95 % | 1049 | 2.67 % | 1068 | 4.40 % | 1073 | 4.85 % |
| E-n101-k8 | 815 | 866 | 5.89 % | 874 | 6.75 % | 887 | 8.12 % | 869 | 6.21 % |
| E-n101-k14 | 1067 | 1136 | 6.07 % | 1143 | 6.65 % | 1157 | 7.78 % | 1131 | 5.66 % |
| F-n72-k4 | 237 | 271 | 12.55 % | 269 | 11.90 % | 266 | 10.90 % | 281 | 15.66 % |
| F-n135-k7 | 1162 | 1307 | 11.09 % | 1358 | 14.43 % | 1390 | 16.40 % | 1364 | 14.81 % |
| M-n101-k10 | 820 | 873 | 6.07 % | 905 | 9.39 % | 891 | 7.97 % | 851 | 3.64 % |
| M-n121-k7 | 1034 | 1129 | 8.41 % | 1225 | 15.59 % | 1149 | 10.01 % | 1116 | 7.35 % |
| M-n151-k12 | 1015 | 1112 | 8.72 % | 1126 | 9.86 % | 1190 | 14.71 % | 1103 | 7.98 % |
| M-n200-k16 | 1274 | 1473 | 13.51 % | 1430 | 10.91 % | 1580 | 19.37 % | 1430 | 10.91 % |
| M-n200-k17 | 1275 | 1482 | 13.97 % | 1466 | 13.03 % | 1580 | 19.30 % | 1416 | 9.96 % |
| Avg. Gap | 0.00 % | | 4.49% | | 4.93% | | 5.44% | | 4.77% |