# OpenReview forum: "RL4CO: an Extensive Reinforcement Learning for Combinatorial Optimization Benchmark"
_NeurIPS.cc/2023/Track/Datasets_and_Benchmarks — Submitted to NeurIPS 2023 Datasets and Benchmarks_

### Official Review · Reviewer_mnwK · 2023-07-16
**Good paper in general**

**Rating:** 8
**Confidence:** 5
**Correctness:** Yes.
**Clarity:** Yes.

**Strengths:**

1. The paper's comprehensive selection of benchmarking problems, ranging from TSP to real-world hardware design, ensures that it caters to a wide range of users with varying requirements.

2. The paper's approach of decoupling the NCO methods into independent modules is highly commendable. By utilizing state-of-the-art software libraries and implementing best practices, the authors facilitate a streamlined and efficient coding process.

3. The paper's experimental design is well-organized and focuses on addressing central issues in NCO, such as sample efficiency, generalization ability, and boosting. By targeting these core challenges, the paper provides valuable insights and contributes to the advancement of knowledge in the field.


**Additional Feedback:**

None.

**Documentation:**

Yes.

**Ethics:**

No.

**Limitations:**

No obvious limitations were identified in the paper. However, the paper can be further improved by taking into consideration the comments provided in the "Opportunities for Improvement" section.

**Opportunities For Improvement:**

1. The paper would benefit from a comparison with benchmark studies in the related field.

2. As a benchmark paper, it would be beneficial to include more baseline methods in the evaluation, such as “https://arxiv.org/abs/2106.05126“, "https://arxiv.org/abs/2210.07686" and "https://arxiv.org/abs/2110.02544". Additional baselines could be presented in the main paper, supplementary materials, or online pages associated with the work.

3. More future directions deserve exploration in RL4CO, such as hardware implementation and parallelism issues. Additionally, advanced types of CO tasks, such as multiobjective optimization or diversity optimization, are suggested to be considered for further investigation.


**Relation To Prior Work:**

Yes.

**Summary And Contributions:**

This paper introduces an extensive benchmark platform for Neural Combinatorial Optimization (NCO), which deserves recognition for its contributions to the field. The key contributions of the paper include:

1. The provision of a benchmark suite for NCO, encompassing both classic combinatorial optimization problems and real-world tasks focusing on hardware design problems.

2. The introduction of three novel metrics that measure sample efficiency, generalization capability, and adaptability in a fair and consistent manner.

3. The development of a unified implementation with decoupled functional modules, aiming to facilitate modular NCO methods.

4. The comprehensive experimental evaluation of several well-known baselines in the NCO domain. The experiments offer valuable insights and observations that can guide future improvements in NCO research.

---

> ### Author Response · Authors · 2023-08-23
> **Thanks for your review!**
>
> Thank you for reviewing our paper - We are thrilled that you have found our paper has several key contributions to the field!
>
> > The paper would benefit from a comparison with benchmark studies in the related field.
> >
>
> Thanks for the suggestion! After the submission, we found recent specific software libraries in the related field. We made a new section to explain the commonality and differences between the related works in Appendix A.
>
> ---
>
> > As a benchmark paper, it would be beneficial to include more baseline methods in the evaluation
> >
>
> We do agree with you on this. Ideally, our benchmark would encompass tens of papers in the area. However, our choice had to be narrowed down because of time and resource constraints for the time being. Let us explain for what standard we decided on the benchmarked algorithms. Our general rule was (1) popularity, (2) higher problem coverage, (3) acceptance of the original paper, and (4) whether the algorithm enhances the learning algorithm itself.
>
> We concluded that the benchmarked AM, POMO, and Sym-NCO had met our rules; hence, we prioritized benchmarking in this work. About other works, we are already working with new collaborators we found after our RL4CO release. We currently maintain and plan to extend the model zoo (https://github.com/kaist-silab/rl4co/tree/main/rl4co/models/zoo). More importantly, we believe the zoo serves as a template for implementing RL models for CO. As shown in the zoo, our abstraction patterns are reusable for various models. Hence, the community can contribute and provide more models using this template.
>
> Finally, we have also expanded to search methods, in particular Active Search [1] and Efficient Active Search [2]! This allows us to showcase the flexibility of our framework in adapting to new tasks, as well as demonstrating the scalability to large-scale instances.
>
> ---
>
> > More future directions deserve exploration in RL4CO, such as hardware implementation and parallelism issues. Additionally, advanced types of CO tasks, such as multiobjective optimization or diversity optimization, are suggested to be considered for further investigation.
> >
>
> Thanks for the suggestion! We also believe implementing advanced types of CO tasks would significantly expand the scope and applicability of RL4CO.
>
> In an attempt to the exploration of advanced types of CO tasks, we are preparing extensions of Pickup and Delivery Problems (PDP) with practical constraints, including multi-depots and capacities of the vehicles as done in [this notebook](https://github.com/kaist-silab/rl4co/blob/main/notebooks/tutorials/2-solving-new-problem.ipynb).
>
> [1] Bello, Irwan, et al. "Neural combinatorial optimization with reinforcement learning." *International Conference on Learning Representations*, 2017.
>
> [2] A. Hottung, Y.-D. Kwon, and K. Tierney. Efficient active search for combinatorial optimization problems. *International Conference on Learning Representations*, 2022.

---

> > ### Comment · Reviewer_mnwK · 2023-08-30
> >
> > Thank you. I will retain my score and eagerly await the future extension of the benchmark, which aims to include more baselines and encompass various CO tasks.

---

> > > ### Author Response · Authors · 2023-08-30
> > > **Thank you!**
> > >
> > > We are also looking forward to extending RL4CO!  We warmly welcome collaboration and would be delighted if you consider exploring our library further. Feel free to reach out to us - even outside OpenReview. Thank you once again for your valuable insights and the time you've dedicated to reviewing our work!

---

### Official Review · Reviewer_AoJ6 · 2023-07-17
**Reject: scalability, few examples, no GNN**

**Rating:** 3
**Confidence:** 3
**Correctness:** yes
**Clarity:** yes

**Strengths:**

Provide  benchmarks for CO problems.

**Additional Feedback:**

no

**Documentation:**

yes

**Limitations:**

Scalability is an important issue in combinatorial optimization (CO) problems when using RL. The experiments of this work use the graphs with 50 nodes and then extend it to 1000 nodes. The size of graphs is too small. Some current research works can deal with graphs with 10000 nodes. Therefore, the experiments are too weak, and this work does not have contribution for current research.

In addition, this work uses only two CO problems as examples, which is few; and the Graph neural networks (GNNs) are not used for this kind of problems, which is also an important issue.

Please refer to the following works: Monte Carlo Policy Gradient Method for Binary Optimization, Learning combinatorial optimization algorithms over graphs

**Opportunities For Improvement:**

1) eqn (2), the discount factor is needed, and it may divergence otherwise.
2) In section 2, authors said solve CO with Autoregressive Sequence Generation, why use it, not others?

**Relation To Prior Work:**

yes

**Summary And Contributions:**

Authors present RL4CO, an extensive reinforcement learning (RL) for combinatorial optimization (CO) benchmark, and also systematically benchmark sample efficiency, zero-shot generalization, and adaptability to changes in data distributions of various models.

However, the model and approach are not detailed. In experiments, the size of the problem is too small.

---

> ### Author Response · Authors · 2023-08-23
> **Thanks for your review! (1/2)**
>
> Thank you for your time spent reviewing our paper! We are pleased to answer the questions.
>
> > Eqn (2), the discount factor is needed, and it may divergence otherwise.
> >
>
> Thanks for your question! We want to denote that the autoregressive solution construction process does not correspond to the state transition trajectory in the usual MDP.
>
> The autoregressive solution construction in Equation (1) is a tractable factorization of solution generation probability distribution (i.e., policy). Even though it shows some structural similarity to MDP transitions, our targeting benchmark algorithms (i.e., AM, POMO, Sym-NCO) do not consider the decoding steps as the MDP transition, as explained in Equations 1 and 9 of [1]. Hence, our formulation doesn’t require adopting the discount factor of MDP in terms of philosophy. Also, in terms of numerical computation of reward (i.e., the negative cost), $R(a,x)$ in equation 2 do not diverge as it has non-zero, finite-value only when the decoding is done.
>
> ---
>
> > In section 2, authors said solve CO with Autoregressive Sequence Generation, why use it, not others?
> >
>
> Your question is on point. This will allow us to reflect on our missing explanations to give a broad view of the NCO field.
>
> Yes, as you suggest, there are numerous different approaches in NCO, such as learning improvement heuristics, Non-autoregressive constructive heuristics,, and learned search methods.
>
> As discussed, NCO is a vast field encompassing various neural approaches to solving CO problems. Given each approach exhibits significant algorithmic differences, we decided to focus on autoregressive approaches motivated by the factors described in the following. However, we would like to extend RL4CO to include more approaches to collaborating with our active community.
>
> The reason we prioritize autoregressive approaches are:
>
> 1. Autoregressive generation allows the solver to construct feasible solutions efficiently.
> 2. RL enables training of the NCO solver without requiring a high-performance solver.
>
> Furthermore, autoregressive sequence generation schemes are the de-facto standard approaches for our target routing problems (e.g., AM, POMO, SymNCO).
>
> Despite the academic and industrial significance of our targeted autoregressive sequence generation schemes, we have observed a high degree of fragmentation in the implementations of autoregressive sequence generation NCO models. These implementations often vary significantly from one another, creating obstacles for researchers looking to implement the models and, more crucially, hindering a fair evaluation of the models.
>
> For instance, re-implementing AM [1] using the codebase of POMO is challenging due to differences in implementation choices and vice versa. This issue becomes even more significant, considering that POMO can be seen as an algorithmic extension of AM. These findings demonstrate the necessity for a unified (re)implementation of the autoregressive sequence generation schemes.
>
> However, we recognize the importance of ensuring the codebase's extensibility to encompass a broader class of NCO solvers beyond our current scope. As a step in this direction, we have incorporated (post) search methods, including active search [2] and efficient active search [3], during the rebuttal period. The results are given in Section 4.4 of the revised manuscript, where we effectively scale pre-trained models to large-scale problems up to 1000 nodes!
>
> ---
>
> > The writting of paper needs improve
> >
>
> Thanks for the suggestion. At the time of submission, we acknowledge we focused more on our library rather than the writing since the priority was to, first and foremost, provide a user-friendly, efficient, and clean implementation of RL4CO methods. During the rebuttal, we revise the manuscript with the followings:
>
> (1) **Better opening sentences:** we provide a better introduction to NCO and a proper comparison between the other NCO approaches and our targeted autoregressive sequence generation approaches.
>
> (2) **Preliminaries with more details and citations:** we revised the manuscript to provide more details of the general structure of autoregressive policies with proper citations.
>
> (3) **In-depth explanation of code implementation of policy in the main section**: we provide an in-depth explanation of policy network architecture in Section 3.
>
> In general, we made extensive improvements to both the writing itself, the presentation, and the clarity of the paper. We invite you to check out our “Common Response” as well.

---

> > ### Author Response · Authors · 2023-08-23
> > **Thanks for your review! (2/2)**
> >
> > > In theory, why use REINFORCE instead of AC mehtods?
> > >
> >
> > The central problem of training NCO solvers bolis down how to solve the optimization problem of Equation 2. As you suggest, we can apply Actor-critic (AC) methods to solve Equation 2.
> >
> > Assuming we solve equation (2) with the AC method, we train value network $V_\phi(x)$ to predict $R(a,x)$ as precisely as possible to find $\theta^*$ successfully. Here, the precise prediction of $V_\phi(x)$ is innately difficult as it requires predicting the objective value of the solution produced by the solver $\pi_\theta$ to the given problem $x$. This becomes more difficult when $\theta$ are changing during the training. Also we empirically found that REINFORCE methods (e.g., AM) outperform AC methods (e.g., AM with critic and PPO) with sufficient training.
> >
> > ---
> >
> > | The model, and approach should be described.
> >
> > Thanks for your constructive feedback! We revised the manuscript to describe the full picture of the model (i.e., policy) structure in Section 3.1 Policy paragraph of the revised manuscript!
> >
> > ### References
> >
> > [1] W. Kool, H. Van Hoof, and M. Welling. Attention, learn to solve routing problems! *International Conference on Learning Representations*, 2019.
> >
> > [2] Bello, Irwan, et al. "Neural combinatorial optimization with reinforcement learning." *International Conference on Learning Representations*, 2017.
> >
> > [3] A. Hottung, Y.-D. Kwon, and K. Tierney. Efficient active search for combinatorial optimization problems. *International Conference on Learning Representations*, 2022.

---

> > > ### Author Response · Authors · 2023-08-28
> > > **Reminder**
> > >
> > > As the author-reviewer discussion phase is coming to an end, please let us know if there are further questions or concerns to address. Your response would greatly contribute to the progress of our paper. Thanks in advance for your time and consideration!

---

> > > ### Comment · Reviewer_AoJ6 · 2023-08-28
> > > **Some explanations do not make sense**
> > >
> > > why is it precise if using REINFORCE not AC methods. you only describe the result, it is not enough.
> > >
> > > Both REINFORCE and AC methods require predicting the objective value of the solution, and both of them are difficult.
> > >
> > > it is not in depth.
> > >
> > > showing the results of AC methods can justify it, and improve the paper

---

> > > > ### Author Response · Authors · 2023-08-28
> > > > **Clarification about our benchmarked AC methods**
> > > >
> > > > Thanks for your prompt response!
> > > >
> > > > We believe there might have been a misunderstanding regarding the Actor-Critic methods (AC):
> > > >
> > > > > Showing the results of AC methods can justify it, and improve the paper
> > > > >
> > > >
> > > > What we refer to as AM-Critic is, indeed, an AC method. It utilizes the learned critic to estimate $R(x,a)$. We also investigate a more sophisticated AC method, Proximal Policy Optimization (PPO). In these regards, we have shown extensive results, including Tables 4.1, B.1, E.1, as well as Figures 4.4, 4.5, and B.4. We also revised our manuscript to support our claim in L97-98.
> > > >
> > > > We can further clarify the theoretical advantage of using REINFORCE over Actor-Critic (AC) methods in our setting. As we discussed, accurately estimating $R(x,a)$ is important to train the NCO solvers with high performance. According to the estimation methods of $R(x,a)$, we can differentiate AC and REINFORCE.
> > > >
> > > > The AC methods (AM-PPO, AM-Critic) typically rely on the learned critics $V_\phi(x)$ to estimate $R(x,a)$ in the spirit of temporal-difference (TD) methods. TD methods can be sample-efficient and typically exhibit lower variance on estimating $R(x,a)$compared to REINFORCE. However, $V_\phi(x)$ is known to be a biased estimator of $R(x,a)$.
> > > >
> > > > On the other hand, the REINFORCE method estimates $R(x,a)$ via the Monte-Carlo method, which is known as an unbiased estimator of $R(x,a)$, instead can exhibit higher variance on estimating $R(x,a)$ than AC methods. Typically, our benchmarked algorithms, such as AM, POMO, and Sym-NCO, utilize proper baselines that can reduce the negative effect of larger variance on estimating  $R(x,a)$. Furthermore, in NCO, generating training samples can be simpler than the other RL applications, as the rollout of the environment can be done simply by plugging in the constructed solution into the CO problem.
> > > >
> > > > Due to such reasons, in our benchmarked NCO settings, REINFORCE with a proper baseline (e.g., AM, POMO, Sym-NCO) tends to perform better than AC methods (e.g., AM-PPO, AM-Critic) while enjoying reduced variance and unbiased on estimating $R(x,a)$.
> > > >
> > > > ---
> > > >
> > > > We hope the above explanation offers a comprehensive response to the concerns raised. If there are any further questions or areas of confusion, please do not hesitate to point them out. We are committed to ensuring the clarity and rigor of our work.

---

> > > > > ### Author Response · Authors · 2023-08-30
> > > > > **Reminder - less than 24 hours left**
> > > > >
> > > > > As the author-reviewer discussion phase is coming to an end, please let us know if there are further questions or concerns to address. Your response would greatly contribute to the progress of our paper. Thanks in advance for your time and consideration!

---

> > > > > > ### Comment · Reviewer_AoJ6 · 2023-08-30
> > > > > >
> > > > > > I appreciate the responses from authors And I keep the score at this moment.

---

### Official Review · Reviewer_Yy73 · 2023-07-21
**Valid benchmark for combinatorial optimization using machine learning but article requires some extra care**

**Rating:** 7
**Confidence:** 4

**Strengths:**

Fully functional NCO solver.

Brings flexibility.

Based on well known libraries: TorchRL, PyTorchLightning and Hydra.

Validation using not only classical benchmarks but also one real-world problem (hardware design).

Experiments demonstrated different performance compared to the state-of-the-art.

**Additional Feedback:**

Interesting work with relevant contributions for NeurIPS and the RL community. More care should be put on the background notions, the RL4CO benchmark description and the actual validity of (some of) the experiments.

**Clarity:**

The preliminaries section is not very clear. It lacks some more details to fully grasp the many concepts involved.

The description of the main contribution, RL4CO is too high level and does not sufficiently present the capabilities of this modular framework. Even in the appendix not much details are provided.

Figure 3.1 unfortunately does not illustrate the full framework as depicted in section 3.

**Correctness:**

The claims are properly argumented with references to the state-of-the-art.

The evaluations conducted on the well-known optimisation benchmarks are scientifically sound.

**Documentation:**

Provided documentation is short in the provided appendix but is sufficient on the GitHub repository (with install, usage and simple example).

**Ethics:**

No ethical concerns for this work that solely considers classical optimisation (routing) problems.

**Limitations:**

The authors have identified several limitations as stated in section A.1. these include the limited size of the tackled instances due to hardware constraints and the limited types of CO problems considered at this stage.

**Opportunities For Improvement:**

The description of the framework is very succinct.

The benchmark experiments section is also limited.

The field of generative hyper-heuristics based on reinforcement learning is very similar (and more specifically considering constructive heuristics as in this work). It would be of interest to consider the literature in this field as well.

Due to computational power limitations, results have been obtained on single runs for the largest instances, which make the validity of results comparisons questionable.

**Relation To Prior Work:**

Good overview of Neural combinatorial optimization  (NCO) usage, benchmarking and current limitations in terms of generalisation capabilities and lack of a unified framework.

The limitations of current implementations could however be a bit more elaborated.

**Summary And Contributions:**

This article proposes a novel benchmark for combinatorial optimization using machine learning named RL4CO.

The authors introduce the context and motivation of the work through a brief state-of-the-art study on Neural combinatorial optimization (NCO) approaches and implementations. Basics on Autoregressive Sequence Generation and NCO solvers training with reinforcement learning (RL) are  provided.

The proposed RL4CO is then briefly described.

Finally some experimental results are provided on the TSP and CVRP using five different NCO solvers with different decoding schemes.

---

> ### Author Response · Authors · 2023-08-23
> **Thanks for your review! (1/2)**
>
> Thank you for your time reading our paper; we are happy that you acknowledge the strengths of our benchmark!
>
> > The description of the framework is very succinct.
> >
>
> Thank you! We also revise the manuscript to further explain the framework's details, especially the autoregressive policy's general structure, one of the core importance of unified and reusable implementation of NCO solvers.
>
> ---
>
> > The benchmark experiments section is also limited.
> >
>
> At the time of submission, the experimental benchmark was a bit limited, especially in the size of the training methods (up to 50 nodes) due to our limited compute. We have extended the results to two new environments other than TSP and CVRP (namely, OP and PCTSP) and we focused on implementing search methods to effectively scale up to large instances - including active search and efficient active search. The results are shown in Section 4.4 and encompass large-scale evaluation of up to 1000 nodes (at this size, active search cannot work due to OOM issues).
>
> ---
>
> > The field of generative hyper-heuristics based on reinforcement learning is very similar (and more specifically considering constructive heuristics as in this work). It would be of interest to consider the literature in this field as well.
> >
>
> Thanks for the suggestion. We update the manuscript to explain a general view of the CO, and NCO field in the introduction and preliminaries sections with citations. Furthermore, we compared the related software that is used to implement NCO in the newly made section in the preliminaries.
>
> ---
>
> > Due to computational power limitations, results have been obtained on single runs for the largest instances, which make the validity of results comparisons questionable.
> >
>
> Thanks for the insight! We acknowledge that ideally, performing multiple runs for the full experiments would be optimal. However, we would like to stress that training a single model is common practice in the NCO field, due to the compute-intensive training process and our limited limitations in terms of computational resources. As a case in point, the seminal work by Kool et al. (2019) only validated a subset of experiments on two seeds, demonstrating the stable nature of training against seed variation. This stability is intrinsic to policy gradient methods. Follow-up papers like POMO and SymNCO did not detail results across multiple seeds.
>
> That said, we want to emphasize the following. Aside from the common practice in NCO, we notice that results are consistent with the literature. Finally, we additionally report multiple runs with 5 different seeds in the sample efficiency comparison where both the mean and standard deviation are presented, which reflect that as training advances, outcomes for all models show a marked trend towards convergence.
>
> ---
>
> > The preliminaries section is not very clear. It lacks some more details to fully grasp the many concepts involved.
> >
>
> Thanks for your feedback and suggestion! We extensively reworked the preliminary sections to convey details of the general architecture of the policy network. This would help the reader to grasp the related concepts.
>
> ---

---

> > ### Author Response · Authors · 2023-08-23
> > **Thanks for your review! (2/2)**
> >
> > > The description of the main contribution, RL4CO is too high level and does not sufficiently present the capabilities of this modular framework. Even in the appendix not much details are provided.
> > >
> > >
> > > Figure 3.1 unfortunately does not illustrate the full framework as depicted in section 3.
> > >
> >
> > Thanks for pointing this out. For a clearer explanation of the proposed modular framework, especially related to the policy network implementation, we updated the manuscript and made a new section for explaining policy network architecture. To summarize,
> >
> > Our policy network is generally composed of `Encoder` → `Decoder`
> >
> > $$
> > \text{Encoder}(x)=
> > \text{NN}(\text{Init Emb}(x))
> > $$
> >
> > where $\text{NN}$ is the environment-agnostic neural network module, $\text{Init Emb}$ is the environment-specific module, and $x$ is the input problem.
> >
> > From our preliminary investigation, we found that the majority of autoregressive constructive NCO models can be viewed as the above function. Following this view, our encoder implementation follows the same flow to maintain higher modularity.
> >
> > The decoder structure is more complex than the encoder as it autoregressively decodes (i.e., constructs) solutions. In a high-level view, the decoder can be defined as
> >
> > $$
> > a=\text{Decoder}(h,\text{env}),
> > $$
> >
> > where $h$ is the encoder output, $\text{env}$ is the environment, and $a=(a_1,a_2,...,a_T)$ is the solution of the CO problem.
> > We found that the decoder commonly follows the patterns of
> >
> > - At every decoding step, it computes the query vector (commonly called `glimpse_q`) with the problem/algorithm-specific `context` computation routines and computes the node selection probabilities with the `action mask`.
> > - compute the attention score with `action_mask` that enforces the constraints of the CO problem.
> >
> > Our decoder allows the swap/override of the listed components that decide the behavior of specific algorithms by passing pre-coded or user-defined modules.
> >
> > Lastly, we disentangle the loss computation routine from the network itself. This allows new policy structures to be trained with off-the-shelf training schemes. This could be one of the important design choices that are not attempted before in the community. We provide a more detailed explanation in the revised manuscript of section 3.1, Policy paragraph.
> >
> > ---
> >
> > > Provided documentation is short in the provided appendix but is sufficient on the GitHub repository (with install, usage and simple example).
> > >
> >
> > We are glad you checked out our repository! We believe the library is our major contribution to the community. We have also made extensive improvements in the Appendix section. We also invite you to check out [ReadTheDocs](https://readthedocs.org/projects/rl4co/), where we provide more detailed documentation, especially for our community’s practitioners.

---

> > > ### Author Response · Authors · 2023-08-28
> > > **Reminder**
> > >
> > > As the author-reviewer discussion phase is coming to an end, please let us know if there are further questions or concerns to address. Your response would greatly contribute to the progress of our paper. Thanks in advance for your time and consideration!

---

> > > > ### Comment · Reviewer_Yy73 · 2023-08-30
> > > >
> > > > Thank you for the detailed replies to my comments.
> > > >
> > > > My concerns have been addressed through a significant amount of updates in the article itself and in the experiments.
> > > >
> > > > For these reasons I increase my ranking of your paper.

---

> > > > > ### Author Response · Authors · 2023-08-30
> > > > > **Thank you!**
> > > > >
> > > > > Thank you for dedicating your time and expertise to review our paper - your insights have been instrumental in refining our work, and we truly value your help!

---

### Official Review · Reviewer_aCTo · 2023-07-21
**Submission falls short in terms of necessary contributions -- Much improved!**

**Rating:** 6
**Confidence:** 3
**Correctness:** The paper appears to be sound.

**Strengths:**

The authors present a study into zero-shot generalization, sample efficiency and adaptability. These topics are of importance to the wider community. Likewise, combinatorial optimization problems are relevant to a large number of researchers.

The authors do a reasonable job of gathering a suite of existing RL environments into a single repository, and provide benchmarking experiments for additional understanding.

**Additional Feedback:**

** Score updated to reflect the significant updates to the manuscript **

**Clarity:**

The paper is not particularly well written, and the structuring is often confusing. For example, a new environment is introduced at the end of page 8, right before the conclusion – and not used throughout the paper.

**Documentation:**

The paper provides details and code that suggests the work is reproducible.

**Limitations:**

Some limitations of works were discussed, but not built upon or addressed.

**Opportunities For Improvement:**

I believe this submission falls short in terms of the contributions necessary for publication in the Datasets and Benchmarks Track. The paper is limited by its lack of scope throughout the experimental process and fails to provide sufficient evidence to justify the need for the library. Benchmarking open-source algorithms on relatively small TSP and CVRP environments (20/50 nodes) does not yield novel insights or make a significant contribution to the field. Moreover, both the algorithms and environments' implementation are readily available in other open-source libraries and benchmark papers, undermining the uniqueness of this work. Notably, the omission of benchmark experiments on more interesting and novel environments, such as DPP, raises questions. What is the rationale behind excluding these domains from the paper and their potential contributions to the research?

Another clear omission is a discussion regarding inference time techniques that can vastly improve search, for example EAS [1], or SGBS [2].
[1] Efficient Active Search for Combinatorial Optimization Problems, Hottung et al., ICLR 2022.
[2] Simulation-guided Beam Search for Neural Combinatorial Optimization, Choo et al. NeurIPS 2022.

The paper claims the library design is flexible, but upon examination, many components appear to be hard-coded. It remains unclear how this library surpasses the original algorithm implementations in terms of flexibility. Please can the authors clarify?

Furthermore, the motivation behind benchmarking scalability and generalizability across CO problems seems unfulfilled. For example, regarding generalization, the experiments only tackle generalization across problem sizes, and not problem distribution.

Overall, the motivation of the paper is to benchmark scalability and generalizability across CO problems, and I do not believe the authors achieve those goals.

> “Unless we explicitly mention, all NCO solvers are implemented while following original
implementations, including the code-level optimizations.”
Each of the baseline solvers are available open-source. Please can the authors clarify what part their modular framework played in evaluating these models?

The paper suggests that new metrics are introduced. Please could the authors clarify what they mean by this? Sample efficiency, zero-shot generalization, and adaptability are all well known metrics.

Some passages of the text are not well written and could be improved. For example, “Solving Eq. (2) via gradient-based methods with the gradients that are computed by differencing through the solution decoding steps as the decoding steps are not differentiable due to the discrete nature of CO.”

**Relation To Prior Work:**

The opening sentence is not informative. It lacks clarity and insight. The paper lacks sufficient comparisons and critical analysis of existing literature. Instead, it presents multiple lists of semi-related works without offering any valuable insights.

The paper falls short in properly situating the work concerning previous research in the field. For example, there are only two sentences in the Preliminaries section that reference citations. It is essential to provide proper citations and references for all the preliminary work.

Furthermore, considering that the objective of RL4CO is to tackle challenges related to autoregressive policies, it is surprising that the subsection (in Section 2) does not cite any examples of prior works.

The paper makes fairly bold, and unsubstantiated claims. For example,
1) “...in practice, it is shown that the policy gradient methods, especially estimating the gradients with REINFORCE estimator [54], outperform the value-based methods.”
Please can you provide up-to-date evidence to support this claim?
Or this claim:
2) “As a result, the majority of competitive autoregressive NCO solver implementations are showing significant coupling with network architecture and targeting CO problems”.

**Summary And Contributions:**

The authors introduce RL4CO, an RL benchmark for combinatorial optimization (CO) problems. RL4CO is designed to be efficient and easily modifiable. The focus is particularly on the importance of scalability and generalization capabilities for diverse optimization tasks, whilst providing decoupled functional modules. The authors evaluate TSP and CVRP on three evaluation metrics: sample efficiency, generalization capability, and adaptability.

---

> ### Author Response · Authors · 2023-08-23
> **Thanks for your review! (1/4)**
>
> > I believe this submission falls short in terms of the contributions necessary for publication in the Datasets and Benchmarks Track. The paper is limited by its lack of scope throughout the experimental process and fails to provide sufficient evidence to justify the need for the library.
> >
>
> Thanks for raising an essential point. We also agree that, at the time of writing, we focused more on the library and usability side and less on the manuscript which may have resulted in the paper losing scope and making it a harder read, and, more importantly, it may have been harder to see the need of a library. We made extensive modifications to clarify our stance in Introductions, Preliminaries, as well as throughout the paper. Thanks to your suggestion, we have also added a new section at the beginning of the Appendix (A.1) entitled “Why Choosing RL4CO?” - in which we incorporate several points clarifying RL4CO and why it is needed.
>
> > Benchmarking open-source algorithms on relatively small TSP and CVRP environments (20/50 nodes) does not yield novel insights or make a significant contribution to the field.
> >
>
> We believe NCO communities can address adaptation challenges, spanning training from small to large scales. While acknowledging the importance of solving large-scale problems, the efficiency of exclusive large-scale pretraining (N=500) can be questionable. In our opinion, NCO communities can focus on efficient adaptation methods that leverage NCO solvers trained in smaller problems and apply them to larger problems, enabling seamless generalization to larger domains.
>
> Renowned adaptation methods, such as active search (AS) [2] or Efficient AS (EAS) [3], hold considerable importance in facilitating adaptations for larger-scale issues. In line with this notion, we conducted experiments involving: (a) model pretraining on a smaller scale (N=50), followed by (b) assessing the adaptability of the pre-trained model using AS and EAS across N=200, 500, and 1000 in the newly made section 4.4 of the revised manuscript.
>
> > Moreover, both the algorithms and environments' implementation are readily available in other open-source libraries and benchmark papers, undermining the uniqueness of this work.
> >
>
> This work aims to (1) provide a unified and reusable implementation of various RL methods that solve CO with autoregressive sequence generation and (2) benchmark algorithms using the same code base to draw new insight.
>
> We found that open-source implementations adopt their own design choices, including the environment, policy network structure, and training pipeline. For instance, AM [1] and POMO [4] adopt different environments, attention networks, and trainer implementation. (e.g..., the default encoder network depth of POMO is deeper than AM, and the decoder attention networks are different).
>
> However, this can risk the integrity of the benchmarking of the NCO algorithms (e.g., AM using rollout baseline and POMO using the shared baseline) in terms of solution quality and execution speed as they are built on largely different code bases.  As we found from the numerical experiment, the AM model with a deeper encoder and trained with the same amount of training instances (AM-XL) generally performs better than POMO. This implies the implementation and the hyperparameters may change the well-known conclusion. This shows the need of having a unified implementation.
>
> Moreover, we also found that the current autoregressive NCO solver implementations are highly entangled with the environment implementation (e.g., AM model policy can be used to solve only TSP, CVRP, PCTSP, and OP). In RL4CO, we allow users to plug-and-play a new environment with the existing code base. We do believe such flexibility is also an important contribution to the community.
>
> ---
>
> > Notably, the omission of benchmark experiments on more interesting and novel environments, such as DPP, raises questions. What is the rationale behind excluding these domains from the paper and their potential contributions to the research?
> >
>
> Due to the restriction of pages, we wanted to focus on first investigating problem classes such as TSP and CVRP, as already done in our current manuscript. However, as you suggested, investigating a new benchmark domain also greatly contributes to the research community. We provide a detailed explanation of the DPP setting and extensive benchmark results in Appendix B of the revised manuscript and their significance. Thanks for the suggestions!

---

> > ### Author Response · Authors · 2023-08-23
> > **Thanks for your review! (2/4)**
> >
> > > Another clear omission is a discussion regarding inference time techniques that can vastly improve search, for example EAS [1], or SGBS [2]. [1] Efficient Active Search for Combinatorial Optimization Problems, Hottung et al., ICLR 2022. [2] Simulation-guided Beam Search for Neural Combinatorial Optimization, Choo et al. NeurIPS 2022.
> > >
> >
> > We agree; thanks for your suggestion! During our rebuttal, we implemented active search [2] and efficient active search [3], and reported the results in the revised manuscript in Section 4.4. This enabled us to scale tested problem sizes up to 1000 nodes. Moreover, our new boilerplate code for search methods will serve as 1) a template for new search methods research and 2) a showcase on extending RL4CO to different optimization routines.
> >
> > ---
> >
> > > The paper claims the library design is flexible, but upon examination, many components appear to be hard-coded. It remains unclear how this library surpasses the original algorithm implementations in terms of flexibility. Please can the authors clarify?
> > >
> >
> > **Flexibility of our implementation**
> >
> > Thanks for the inspection of our code! As you suggested, our previous code had some parts that were hard-coded, majorly the components that transform the environment-specific features to the embedding; in our code, they are referred to as `init_embedding`,  `context_embedding` , and `dynamic_embedding`. Between the initial submission and the rebuttal, we made several improvements to the library based on community feedback. As an example the newest version, we allow the encoder and decoder can be initialized with the user's custom embedding modules with support with PyTorch Geometric. This new design seamlessly supports famous algorithms’ components via a hard-coded manner but simultaneously allows the users to use custom methods with a single code. Similar to embeddings, we also modify the decoder so that it can be used to decode a custom environment if needed. Starting from `v0.1.0`, we have made several modifications to the library, restructuring models into their base classes (e.g. `AutoRegressivePolicy`). With the `v0.2.0` release we have fully decoupled all components, including environment embedding. Now, it is even possible to directly create a new model and environment without touching the source code of the library, directly in Google Colab in this [notebook](https://github.com/kaist-silab/rl4co/blob/main/notebooks/tutorials/2-solving-new-problem.ipynb)!
> >
> > **Comparison to the original implementations**
> >
> > During our preliminary investigation, the many autoregressive NCO models are usually implemented upon the AM implementation.  AM implementation encapsulates the environment, but is hard-coded and adopts tailed routines for each environment. Thus, using the AM implementation for different environments becomes tricky.
> >
> > To this problem, we disentangle the dependency between the environment and policy to attain higher flexibility. Specifically, we separate `init_embedding`,  `context_embedding`, and `dynamic_embedding` from the encoder and decoder while allowing them to be passed from outside the policy when the policy is instantiated.
> >
> > Regarding the training pipeline,  AM implementation is based on a custom trainer initially written 4 yrs ago and is not currently under active maintenance. This can limit the adoption of recent training methods (e.g., advanced parallelism, mixed precision training)
> >
> > Instead, we adopt PyTorch Lightning to integrate more complex training methods, including multi-GPU and mixed-precision training, logging, and checkpointing with minimal boilerplate codes. We do believe this greatly improves the productivity of NCO research, decoupling the science from the engineering.
> >
> > We want to note that POMO utilizes a vastly different implementation from the AM. However, POMO implementation is quite fragmented as they utilize different code bases for each environment.
> >
> > > Furthermore, the motivation behind benchmarking scalability and generalizability across CO problems seems unfulfilled. For example, regarding generalization, the experiments only tackle generalization across problem sizes, and not problem distribution.
> > >
> >
> > Thanks for the great suggestion! We also believe investigating the different synthetic distributions would be helpful, and we plan to do so in future work (we added a small paragraph under “Limitations”). We do note that  we measured the model's performance on real-world TSPLib and CVRPLib instances, not generated from the training distributions, which benchmark both 1) generalization to different numbers of nodes and 2) data distribution unseen during training (i.e. non-uniform).

---

> > > ### Author Response · Authors · 2023-08-23
> > > **Thanks for your review! (3/4)**
> > >
> > > > “Unless we explicitly mention, all NCO solvers are implemented while following original implementations, including the code-level optimizations.” Each of the baseline solvers are available open-source. Please can the authors clarify what part their modular framework played in evaluating these models?
> > > >
> > >
> > > Thanks for your question! This actually helps us in clarifying our contribution and the need for RL4CO. The statement on "optimizations" refers broadly to specific "details" or modifications implemented to ensure consistency, efficiency, and a shared approach across different model evaluations.
> > >
> > > 1. **Implementation Consistency:** For instance, we noticed variations in the implementation of Multi-head attention across models, along with modifications to the projections (such as the use of biases). To address this, we standardized it and introduced FlashAttention, which can effectively speed up the computation compared to the source implementation. We are also actively looking into integrating FlashAttention2, which was recently introduced [5].
> > > 2. **Shared Evaluation Methods:** A major advantage of our framework is sharing different evaluation methods across models. This facilitates, for example, evaluating AM with multi-greedy starts without having to reimplement it for every solver.
> > > 3. **Enhanced Modularity:** In the case of POMO, its original implementation was tailored to specific problems. We enhanced its versatility by creating modular environment embeddings, allowing researchers to apply POMO to various other environments, such as OP, without extensive reconfiguration.
> > >
> > > In essence, while evaluating different models with their unique implementations can be intricate, our modular framework streamlines this process, ensuring a consistent and robust evaluation across the board. We have modified the manuscript to reflect these changes better.
> > >
> > > ---
> > >
> > > > The paper suggests that new metrics are introduced. Please could the authors clarify what they mean by this? Sample efficiency, zero-shot generalization, and adaptability are all well known metrics.
> > > >
> > >
> > > Thanks for pointing this out. Yes, those are not new metrics. We fixed this problem in the abstract. We meant is that we test models on metrics that had not been tested before, “new”, so to speak, from the model’s perspective. For instance, generalization performance was not shown in the original POMO paper.
> > >
> > > ---
> > >
> > > > Some passages of the text are not well written and could be improved. For example, “Solving Eq. (2) via gradient-based methods with the gradients that are computed by differencing through the solution decoding steps as the decoding steps are not differentiable due to the discrete nature of CO.”
> > > >
> > >
> > > You are correct; thanks for pointing this out! We update the sentences as “Solving Eq. (2) via gradient-based optimization method, calculating the gradient of the objective function w.r.t. $\theta$ is required. However, due to the discrete nature of the CO, the computation of the gradient is not straightforward and often requires certain levels of approximation [6, 7]. Even though few researchers show breakthroughs for solving (2) with gradient-based optimization, they are restricted to some relatively simpler cases of CO problems.”
> > >
> > > ---
> > >
> > > > The paper is not particularly well written, and the structuring is often confusing. For example, a new environment is introduced at the end of page 8, right before the conclusion – and not used throughout the paper.
> > > >
> > >
> > > We agree with you on this. Indeed, we focus firstly on releasing a fully functional library as our main contribution to NCO and writing was not our primary objective; undoubtedly, writing is equally important to convey our message. We now made several changes throughout the paper. Our primary intention for having the electrical design automation (EDA) problem is to introduce the TSP-like problem with practical application and restriction — evaluation of the solution (i.e., computing $R(x,a)$ ) is costly — to the community. Given the space limitations and the peculiarities of EDA (i.e., a proper introduction to the problem has to be made for those unfamiliar with the problem), we have dedicated a full chapter - Appendix B - to the problem!

---

> > > > ### Author Response · Authors · 2023-08-23
> > > > **Thanks for your review! (4/4)**
> > > >
> > > > > The opening sentence is not informative. It lacks clarity and insight. The paper lacks sufficient comparisons and critical analysis of existing literature. Instead, it presents multiple lists of semi-related works without offering any valuable insights. The paper falls short in properly situating the work concerning previous research in the field. For example, there are only two sentences in the Preliminaries section that reference citations. It is essential to provide proper citations and references for all the preliminary work. Furthermore, considering that the objective of RL4CO is to tackle challenges related to autoregressive policies, it is surprising that the subsection (in Section 2) does not cite any examples of prior works.
> > > > >
> > > >
> > > > Thanks for the constructive feedback!  We extensively revised the opening paragraphs of the manuscript so that they can well explain the existing approaches and comparisons to provide more value and insight to the readers.  For the preliminaries, we also provide references for the contents we discussed so that the readers can refer to the related work more easily, including an introduction to autoregressive policies. Finally, we also discuss in Appendix A.5 related works in terms of other recent software libraries we came across to better understand the stance of RL4CO.
> > > >
> > > > ---
> > > >
> > > > > “...in practice, it is shown that the policy gradient methods, especially estimating the gradients with REINFORCE estimator [54], outperform the value-based methods”. Please can you provide up-to-date evidence to support this claim?
> > > > >
> > > >
> > > > In the Attention Model paper [1], the Table 2 of the Appendix provides the results of actor-critic (AM-Critic) along with pure policy-gradient results (AM-Rollout). From the table, AM-Rollout shows better solution quality compared to AM-Critic, which we also found out in practice in both routing problems and EDA (Appendix B). We also found a similar trend during our experiments: PPO generally performs slightly worse than AM, even using the same policy network architecture.
> > > >
> > > > ---
> > > >
> > > > > “As a result, the majority of competitive autoregressive NCO solver implementations are showing significant coupling with network architecture and targeting CO problems”. Please can you provide up-to-date evidence to support this claim?”
> > > > >
> > > >
> > > > We can take as an example the three major methods we benchmarked. We found this tendency from the open-source implementation of AM, POMO, and Sym-NCO.
> > > >
> > > > - The policy model of AM implementation (https://github.com/wouterkool/attention-learn-to-route/blob/6dbad47a415a87b5048df8802a74081a193fe240/nets/attention_model.py#L63) infers the problem types from the given environment so that it can apply environment-specific NN-based routines.
> > > > - The POMO implementation (https://github.com/yd-kwon/POMO/tree/master/NEW_py_ver, https://github.com/yd-kwon/POMO/blob/master/NEW_py_ver/TSP/POMO/TSPModel.py) used the separate code bases for each problem type (i.e., CVRP and TSP) due to the coupling.
> > > > - The Sym-NCO implementation is based on the AM and POMO implementation; hence the same couplings are found (https://github.com/alstn12088/Sym-NCO/tree/main). In particular, the library is split into two implementations, each with its codebase (AM or POMO), which is harder in practice not only to maintain but also to provide fair comparisons
> > > >
> > > > ### Reference
> > > >
> > > > [1] W. Kool, H. Van Hoof, and M. Welling. Attention, learn to solve routing problems! *International Conference on Learning Representations*, 2019.
> > > >
> > > > [2] I. Bello, H. Pham, Q. V. Le, M. Norouzi, and S. Bengio. Neural combinatorial optimization with reinforcement learning, 2017.
> > > >
> > > > [3] A. Hottung, Y.-D. Kwon, and K. Tierney. Efficient active search for combinatorial optimization problems. *arXiv preprint arXiv:2106.05126*, 2021.
> > > >
> > > > [4] Y.-D. Kwon, J. Choo, B. Kim, I. Yoon, Y. Gwon, and S. Min. POMO: Policy optimization with multiple optima for reinforcement learning. *Advances in Neural Information Processing Systems*, 33:21188–21198, 2020.
> > > >
> > > > [5] Dao, Tri. "FlashAttention-2: Faster Attention with Better Parallelism and Work Partitioning." *arXiv preprint arXiv:2307.08691* (2023).
> > > >
> > > > [6] Vlastelica, M., Paulus, A., Musil, V., Martius, G. and Rolínek, M., 2019. Differentiation of blackbox combinatorial solvers. *arXiv preprint arXiv:1912.02175*.
> > > >
> > > > [7] Qiu, R., Sun, Z. and Yang, Y., 2022. Dimes: A differentiable meta solver for combinatorial optimization problems. *Advances in Neural Information Processing Systems*, *35*, pp.25531-25546.

---

> > > > > ### Author Response · Authors · 2023-08-28
> > > > > **Reminder**
> > > > >
> > > > > As the author-reviewer discussion phase is coming to an end, please let us know if there are further questions or concerns to address. Your response would greatly contribute to the progress of our paper. Thanks in advance for your time and consideration!

---

> > > > > > ### Comment · Reviewer_aCTo · 2023-08-29
> > > > > > **Much improved manuscript**
> > > > > >
> > > > > > I thank the authors for engaging in the discussion phase, you have provided many helpful clarifications. Furthermore, you have taken onboard the reviews to drastically update the paper with a better overall structure, improved motivation and more clarity on topics (i.e. response 3/4). Also adding additional experiments transferring to larger problems (response 1/4) and implementing inference time search algorithms.
> > > > > >
> > > > > > This version of the manuscript is vastly improved and I have increased my score to reflect this.

---

> > > > > > > ### Author Response · Authors · 2023-08-29
> > > > > > >
> > > > > > > Thank you for reviewing our manuscript! We'd like to express our gratitude for your input, which has greatly assisted us in identifying our previous shortcomings and providing clear directions for improvement. Once again, thank you for your help!

---

### Official Review · Reviewer_ajbT · 2023-08-01
**The current shape of this work is a bit off the research track (reproduction, scalability, and graph learning )**

**Rating:** 4
**Confidence:** 4
**Correctness:** There is no good evidence of "correct…

**Strengths:**

1. The target problem is fundamentally important.
2. It is good to have a modularized development with good documentation on Github (and website).

**Additional Feedback:**

In the rebuttal, the authors are expected to explain about "reproduction of popular RL4CO papers (there are over 100 highly relevant ones)" scalability, and "graph embedding or graph neural networks".

**Clarity:**

The presentation is focusing on less important aspects. It is expected to address the fundamental issues for the RL4CO field, like reproduction, scalability, or graph learning.

**Documentation:**

The website is clean and organized

**Ethics:**

It is about CO problems. It seems to have no ethics issues.

**Limitations:**

1. There are various RL algorithms for CO problems. It is highly expected to reproduce a dozen of those algorithms for the RL4CO community. Note that "reproduction" of important papers is very important for such a benchmark project.
2. Many papers in this domain have pointed out that graph representation or graph neural networks are important (since many CO problems are defined on graphs), which is a part of the policy network. This project does not provide good support for graph neural networks.
3. Well, problem size of 200 is quite small. "Scalability" is really important, and the sizes of 1000, 5000 are relevant to real-world.
4. As a "benchmark" project, it is also necessary to include solvers' performance, say Gurobi and SCIP.

**Opportunities For Improvement:**

The coverage of this paper is narrow. The authors are expected to check the following three survey papers:

[1] Mazyavkina, Nina, et al. "Reinforcement learning for combinatorial optimization: A survey." Computers & Operations Research 134 (2021): 105400.
[2] Bengio, Yoshua, Andrea Lodi, and Antoine Prouvost. "Machine learning for combinatorial optimization: a methodological tour d’horizon." European Journal of Operational Research 290.2 (2021): 405-421.
[3] Peng, Yun, Byron Choi, and Jianliang Xu. "Graph learning for combinatorial optimization: a survey of state-of-the-art." Data Science and Engineering 6, no. 2 (2021): 119-141.

and also the following competition:

Machine Learning for Combinatorial Optimization

https://www.ecole.ai/2021/ml4co-competition/

**Relation To Prior Work:**

Many popular papers are not reproduced.

**Summary And Contributions:**

In this paper, the authors aim to provide a benchmark of RL4CO, which is fundamentally important.
The authors claim emphasize on scalability and generalization capabilities, and also provide benchmark for sample efficiency, zero-shot generalization, and adaptability to distribution changes.

__After the rebuttal phase__, the reviewer appreciates the authors detailed responses (including those to other reviewers), but still holds __BIG__ concerns on "__reproduction (since it is dataset and benchmark track) and scalability (as also pointed out be other reviewers)__". Therefore, the score is decreased. As it is pointed out by the authors, the major contributions are about software engineering like "modularity; Easier Comparison; Leveraging Standardized and Verified Open-Source Libraries"...which do not align well with this research track.

---

> ### Author Response · Authors · 2023-08-23
> **Thanks for your review! (1/2)**
>
> Thank you for your review! We are happy to answer your questions and clarify your concerns.
>
> > The coverage of this paper is narrow. The authors are expected to check the following three survey papers:
> >
> >
> > [1] Mazyavkina, Nina, et al. "Reinforcement learning for combinatorial optimization: A survey." Computers & Operations Research 134 (2021): 105400. [2] Bengio, Yoshua, Andrea Lodi, and Antoine Prouvost. "Machine learning for combinatorial optimization: a methodological tour d’horizon." European Journal of Operational Research 290.2 (2021): 405-421. [3] Peng, Yun, Byron Choi, and Jianliang Xu. "Graph learning for combinatorial optimization: a survey of state-of-the-art." Data Science and Engineering 6, no. 2 (2021): 119-141.
> >
>
> Thank you for suggesting the survey papers to guide us and our readers in understanding the field of Neural Combinatorial Optimization (NCO)!
>
> We have updated our manuscript to introduce NCO methods and to explain our focus on "RL methods that solve CO with autoregressive sequence generation (Section 2). We also acknowledge the value of considering a larger class of NCO methods. However, at this moment, our concentration has remained on autoregressive sequence generation for the following reasons.
>
> As discussed in the suggested survey papers, NCO is a vast field encompassing various neural approaches to solving CO problems. To achieve our goal of providing a unified implementation with decoupled functional module implementation of NCO, it becomes helpful for us to narrow down our focus to a specific subset of NCO classes. Therefore, we have chosen to concentrate on NCO solvers that construct solutions using autoregressive sequence generation, particularly those trained with RL. This choice is motivated by several factors:
>
> 1. Autoregressive generation allows the solver to construct feasible solutions efficiently
> 2. RL enables training of the NCO solver without requiring a high-performance solver
>
> Furthermore, autoregressive sequence generation schemes are the de-facto standard approaches for our target routing problems (e.g., AM, POMO, SymNCO). In particular, autoregressive sequence generation is a standard approach for problems with complex constraints that may not be possible to solve with non-autoregressive models.
>
> Despite the academic and industrial significance of our targeted autoregressive sequence generation schemes, we have observed a high degree of fragmentation in the implementations of autoregressive sequence generation NCO models. These implementations often vary significantly from one another, creating obstacles for researchers looking to implement the models and, more crucially, hindering a fair evaluation of the models.
>
> For instance, re-implementing AM [1] using the codebase of POMO is challenging due to differences in implementation choices and vice versa. This issue becomes even more significant, considering that POMO can be seen as an algorithmic extension of AM. These findings demonstrate the necessity for a unified (re)implementation of the autoregressive sequence generation schemes.
>
> However, we recognize the importance of ensuring the codebase's extensibility to encompass a broader class of NCO solvers beyond our current scope. As a step in this direction, we have incorporated (post) search methods, including active search [2] and efficient active search [3], during the rebuttal period.
>
> ---
>
> > There are various RL algorithms for CO problems. It is highly expected to reproduce a dozen of those algorithms for the RL4CO community. Note that "reproduction" of important papers is very important for such a benchmark project.
> >
>
> We are currently maintaining the model zoo (https://github.com/kaist-silab/rl4co/tree/main/rl4co/models/zoo). Even though the implemented algorithms in the zoo may not exhaustive enough right now to cover all the major important papers, we plan to extend RL4CO to more models in the near future.
>
> More importantly, we consider the zoo a template for implementing several baseline methods starting from autoregressive sequence generation.  As shown in the zoo, our abstraction patterns are reusable for various models. Hence, we believe that by using this template, the community can contribute and provide more models - as demonstrated by our new implementation of the AS and EAS search methods.

---

> > ### Author Response · Authors · 2023-08-23
> > **Thanks for your review! (2/2)**
> >
> > > Many papers in this domain have pointed out that graph representation or graph neural networks are important (since many CO problems are defined on graphs), which is a part of the policy network. This project does not provide good support for graph neural networks.
> > >
> >
> > Thanks for your suggestion! We have included RL4CO abstracts thme policy, which takes the problem instance and constructs the solution into two large chunks; encoder and decoder. The supports of graph neural networks (GNN) are seamlessly doable by replacing the encoder network. We added official support for PyTorch Geometric and provided two additional GNN encoders (GCN and MPNN)  in our code. We also provide a notebook that uses a GCN encoder as the encoder of AM [here](https://github.com/kaist-silab/rl4co/blob/main/notebooks/tutorials/3-change-encoder.ipynb) and can be extended with any new user-made one.
> >
> > ---
> >
> > > Well, problem size of 200 is quite small. "Scalability" is really important, and the sizes of 1000, 5000 are relevant to real-world.
> > >
> >
> > To our knowledge, solving larger-sized problems such as 1000 and 5000 is commonly done by applying post-processing methods (e.g., fine-tuning or search method) to the pre-trained autoregressive sequence generation models.  During the rebuttal period, we prepared the search methods AS and EAS and applied them to the pre-trained models to solve larger instances - we added a new Section 4.4 to showcase the results up to 1000 nodes.
> >
> > ---
> >
> > > As a "benchmark" project, it is also necessary to include solvers' performance, say Gurobi and SCIP
> > >
> >
> > We agree with your suggestion. Table 4.1 now reports the results of classical methods (e.g., Gurobi for TSP, HGS for CVRP). The same goes for Table E.1 (newly added OP and PCTSP results).
> >
> > ### References
> >
> > [1] W. Kool, H. Van Hoof, and M. Welling. Attention, learn to solve routing problems! *International Conference on Learning Representations*, 2019.
> >
> > [2] I. Bello, H. Pham, Q. V. Le, M. Norouzi, and S. Bengio. Neural combinatorial optimization with reinforcement learning, 2017.
> >
> > [3] A. Hottung, Y.-D. Kwon, and K. Tierney. Efficient active search for combinatorial optimization
> > problems. *arXiv preprint arXiv:2106.05126*, 2021.

---

> > > ### Author Response · Authors · 2023-08-28
> > > **Reminder**
> > >
> > > As the author-reviewer discussion phase is coming to an end, please let us know if there are further questions or concerns to address. Your response would greatly contribute to the progress of our paper. Thanks in advance for your time and consideration!

---

> > > > ### Author Response · Authors · 2023-08-30
> > > > **Reminder - less than 24 hours left**
> > > >
> > > > As the author-reviewer discussion phase is coming to an end, please let us know if there are further questions or concerns to address. Your response would greatly contribute to the progress of our paper. Thanks in advance for your time and consideration!

---

### Author Response · Authors · 2023-08-23
**Response to All Reviewers (1/3)**

# Common Response

First, we would like to thank the reviewers for their valuable comments and feedback on our work! We are thrilled to receive an overall positive evaluation of our work.

We would like to express our gratitude to the reviewers for acknowledging the significance of our work [ajbT, aCTo, mnwK], its modularized implementation [ajbT, Yy73], comprehensive documentation [ajbT], novelty [Yy73], problem-solving capabilities [Yy73] and its valuable contribution to the community [aCTo, mnwK]. These recognitions greatly encourage us.

Based on the reviewers’ feedback, we identified several key points and areas for improvement in our library and our paper, which we list below.

### Why is our RL4CO benchmark needed?

In this paper, we introduce RL4CO, a *modular*, *flexible*, and *unified* implementation of NCO. In design, we intended to use RL4CO for various purposes which range from the research to the production end. RL4CO enables users to have the following benefits.

**Minimal Implementation of Boilerplate Codes for NCO:** As with the other RL projects, implementation of NCO with RL involves designing and coding the systems composed mainly of agents (i.e., policy) and environment (i.e., CO problem). However, this often involves a serious amount of engineering, especially to attain higher executions in training routines. Moreover, we found that a significant chunk of NCO solvers are based on AM or POMO implementation and the subroutines that have been done in the implementation. Regarding those practical aspects, RL4CO provides a modularized code base for each routine of NCO, including environment, policy network architecture, RL algorithm, and training. So that the users can easily mix and match state-of-the-art NCO practices and user-defined modules while having full control over the entire RL pipeline.

**Easier Comparison Among NCO Algorithms:** Current NCO research shows a tendency to rely on two cornerstone implementations: AM and POMO. However, due to differences in implementation (e.g., network architecture, training scheme), direct head-to-head comparisons among the algorithms might not be straightforward. For example, applying POMO's state augmentation to AM's policy to reveal the effect of augmentation from the baseline selections can be challenging.

RL4CO provides a unified implementation of NCO models (and their subroutines) to offer higher adaptability of routines from one algorithm to another. We believe this will promote easier comparisons among the models during the development of novel NCO solvers to address various CO problems.

**Leveraging Standardized and Verified Open-Source Libraries:** We have made the decision to utilize standardized and reputable open-source libraries based on extensive research and expertise at the edge of software engineering and research, such as the recent TorchRL [1]. We believe this design choice will yield various practical benefits both in research and production. For instance, by disentangling the RL algorithm from the training subroutines of RL4CO, NCO solvers can undergo training using an array of state-of-the-art training methods supported by PyTorch Lightning [2]. Additionally, deploying the trained NCO solvers to production becomes seamless through the utilization of tools such as TorchServe, to name just a couple of examples.

---

> ### Author Response · Authors · 2023-08-23
> **Response to All Reviewers (2/3)**
>
> ### Improvements to the RL4CO library
>
> We have made several improvements to the library both before and during the rebuttal period. We thank the reviewers for pointing out ways to improve our library. We are also grateful to new active users in the RL4CO community who gave us feedback and helped us both improve our library and gave us evidence that our library will enjoy an active community!
>
> - **Documentation**: we made a complete overhaul of it on [ReadTheDocs](https://rl4co.readthedocs.io/en/latest/), now it is much more informative and easy to follow. Moreover, we provide tutorial notebooks that are fully runnable in Colab, starting from simple quickstart to how to implement a new environment and model from scratch, with the ability to even change decoders and leverage PyTorch Geometric.
> - **Modularity of the framework**:  Most notably, starting with the release of `v0.1.0`, we introduced new trainers and base classes and restructured the models to make interdependencies minimal and as customizable as possible. For example, now the `AutoRegressiveModel` base class (e.g., for AM, POMO, and SymNCO) is modularized such that the models are subclasses of it - in other words, creating a new model from the base model will be simple - for example, in POMO one only needs to modify the loss function with no need for restructuring the model itself. Moreover, starting from `v0.2.0`, we have made environment embeddings fully modular, enabling from-scratch implementation of new problems and models.
> - **Active search methods:** we have also implemented and benchmarked search methods for NCO including active search (AS) [3] and efficient active search (EAS) [4]. We also made an extension to the original EAS implementation [4] that supports the use of various baselines including, shared, symmetric baselines.  The implemented search modules are compatible with the children classes of the new `AutoRegressiveModel`, which enables perform (E)AS with AM, POMO, and SymNCO seamlessly. It is noteworthy that the official EAS implementation only supports POMO.
>
> Note that these are just a preview - we have worked hard on this library since we submitted the preprint and during the rebuttal. The [extensive commits](https://github.com/kaist-silab/rl4co/commits/main) , show more than 700 to the date of writing!
>
> ---
>
> ### Improvements to the paper
>
> We also made several improvements based on the reviewers’ feedback. We list here the changes made point-by-point:
>
> - **Revising the opening sentences:** We revise the opening sentences to deliver broader ideas related to CO and NCO and RL approaches for NCO.
> - **Enhancing citations in preliminaries:** we provide the citations of the canonical approaches discussed in the preliminaries section.
> - **Clarification of the explanation:** in the preliminaries section, we clarify the explanation of the gradient-based methods with the proper citations.
> - **Better explanation of framework, including the policy paragraph in section 3.1:** we provide an in-depth explanation of the policy structure and also highlight what hinders the modularity of NCO solvers and how RL4CO secures the higher modularity.
> - **Evaluation of new methods:** during the rebuttal, we implemented search methods AS and EAS evaluated them. We provide the new results in section 4.4. Notably, now we test large-scale problems up to 1000 nodes.
> - ************OP and PCTSP:************ we added new benchmark results on these two environments in Table E.1
> - **Addition of related work:** we now provide some literature review of recent related software in Appendix
> - **Adjusting the positioning of the checklists and Acknowledgment:** we positioned the checklists and acknowledgment section after the main manuscripts and before the references

---

> > ### Author Response · Authors · 2023-08-23
> > **Response to All Reviewers (3/3)**
> >
> > ### Scaling RL4CO with more baselines
> >
> > During the rebuttal phase, we had the opportunity to refine RL4CO based on valuable constructive feedback. One of the improvements involves enhancing the modularity of the policy modules, which has led to the development of an updated model zoo with greater flexibility and modularity compared to previous versions.
> >
> > As we have demonstrated through our model zoo implementation, we anticipate that the model zoo will serve as a highly reusable template for creating new types of NCO solvers or environments. We hope that this enhancement will facilitate a much simpler implementation of NCO solutions and, furthermore, encourage active participation from the community to contribute additional 'animals' to the zoo. As an example, implemented flexible search methods such as AS and EAS that allow for trained policies to be adapted to specific instances.
> >
> > ### Future plans of RL4CO
> >
> > Regardless of the decision, we do not plan to stop the project here. We are already working on projects that are entirely based on our library, and we expect other researchers and practitioners to join soon.  We are actively collaborating with other people across the globe to extend RL4CO. We believe in the power of community participation in improving the library and ultimately aim to make RL4CO the to-go library in the RL for CO research area!
> >
> > ### References
> >
> > [1] Bou, Albert, et al. "TorchRL: A data-driven decision-making library for PyTorch." *arXiv preprint arXiv:2306.00577* (2023).
> >
> > [2] W. Falcon and The PyTorch Lightning team. PyTorch Lightning, 3 2019. URL https:// [github.com/Lightning-AI/lightning](http://github.com/Lightning-AI/lightning).
> >
> > [3] Bello, Irwan, et al. "Neural combinatorial optimization with reinforcement learning." *International Conference on Learning Representations*, 2017.
> >
> > [4] A. Hottung, Y.-D. Kwon, and K. Tierney. Efficient active search for combinatorial optimization problems. *International Conference on Learning Representations*, 2022.

---

### Decision · Program_Chairs · 2023-09-22

**Decision:**

Reject

**Comment:**

The authors aim to provide a benchmark of RL4CO.

The authors claim emphasize on scalability and generalization capabilities. They provide benchmark for sample efficiency, zero-shot generalization, and adaptability to distribution changes.

According to the existing literature, the paper is not solid to be published in the conference.